# Simultaneous measurements of new particle formation in 1-second time resolution at a street site and a rooftop site

Yujiao Zhu[1#], Caiqing Yan[2#], Renyi Zhang[2,3], Zifa Wang[4], Mei Zheng[2], Huiwang Gao[1], Yang Gao[1], Xiaohong Yao[1,5*]

[1]Key Lab of Marine Environmental Science and Ecology, Ministry of Education, Ocean University of China, Qingdao 266100, China

[2]State Key Joint Laboratory for Environmental Simulation and Pollution Control, College of Environmental Sciences and Engineering, Peking University, Beijing 100871, China

[3]Departments of Atmospheric Sciences and Chemistry, Center for the Atmospheric Chemistry and the Environment, Texas A&M University, College Station, TX 77843, USA

[4] State Key Laboratory of Atmospheric Boundary Layer Physics and Atmospheric Chemistry (LAPC), Institute of Atmospheric Physics, Chinese Academy of Sciences, Beijing, China

[5]Qiangdao Collaborative Center of Marine Science and Technology, Qingdao 266100, China

[#] both have the same contribution, * *Correspondence to*: Xiaohong Yao, Phone: (86) 532-66782565; Fax: 86-532-66782810; Email address: xhyao@ouc.edu.cn

**Abstract**

This study for the first time uses two identical Fast Mobility Particle Sizers for simultaneously measuring particle number size distributions (PNSD) at a street site and a rooftop site within 500 m distance in winter and spring times to investigate new particle formation (NPF) in Beijing. The collected datasets in 1-second time resolution allow deducting the freshly emitted traffic particle signal from the measurements at the street site and thereby evaluating the effects on NPF in urban atmospheres through a site-by-site comparison. The number concentrations of 8-20 nm newly formed particles and the apparent formation rate (FR) in the springtime were smaller at the street site than at the rooftop site. In contrast, NPF was enhanced in the wintertime at the street site with FR increased by 3-5 times, characterized by a shorter NPF time and higher new particle yields than those at the rooftop site. Our results imply that the street canyon likely exerts distinct effects on NPF under warm or cold ambient temperature because of on-road vehicle emissions, i.e., stronger condensation sinks that may be responsible for the reduced NPF in the springtime but efficient nucleation and partitioning of gaseous species that contribute to the enhanced NPF in the wintertime. The occurrence or absence of apparent growth for > 10 nm new particles is also analyzed. The oxidization of biogenic organics in the presence of strong photochemical reactions is suggested to play an important role in growing >10 nm new particles, but sulfuric acid is unlikely the main species for the apparent growth. However, the number of datasets used in this study is relatively small and larger ones are essential to draw a general conclusion.

**Key words:** new particle formation, street site, enhanced nucleation, vehicle emissions, semi-volatile organics

## 1. Introduction

New particle formation (NPF) has been measured under diverse environmental conditions, accounting for approximately 50% of the aerosol number production in the troposphere, but the chemical mechanism and species leading to aerosol nucleation and growth remain highly uncertain (Merikanto et al., 2009; Yao et al., 2010; Zhang et al., 2012). NPF occurs in two distinct stages, i.e., nucleation to form critical nuclei and subsequent growth of freshly nucleated particles to larger sizes (Kulmala et al., 2004, 2013; Zhang, 2010, 2012). In addition, the growth process competes with capture/removal of nanoparticles by coagulation with pre-existing aerosols. Currently, considerable uncertainty exists concerning the mechanism and the identity of chemical species responsible for aerosol nucleation and growth (Zhang et al., 2012; Wang et al., 2017). Sulfuric acid has been commonly considered as one of the main precursors of aerosol nucleation and growth, but is insufficient to explain the observed NPF under various ambient conditions (Kulmala et al., 2004, 2008; Zhang et al., 2012). Earlier studies indicated that $NH_3$ can enhance aerosol nucleation, but recent lab experimental and theoretical studies have suggested that amines and highly oxygenated molecules (HOMs) play vital roles in enhancing nucleation and promoting the initial growth of newly formed particles in the atmosphere (Zhang et al., 2004, 2009, 2012; Wang et al., 2010; Riipinen et al., 2011; Kulmala et al., 2013; Qiu and Zhang, 2013; Schobesberger et al., 2013; Ehn et al., 2014; Riccobono et al., 2014; Tröstl et al., 2016). Therefore, it is critical to evaluate the effects of nucleating species other than sulfuric acid and the dependence of NPF on pre-existing particles in the atmosphere. Urban street canyons provide semi-enclosed environments trapping vehicle exhausts that likely contain aromatic and aliphatic hydrocarbons, $SO_2$, NOx, amines, black carbon, etc. (Pierson et al., 1983; Stemmler et al., 2005; Burgard et al., 2006; Liu et al., 2008; Buccolieri et al., 2009; Sun et al., 2012; Gentner, et al., 2012), thus serving effectively as environmental chambers to investigate the effects of on-road vehicle emissions, i.e., gaseous species and primary particles, on NPF.

A major challenge exists for studying NPF at street sites because of the interference from primarily emitted vehicular particles. The primary particles can be generated directly in the engine during fuel combustion or can be nucleated in the air during dilution and cooling of hot exhausts (Kittelson, 1998; Kittelson et al., 2008; Arnold et al., 2012; Rönkkö et al., 2013; Vu et al., 2015). The former primary particles consist mostly of soot and mainly exist in the Aitken mode and accumulation mode, ranging from 30 nm to 500 nm, and the freshly nucleated vehicular particles during the initial 1-2s exhaust cooling

and dilution processes reportedly exhibit a nucleation mode at 10-20 nm (Shi et al., 2000; Zhu et al., 2002a, b, 2006; Vu et al., 2015). The particle number concentration (PNC) at roadside can vary a lot, depending on the traffic flow, composition and speed as well as wind speed and direction (Yao et al., 2005). Scanning Mobility Particle Sizer (SMPS) and other scanning sizers, e.g., commonly operating in 5-10 minutes time resolution and occasionally down to 2-3 minutes, had been widely used to measure particle number size distributions (PNSD) at roadside. Dramatically varying PNC at roadside alone raise a challenge to accurately measure PNSD using such low time resolution scanning sizers and the measured PNSD may severely distort from the real ones (Yao et al., 2006a, b). In addition, PNSD at roadside can also vary a lot because of the short distance between the sampling site and the traffic flow, e.g., the PNSD sometimes represent the overwhelming contribution from a single vehicle emission, but sometimes represent the combining contribution from a few vehicle emissions or the combining contribution from the traffic flow. Highly varying PNSD at roadside may further worsen the accuracy of the PNSD measured by low time resolution scanning sizers. When a high time resolution particle sizer, e.g., Fast Mobility Particle Sizer or Engine Exhaust Particle Sizer, was used, the uncertainty can be greatly reduced (Yao et al., 2006a, b). The measured PNSD in 1-seond time resolution can allow extracting the new particle signal from the mixing signal of newly formed particles, pre-existing ambient particles and freshly emitted particles from combustion (Liu et al., 2014).

NPF has been recognized as a major contributor to severe haze in Beijing (Guo et al., 2014). Furthermore, organic species have been shown to play a key role in the growth and aging of nanoparticle (Peng et al., 2016). Ultrafine particles (<100 nm) have been implicated to have adverse human health impacts through the deposition in the pulmonary region and penetration into the bloodstream (Oberdörster et al., 2004; Schlesinger et al., 2006; Zhang et al., 2012, 2015). On NPF days, the ultrafine particle number concentration is sharply increased and potential health impacts are largely dependent on particle loadings (Guo et al., 2014). How enhancement and scavenging affect the net production of newly formed particles is still poorly understood, althrough a large quantity of studies have focused mainly on the effects of enhancement and scavenging on nucleation rates or apparent formation rates observed for new particles. In addition, the newly formed particles inside a street canyon might become toxic when vehicles released organics were involved in the nucleation process (Sgro et al., 2009; Gualtieri et al., 2014).

In this study, we present simultaneous aerosol measurements at a street site and a rooftop site of 20-m high in winter and spring times in Beijing (Fig. 1). PNSD at these two locations were measured by two identical Fast Mobility Particle Sizers with 1-second time resolution. We focused on analyzing the differences in apparent formation rates and particle yields of new particles between the street and rooftop sites, in order to evaluate the effects of street canyon on NPF. In addition, we also discussed the occurrence or absence of apparent growth in NPF events in terms of characteristics related to warm or cold ambient temperature, potential condensation vapors, etc.

## 2. Method

### 2.1 Sampling sites, periods and meteorological conditions

Two urban sites, approximately 500 m away from each other, were adopted for sampling in this study (Fig. 1). One site was 18 m away from the curb of a heavy traffic (Chengfu) Road at the north western area in Beijing, which was physically located inside a street canyon. It was referred to as street site afterwards. The daily average traffic volume on this road was $1.9 \times 10^3$ vehicles h$^{-1}$ with the maximum of $2.2\text{-}2.4 \times 10^3$ vehicles h$^{-1}$ in the morning and afternoon rush hours. A space-heating boiler with a stack in ~50 m height was approximately 200 m away from the street site at the northeast. The site on the rooftop of an academic building inside the campus of Peking University (~20 m above the ground level) was approximately 200 m away from the nearest Zhongguancun North Street. The site is referred to as the rooftop site in this study and was assumed to represent the urban background.

Two sampling campaigns were conducted in winter 2011 and spring 2012, respectively. The winter sampling in 2011 included two phases, i.e., 1) only one FMPS operated during 10-15 December at the street site; 2) two FMPS operated during 16-23 December at the street site and rooftop site, respectively. The weather is typically sunny and dry during the sampling campaign with surrounding air temperature from -9.5℃ to 12.8℃ (see Supplementary, Fig. S1). Relative humidity (RH) varied from 13% to 49% during NPF days. During 12-17 April 2012, the two identical particle sizers were deployed at the rooftop site for inter-comparison. The simultaneous measurements at two sites started from 18 to 27 April 2012. The ambient temperature ranged from 8.2℃ to 31.5℃ during the spring sampling period(Fig. S1). Sunny days occurred during 12-17 April and 25-27 April 2012 with ambient RH below 55%. Either rainy or cloudy days occurred in the remaining 7 days.

## 2.2 Sampling Instruments

Two identical Fast Mobility Particle Sizers (FMPS, TSI Model 3091) downstream of two dryers (TSI, 3062) were used in this study. FMPS were used to continuously measure PNSD ranging from 5.6 nm to 560 nm in 1-second time resolution, facilitating the investigation of rapid changes of nanoparticles due to formation and mixing from different sources (Yao et al., 2005, 2006b). Conductive tubes (TSI 3/8'', each was cut a length of 2.8 m) were used for sampling at the two sites. The two FMPS operated side-by-side during 12-17 April 2012 for inter-comparison. The correlation coefficients ($R^2$) of the measured number concentrations between the two sizers were greater than 0.95 for particles of 9.3 nm-107.5 nm, 0.81 for particles in the size bin of 8.06 nm. However, the correlation coefficients are fairly small ($< 0.3$) for the remaining two size bins (6.04 nm and 6.98 nm), thus are discarded in the analysis. The relative error between the two FMPS was less than 30% for 8.06 nm-107.5 nm particles (Table S1) and the difference for the measured number concentrations of particles $> 8$ nm was tested to be negligible by installing or removing the sampling line. However, FMPS was reported to underestimate the particle sizes compared to the SMPS and HR-ToF-AMS (Lee et al. 2013). Zimmerman et al. (2015) proposed that an independent measurement of Condensation Particle Counter (CPC) simultaneously with FMPS can be used to accurately correct the FMPS data. In this study, a CPC (TSI Model 3785) was operating simultaneously with a FMPS at the street site and the FMPS was thereby used as the reference to correct the number concentration measured by the other. The correction was detailed in the supplementary. Noted that the growth factors of $< 50$ nm particles were negligible under the RH levels on NPF days (Hämeri et al., 2000), and the effect of the dryers on the measured NPF events was not considered.

In addition to the two FMPS instruments and one CPC instrument measuring PNC, a few other instruments were deployed at the same time. For example, $SO_2$, $NO_x$, $NO$, $O_3$, $CO_2$, $CO$ were measured and recorded every minute at the rooftop site close to FMPS. Other available instruments, including aethalometer, DustTrak, Q-Trak photometer were used for filter sampling or semi-continuously measuring air pollutants at the street site. A meteorological station was located on the roof of one sixth-floor building about 50 m from the street site, measuring air temperature, RH, wind speed and direction. The solar radiation was measured by State Key Laboratory of Atmospheric Boundary Layer Physics and Atmospheric Chemistry (LAPC) (Hu et al., 2010, 2012).

## 2.3 Computational Methods

In this study, particles less than 20 nm are defined as the nucleation mode (Kulmala et al., 2004). The apparent formation rate of new particles larger than 8 nm (FR, $J_8$), taking consideration of the coagulation and growth losses, is calculated based on the Eq. (1) (Sihto et al., 2006):

$$J_8 = \frac{dN_{8-20}}{dt} + CoagS_{8-20} \cdot N_{8-560} + \frac{GR_{8-20}}{12} \cdot N_{8-20} + S_{losses} \tag{1}$$

where the coagulation loss for particles with diameter of 8-20 nm ($CoagS_{8-20} \cdot N_{8-560}$) is the sum of particle-particle inter- and hetero-coagulation rates calculated similarly to Yao et al. (2005). The growth loss is due to condensation growth ($GR_{8-20}$) out of the 8-20 nm size range during the calculation period. $S_{losses}$ includes additional losses and is assumed to be zero in this study. Thus, $J_8$ reflects a combination result of nucleation and subsequent initial growth of new particles.

The apparent growth rate (GR) of new particles is determined by the slope of the fitted geometric median diameter of new particles ($D_{pg}$, calculated as Whitby et al., 1978; Zhu et al., 2014) to the growth duration shown in Eq. (2):

$$GR = \frac{\Delta D_{pg}}{\Delta t} \tag{2}$$

The condensation sink is the loss rate of condensable vapor molecules onto the pre-existing particles, and calculated similarly to Dal Maso et al. (2005) and Kulmala et al. (2001, 2005):

$$CS = 2\pi D \int D_p \beta_M(D_p) n(D_p) dD_p = 2\pi D \sum_i \beta_M D_{pi} N_{pi} \tag{3}$$

where D is the diffusion coefficient, $\beta_M$ is the transitional regime correction factor, $D_{pi}$ is the particle diameter of size class i, and $N_{pi}$ is the particle number concentration in size class i.

Gas-phase sulfuric acid concentration is estimated based on global solar radiation (SR), $SO_2$ concentration and condensation sink (Petäjä et al., 2009):

$$[H_2SO_4] = k \cdot \frac{[SO_2] \cdot SR}{CS} \tag{4}$$

155 where $k$ is a constant value of $2.3 \times 10^{-9}$ $m^2$ $W^{-1}$ $s^{-1}$. In this study, the mixing ratio of $SO_2$ was measured only on the rooftop as

discussed in Section 2.2.

 The contribution of sulfuric acid vapor to the particle growth from $D_{p0}$ to $D_{p1}$ can be expressed in Eq. (5) based on

Kulmala et al. (2001):

$$R=([H_2SO_4]_{average}/C) \times 100\% \tag{5}$$

160 where $[H_2SO_4]_{average}$ is the mean concentration of $H_2SO_4$ during the entire growth period, and the concentration of

condensable vapor (C) for particle growth from $D_{p0}$ to $D_{p1}$ is calculated following Kulmala et al. (2001).

## 3. Results and discussion

### 3.1 Overview of NPF events during two campaigns

165 The NPF events occurred frequently in the sampling days, 7 out of 16 days in spring 2012 and 7 out of 14 days in winter

2011 (Fig. 2), consistent with previous studies showing that spring and winter were the seasons with highest frequency of

NPF events in Beijing (Wu et al., 2007; Wehner et al., 2008; Wang et al., 2017). In this study, regional NPF events represent

NPF events lasting over 1 hour, and short-lived NPF events represent the NPF lasting only for 10-20 minutes.

 During the spring campaign, the FRs at the rooftop site in regional NPF events ranged from 1.9 to 12.2 particle $cm^{-3}$ $s^{-1}$

170 on average of 8.0±3.5 particle $cm^{-3}$ $s^{-1}$ (Table S2). Two different growth patterns of new particles were observed, i.e., Class I

was characterized by a typical "banana shape" growth when the geometric median diameter of new particles ($D_{pg}$) was

increased from ~10 nm to 30-60 nm in 3-10 hours, which occurred during 12-14 and 16 April with the GR of 6.4±3.1 nm $h^{-1}$

(Figs. 2a and S2); Class II was characterized by the initial $D_{pg}$ of new particles at ~11 nm and no apparent growth being

observed during the next 6-8 hours until the signal of new particles dropped to negligible levels, which occurred on 15, 25

175 and 27 April (Fig. 2a). These two growth patterns have been frequently observed in Beijing (Wehner et al., 2004, 2008; Shi

et al., 2007; Wu et al., 2007), and no evident difference for the FRs between Class I and Class II. At the street site, Class II

NPF events were also observed on 25 and 27 April 2012 and the $D_{pg}$ maintained at ~11 nm for 6-8 hours without apparent

growth, consistent with the observed phenomenon on the rooftop (Fig. 3 a,b,e,f). The FRs in regional NPF events were 1.9

particle $cm^{-3}$ $s^{-1}$ at both sites on 25 April, and 10.2 particle $cm^{-3}$ $s^{-1}$ on the rooftop versus 8.1 particle $cm^{-3}$ $s^{-1}$ at the street site

on 27 April (Table S2). Four short-lived NPF events were observed only on 25 April 2012 and showed a larger FRs (13-49

particle $cm^{-3}$ $s^{-1}$) at the two sites. However, the FRs at the street site were decreased by 7%-50%. Note that Class II NPF

events were once argued as plume events in literature. The number concentrations of Aitken mode relative to nucleation

mode particles were negligible during the Class II NPF events, implying a negligible contribution from the plumes to the

observed total number concentrations. Our detailed analysis in the supplementary strongly indicated that Class II NPF events

should be regarded as regional NPF events instead of plume events.

On the 7 NPF days in the wintertime, the FRs at the street site were 7.0±2.9 particle $cm^{-3}$ $s^{-1}$ (Table S2). The values

were comparable to those in the springtime observed at the two sampling site. All these observed NPF events in the

wintertime were subject to Class II, i.e., the $D_{pg}$ of new particles remained at ~11 nm absent of apparent particle growth (Fig.

2b). Simultaneous NPF events were observed at two sites on 21, 22 and 23 December 2011 and the FRs were 0.9, 1.9 and

0.8 particle $cm^{-3}$ $s^{-1}$ on the rooftop, which were only 1/6-1/4 of those corresponding values at the street site, i.e., 4.0, 7.9 and

4.4 particle $cm^{-3}$ $s^{-1}$ (Table S2).

**3.2 Reduced NPF at the street site in the springtime**

On 25 and 27 April 2012, NPF events were simultaneously observed at the two sites and lasted for 6-8 hours (Fig.

3a,b,e,f). The long lasting time for NPF implied that the events occur in regional or semi-regional scale. We first analyzed

the stronger NPF on 27 April. The NPF event occurred around 09:37-09:40 (local standard time was used in this paper) and

was strongly associated with the increased speed of the northwest wind, i.e., from <1 m $s^{-1}$ at 08:00 to >6 m $s^{-1}$ after 09:45

(Fig. S3). The mixing ratio of $SO_2$ was increased from 1.5 ppb to 3 ppb during the initial half hour of the event and then

rapidly dropped down to <1 ppb for the remaining five hours (Fig. 3c). The nucleation mode PNC varied during the whole

event with three peaks observed at ~10:20, ~11:30 and ~13:45, suggesting the heterogeneity of NPF, which were likely

caused by the heterogeneity of precursors including sulfuric acid vapor, amine and/or other low volatile species in the

regional scale (Zhang et al., 2012; Kulmala et al., 2013; Ehn et al., 2014). The FR of 8.1 particle $cm^{-3}$ $s^{-1}$ at the street site was

slightly lower than that of 10.2 particle $cm^{-3}$ $s^{-1}$ on the rooftop, the condensation sinks were 1.2±0.37 ($\times10^{-2}$ $s^{-1}$) and

0.75±0.21 ($\times10^{-2}$ $s^{-1}$) for the street site and rooftop site, respectively (Table S2), with the higher condensation sink at the

street site partially responsible for the lower FR. There were obvious differences in the initial new particle burst time (defined as the time of nucleation mode particles reaching the maximum number concentration) between these two sites, i.e., 25 minutes at the street site and 36 minutes at the rooftop site, leading to a larger increase in nucleation mode PNC on the rooftop (Fig. 3c). This phenomenon was firstly observed in this study by adopting the high time resolution instrument. In order to estimate the net production of newly formed particles at the two sites, we defined $t_0$ as the time immediately before the apparent NPF was initially observed and $t_1$ as the time when the nucleation mode PNC reaches the maximum value. The net maximum increase in nucleation mode PNC (NMINP) was calculated as $N_{8-20nm}(t_1)-N_{8-20nm}(t_0)$. The NMINP at the street site was reduced by 30% relative to that on the rooftop (Fig. 3c,d), implying the reduced NPF at the street site.

On 25 April, NPF events were also associated with a high wind speed of the northwest wind (Fig. S4). Four short-lived NPF events together with one regional NPF event were observed at both sites (Fig. 3e-h). Each short-lived NPF event only lasted for 10-20 minutes (e.g., 10:07-10:26, 10:27-10:36, 10:38-11:02, 11:40-11:50 in Fig. 3f) concurrently with spikes of $SO_2$ at 1-2 ppb, but the calculated FRs were high, i.e., 14-49 particle $cm^{-3}$ $s^{-1}$ at the rooftop site and 13-38 particle $cm^{-3}$ $s^{-1}$ at the street site. The short-lived events strongly implied a key role of sulfuric acid vapor in NPF and the heterogeneity of NPF in both horizontal and vertical directions.

The regional NPF event on 25 April lasted for ~8 hours with varying nucleation mode PNC. On the rooftop, a longer new particle burst time and higher PNC was observed (Fig. 3g), similar to those observed on 27 April 2012. The FRs of 1.9 particle $cm^{-3}$ $s^{-1}$ were similar between the two sites. The calculated condensation sinks were $0.65\pm0.23(\times10^{-2}$ $s^{-1})$ and $0.16\pm0.02(\times10^{-2}$ $s^{-1})$ at the street site and rooftop site, respectively. The larger condensation sink at the street site was partially responsible for the reduced NPF during the four short-lived events, but did not affect the FR in the regional NPF event. However, the NMINP at the street site was reduced by 24% mainly due to the shorter initial new particle burst time (Fig. 3h).

It can be argued that the simple comparison, referred as Evidence 1 in the latter discussion, might be insufficient to confirm the reduced NPF at the street site because of the complicated micro-meteorology such as different scale turbulences therein. Micro-meteorology at street sites may cause accumulation or dilution of atmospheric particles. To solidify the

reduced NPF at the street site, we provided two types of additional evidences which were less affected by the micro-meteorology at the street site.

Evidences 2-3 were obtained by subtracting the PNC of different sized particles at the rooftop site from the corresponding one at the street site and the size-segregated difference of PNC between the two sites in April was thereby calculated (Fig. 4). The difference was largely negative for particles <14 nm during the NPF periods on 25 and 27 April (solid lines) against the positive difference of Aitken mode particles. In contrast, such a difference was slightly positive for particles <14 nm during the non-NPF days and during the morning rush hours on 25 and 27 April prior to the occurrence of NPF events (Fig. 4, dash lines), because of increasing contributions from on-road vehicles at the street site as well as the accumulation effect associated with micro-meteorology at the street site. The same accumulation effect should theoretically exist during the NPF periods on 25 and 27 April, but the observed result showed the reverse. We thus obtained Evidence 2, i.e., the negative difference of nucleation mode particles on NPF days against the positive difference of those on non-NPF days. Considered the positive difference of Aitken mode particles during the NPF periods on 25 and 27 April (solid lines), it can be inferred that micro-meteorology favored an increase of nucleation mode particle number concentration at the street site. However, the observed result was contradictory to the interference. We thus obtained Evidence 3, i.e., the negative difference of nucleation mode particles on NPF days against the positive difference of those Aitken mode particles.

Overall, the reduced NPF events were always observed at the street site and supported by three types of Evidences 1-3. Although the number of the cases was not large, the reduced NPF events at the street site were theoretically expected on basis of well recognized factors in literature, e.g., 1) a larger condensation sink associated with more pre-existing atmospheric particles from primary emissions; 2) tall buildings along both the sides of urban streets can provide additional surface areas to scavenge gases and atmospheric particles (Yao et al., 2011); 3) vehicle-emitted NO reacting with $RO_2$ and suppressing NPF (Wildt et al., 2014).

### 3.3 Enhanced NPF at the street site in the wintertime

On 21-23 December 2011, NPF events were simultaneously observed at the rooftop site and street site with the FRs of 0.8-1.9 particle $cm^{-3}$ $s^{-1}$ and 4.0-7.9 particle $cm^{-3}$ $s^{-1}$, respectively (Figs. 5 and S5). The different FRs implied that NPF was

always enhanced greatly at the street site, which was referred to as Evidence 1 for the enhanced NPF in the latter discussion.

Larger condensation sinks were, however, calculated at the street site ($1.3 \times 10^{-2}$ s$^{-1}$ at the street site and $0.45\text{-}0.98 \times 10^{-2}$ s$^{-1}$ at rooftop site, Table S2), implying the influence of the larger condensation sink on NPF apparently to be overwhelmed by unknown factors.

The strongest NPF event was observed on 22 December. The NPF event initially observed at 09:40-09:45 at both sites (Fig. 5a-d) was also apparently correlated with increasing speed of the northwest wind (Fig. S6). The initial new particle

burst time periods were different at two sites. For example, nucleation mode PNC at the rooftop site were gradually increased from $0.2 \times 10^4$ particle cm$^{-3}$ at that time to the maximum value of $1.4 \times 10^4$ particle cm$^{-3}$ during the initial 2 hours (Fig. 5c). At the street site, vehicle emissions frequently influenced the sampling site, leading to numerous spikes in PNC and large uncertainty in calculating the FR. We thus used the 25% minimum coefficient of variation (CV) of PNC as an indicator to eliminate the vehicle spikes (see supplementary for the calculation method). The approached results showed that

the nucleation mode PNC was rapidly increased within the initial 26 minutes and then decreased. The FR of 7.9 particle cm$^{-3}$ s$^{-1}$ at the street site was three times larger than that of 1.9 particle cm$^{-3}$ s$^{-1}$ on the rooftop. The larger FR at the street site was mainly associated with a shorter time for the new particle burst. The NMINP at the street site during the whole NPF event was about 50% higher than that on the rooftop (Fig. 5d). When the nulceation mode PNC at the street site reached the maximum, the corresponding PNC on the rooftop was only one-third of its own maximum value. The different increasing

patterns of nucleation mode particles at two sites strongly indicated that they were subject to different NPF mechanisms.

We also directly deducted the contribution of vehicle spikes using the second approach described in the supplementary. The newly obtained PNC at the street site were shown in Fig. 6a. The results obtained from the new approach showed 1) the initial new particle burst time was 30 minutes, 2) the calculated FR was 7.0 particle cm$^{-3}$ s$^{-1}$, and 3) the NMINP was 61% higher than that on the rooftop. The results reasonably agreed with the previous results using the 25% minimum CV (Fig. 6b).

The two approaches strongly implied NPF being greatly enhanced at street site.

Black carbon (BC) or NOx were also proposed to deduct the contribution of vehicle spikes (Fruin et al., 2004; Wang et al., 2012). The measured concentration of BC was thereby tried for deduction and much poor correlation was obtained due to one-minute time resolution can't allow successfully capturing vehicle spikes varying in a few seconds (see supplementary).

Under such poor correlation, the regression equation between PNC and BC was invalid to accurately deduct the contribution from vehicle spikes. Theoretically, the correlation could be improved substantially with increasing distance of the sampling site from traffic roads because of less dynamic changes in PNC and PNSD (Zhu et al., 2002 a,b, 2006). Although the two approaches used in this study may still suffer from uncertainty to some extent, they should be much better than those reported in literature.

On 21 December 2011, the NPF event at the street site was also characterized by a shorter initial new particle burst time and a larger FR (Fig. 5e-h). Using the 25% minimum CV approach, the NMINP at the street site was 24% higher than that on the rooftop (Fig. 5h). When the nucleation mode PNC at the street site reached the maximum, the corresponding PNC on the rooftop was only one-fifth of its own maximum value. The NMINP at the street site was 46% large than that at the rooftop site on 23 December (Fig. S5). The second approach was invalid in the two days because of the weak NPF events.

We further seek additional evidences to support the enhanced NPF at the street site while the evidences should be less affected by complicated micro-meteorology. We also calculated the difference of PNC between the two sites in December 2011 using the number concentrations at the street site subtracting the corresponding concentration at the rooftop site (Fig. 7). Theoretically, larger condensation sinks should cause stronger scavenging effects at the street site during the NPF periods in the wintertime than in the springtime. However, the difference of nucleation mode particles in the wintertime was positive. We then obtained Evidences 2 to solidify the enhanced NPF at the street site, i.e., the positive difference of nucleation mode particles in the wintertime against the negative difference in the springtime on NPF days. In Fig. 7, on the strongest NPF day (22 December), the positive difference for <20 nm particles during the NPF periods was evidently larger than the difference obtained during approximately two hours prior to the NPF. The reverse was true for 20-80 nm particles. The former larger difference was unlikely due to the low-ambient-temperature-favored stronger formation of primary vehicular particles during the initial 1-2 seconds dilution (Burgard et al., 2006; Bishop et al., 2010) and the poor dispersion condition because of the higher ambient temperature and larger wind speed during the NPF period. In fact, the differences for <20 nm particles during the NPF period on 22 December were larger than the differences averaging over non-NPF days (17-19 December) in December and the average value observed on 20 December alone, when the most frequent spikes of vehicular particles occurred among all non-NPF days in December, at the corresponding particle size ranges. We thereby obtained Evidence 3,

i.e., the larger positive difference of nucleation mode particles on NPF days against those on non-NPF days in the wintertime

(Fig. 7). All three evidences indicated the NPF being enhanced at the street site on 22 December. When the vehicular particle

spikes during the NPF on 22 December were eliminated using the 25% minimum CV approach, the newly obtained

difference also supported the NPF being enhanced at the street site. The differences for <20 nm particles on 21 and 23

December during the NPF were also positive, but the NPF events on the two days were weak.

**3.4 Analysis of NPF being enhanced at the street site**

Varying FRs, sulfuric acid concentrations and condensation sinks during the two measurement campaigns were shown

in Fig. 8. The calculated concentrations of sulfuric acid were between $2 \times 10^6$ and $2 \times 10^7$ cm$^{-3}$ during NPF periods, which

was comparable with previous observations in Beijing (Yue et al., 2010; Zheng et al., 2011; Wang et al., 2017). Although the

estimated concentrations of sulfuric acid were the highest on the rooftop in December when the calculated condensation

sinks were comparable to those in April, the corresponding FRs were the smallest. It is well known that nucleation of

sulfuric acid enhanced by organics dominantly determine FRs in the urban atmosphere (Zhang et al., 2004, 2009, 2010;

Wang et al., 2010). The smallest FRs at the rooftop site in December can be due to the lack of sufficient low-volatility

organics or amines, which were possibly associated with low biogenic emissions at the low ambient temperature. However,

the number of datasets in this study was too small to gain a reasonable equation to link the calculated concentrations of

sulfuric acid with FRs.

Relative to the rooftop site, the largely increased FRs at the street site in December were unlikely due to the increased

concentrations of sulfuric acid. Our arguments were presented as below: 1) using the measurements in April as a reference,

the concentrations of sulfuric acid were probably lower at the street site than at the rooftop site on basis of lower FRs and

larger condensation sinks at the street site, the scavenging effect should also occur in December and lower the concentrations

of sulfuric acid at the street site; 2) in the December, the mixing ratios of $SO_2$ at 1-2 hours immediately before NPF were 3-5

ppb at the rooftop site. The $SO_2$ was mainly from domestic heating and the traffic-derived $SO_2$ at the street site was roughly

estimated to be <1.3 ppb according to the results in our previous studies (Meng et al., 2015 a,b). The concentrations of

sulfuric acid at the street site may be very likely close to or even lower than those at the rooftop site since the stronger scavenging effect probably canceled out the traffic-derived contribution to sulfuric acid.

330  Combining possibly low emissions of biogenic precursors in December, stronger condensational sinks and larger FRs at the street site in December, we inferred that NPF enhanced at the street site in December were very likely due to the nucleation of $H_2SO_4$ enhanced by additional chemicals such as organics and amines from vehicle emissions. There is a need for further simultaneous measurements of vapor precursors such as HOMs and organic acids (Zhao et al., 2009), and chemical compositions of ~10 nm particles at the rooftop site and street site simultaneously.

### 3.5 Limiting factors for the growth of new particles

  The two particle growth patterns of NPF events were further discussed. The new particles in Class I and Class II may exert severe health problems to human beings considering its large PNC. At the meantime, the new particles in Class I could potentially have impacts on climate through radiation feedback. Theoretically, the GR of new particles is largely dependent

340 on the amount of condensable vapors conquering over the thermodynamic force plus Kelvin Effect. The amounts of condensable vapors were determined by the emission rates of vapor precursors, photochemical reactions of the precursors and scavenging rates through the gas-particle condensation and deposition.

  In Class I which was observed only in April, sulfuric acid condensation was estimated to account for only 2.3%-18% of the new particle growth, which was consistent with previous studies in Finland and Mexico (Smith, et al., 2008). All NPF

345 events in December were subject to Class II when the estimated concentrations of sulfuric acid were larger than that in April. It is reasonable to say, therefore, sulfuric acid was not the crucial species in determining two particle growth patterns.

  As reported in the literature, the oxidation products of biogenic organic gases likely overwhelmingly determined the condensation growth of newly formed particles between 10-50 nm (Riipinen et al., 2011; Pierce et al., 2012; Schobesberger et al., 2013; Ehn et al., 2014; Liu et al., 2014; Ortega et al., 2015; Tröstl et al., 2016). In this study, the north or northwest

350 wind dominated during the NPF periods, and the north and northwest directions of the sampling site subject to mountain areas have a high percentage of land-covered forests. Extensive biogenic VOC is theoretically expected in spring and may act as important precursors in NPF. No apparent growth of newly formed particles at ~11 nm in Class II in December was

possibly related to low biogenic emissions of organic gases in cold seasons. The hypothesis was not applicable for Class II in April. NPF events in April occurred in sunny, windy and warm days, regardless of Class I and Class II. Moreover, there were no significant difference for condensation sinks between in Class I and Class II, suggesting that the scavenging effect may not be the key factor in determining the presence or absence of the apparent new particle growth. To further understand the mechanism modulating the differences between Class I and Class II, photochemical reactions are discussed as follows. As shown in Fig. 2, the mixing ratio of $(NO_2+O_3)$ in Class I was generally larger than that in Class II, indicating that Class II in April may be related to weaker photochemical reactions, albeit uncertainty exists (i.e., the mixing ratio of $(NO_2+O_3)$ was 40-50 ppb on 12 April concomitant with particle growth, comparable to that in Class II).

When four NPF events in Class I were examined, the observed growth rates of > 10 nm new particles were very low (0-0.6 nm h$^{-1}$) in the initial 20-70 minutes and then rapidly increased to 2.2-9.3 nm h$^{-1}$ in the next 2-7 hours (Fig. S2). The smooth variations of $NO_2+O_3$ cannot explain the sudden and rapid growth of new particles after the initial 20-70 minutes. Alternatively, the sudden shift of the gas-particle system equilibrium seemed to be the reason. When the product of gases started to be larger than the thermodynamic equilibrium constant plus the Kelvin Effect term, the reaction should proceed to the solid state, i.e., the gases start to partition on the particle phase, leading to the sudden growth of new particles. This also implies that semi-volatile species may play a role in the particle growth.

As mentioned earlier, there was no evident difference for the FRs between Class I and Class II in April. It can be inferred that the organics driving the apparent growth of > 10 nm new particles were probably different from the organics involved in nucleation (Kulmala and Kerminen, 2008; Zhang et al., 2010). That also applies to the case without apparent growth of new particles with the increased FRs at the street site in December.

Again, it is difficult to detect the organic and inorganic species in 10-50 nm particles (Smith, et al., 2008; Yue et al., 2010; Bzdek et al., 2012). The same can be said to confirm the actual organics driving the growth of >10 nm new particles. Oxidized anthropogenic VOCs could theoretically participate in the growth of newly formed particles while the study is limited (Zhang et al., 2009; Hoyle et al., 2011), but the role of oxidized anthropogenic VOCs needs further study.

**3.6 Relationship between FR and new particle yield**

Sulfuric acid vapor has been identified as a key component for nucleation in urban atmospheres (Weber et al., 1996; Kulmala, 2003; Berndt et al., 2005; Fiedler et al., 2005). Supposed that sulfuric acid vapor are completely nucleated,

followed by the nucleated particles growing to the detectable size, the yields of newly formed particles are determined mainly by the supply of sulfuric acid vapor and are less affected by the formation rate. However, it will take a long time to completely convert sulfuric acid vapor to new particles in a slow formation rate. In the atmosphere, sulfuric acid vapor, newly formed clusters and particles can be largely scavenged by preexisting particles. The increased formation rate favors coagulation growth of <8 nm new particles, shortening the time of new particles growing to be over 8 nm (detectable in this

study) and therefore increase the conversion efficiency of sulfuric acid vapor to >8 nm particles. How the increased formation rates affect the production of > 8 nm particles were examined as below:

1) The FR was increased by ~3 times at the street site relative to the rooftop site on 22 December 2011. This resulted in additional increase in nucleation mode PNC by $4.9 \times 10^3$ particle $cm^{-3}$, equal to 50% of the NMINP at the rooftop site; 2) The FRs at the street site were increased by ~4-5 times relative to the rooftop site on 21 and 23 December 2011. This led to

390 additional increase of new particles by $1.1 \times 10^3$ particle $cm^{-3}$ (equal to 24% of the NMINP at the rooftop site) and $2.1 \times 10^3$ particle $cm^{-3}$ (equal to 46% of the NMINP at the rooftop site), respectively. The largely increased FRs apparently yielded a small influence on the NMINP.

To further exploring the relationship between FR and new particle yield, we summarized 139 cases of NPF events (only four short-lived events in this study being included, all the others were subject to regional NPF events) from our published

and unpublished database measured in Beijing, Qingdao and marginal seas of China, etc. (Fig. 9a, Liu et al., 2014; Zhu et al., 2014; Man et al., 2015; Guo et al., 2016). Considered 1) formation rate of new particles, e.g., $J=k_{NucOrg}[H_2SO_4]^m[NucOrg]^n$ (Zhang et al., 2012) where $k_{NucOrg}$ is a constant, NucOrg represents organics involved in nucleation, m and n are two integers, 2) the subsequent particle growth, and 3) $H_2SO_4$ vapor to be necessary for nucleation in ambient air except at sea beach, two scenarios are analyzed. Scenario 1: $H_2SO_4$ vapor is relatively sufficient against NucOrg and $J_8$ is thereby mainly determined

by availableness of NucOrg vapor. A good correlation is theoretically expected for $J_8$ and NMINP. Scenario 2: NucOrg vapor is relatively sufficient against $H_2SO_4$ vapor and $J_8$ is thereby mainly determined by availableness of $H_2SO_4$ vapor. $J_8$ could be high, but the total yield of new particles could be low because of a rapid consumption of $H_2SO_4$ vapor. A poor or no

correlation is theoretically expected for $J_8$ and NMINP. For the cases summarized in Fig. 9a, the FRs and the NMINP had a moderately good correlation under FRs $\leq 8$ particle cm$^{-3}$ s$^{-1}$, with r=0.76 and p<0.01. For the FRs >8 particle cm$^{-3}$ s$^{-1}$, the two variables had no correlation. When the FR was increased from 1 to 8 particle cm$^{-3}$ s$^{-1}$, the nucleation mode PNC was increased from $0.4 \times 10^4$ to $3.3 \times 10^4$ particle cm$^{-3}$ according to the regression equation. The statistical response of the NMINP to the increased FR was stronger than the results observed at the street site in December. This allows speculating that the NMINP in most of the atmospheric observations was possibly determined by the concentration of sulfuric acid vapor slightly more than additional organics. To support the hypothesis, we plotted NMINP against the mixing ratios of $SO_2$ measured during part of the events (Fig. 9b). The correlations were positive when the limited datasets were artificially seperated into two categories, i.e., r=0.81 and p<0.01 for Categroy 1, r=0.65 and p=0.16 for Category 2. This of course needs both vapors' data (e.g., HOMs) and chemical composition data in nucleation mode particles to confirm in future. For the FRs >8 particle cm$^{-3}$ s$^{-1}$, the concentration of additional organics vapor appeared to overwhelmingly determine the FRs, and the NMINP appeared to be determined mainly by the concentration of sulfuric acid vapor instead of additional organics.

## 4. Conclusions

The simultaneous aerosol measurements at a street site and a rooftop site were conducted using two FMPS in 1-second time resolution during two seasons in Beijing. At the street site, the reduced NPF events always occurred in the springtime while the enhanced NPF events always occurred in the wintertime. In the springtime, the reduced NPF was characterized by: 1) the lower PNC of nucleation mode particles at the street site mainly because of a shorter initial burst time, 2) the negative difference of nucleation mode particles against the positive difference of Aitken mode particles on NPF days, 3) the negative difference of nucleation mode particles on NPF days against the positive difference of that on non-NPF days. We inferred that the reduced NPF at the street site was likely attributed to the scavenging effect where pollutants emitted from on-road vehicles were accumulated. In contrast, the enhanced NPF at the street site relative to the rooftop site in the wintertime was observed and supported by: 1) the significantly larger PNC of nucleation mode particles (24%-50% increasing) at the street site and a larger FR (3-5 times higher) mainly because of a shorter initial burst time, 2) the positive difference of nucleation mode particles in the wintertime against the negative difference of nucleation mode particles in the springtime on NPF days,

3) the larger positive difference of nucleation mode particles on NPF days against that on non-NPF days in the wintertime. Through in-depth analysis, the largely increased FRs were argued due to the nucleation of $H_2SO_4$ enhanced by additional

chemicals such as organics and amines from vehicle emissions, although further validations including direct measurements of amines and HOMs in the newly formed particles are still needed.

Two growth patterns of new particles were observed and occurred seasonally during our measurements, i.e., Class I showed a clear "banana shape" growth of new particles and occurred only in the springtime (4 days out of 7 NPF days), while Class II showed no apparent growth of new particles at ~11 nm and occurred in the springtime (3 days out of 7 NPF

435    days) and always in the wintertime. Sulfuric acid can explain only 2.3%-18% of the new particle growth in Class I, and therefore are unlikely the crucial species determining two particle growth patterns. Through a comprehensive analysis and combining widely recognized contribution of oxidized biogenic organics in growing particles in literature, we suggest that semi-volatile species oxidized from biogenic organics in presence of strong photochemical reactions play an important role in the growth of new particles > 10 nm.

**Acknowledgments**

We would like to thank the support from grant National Program on Key Basic Research Project (973 Program, 2014CB953703) and National Natural Science Foundation of China (NFSC, 41576118). We thank the instrument from Prof. Tong Zhu and valuable comments from Prof. Ming Fang.

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

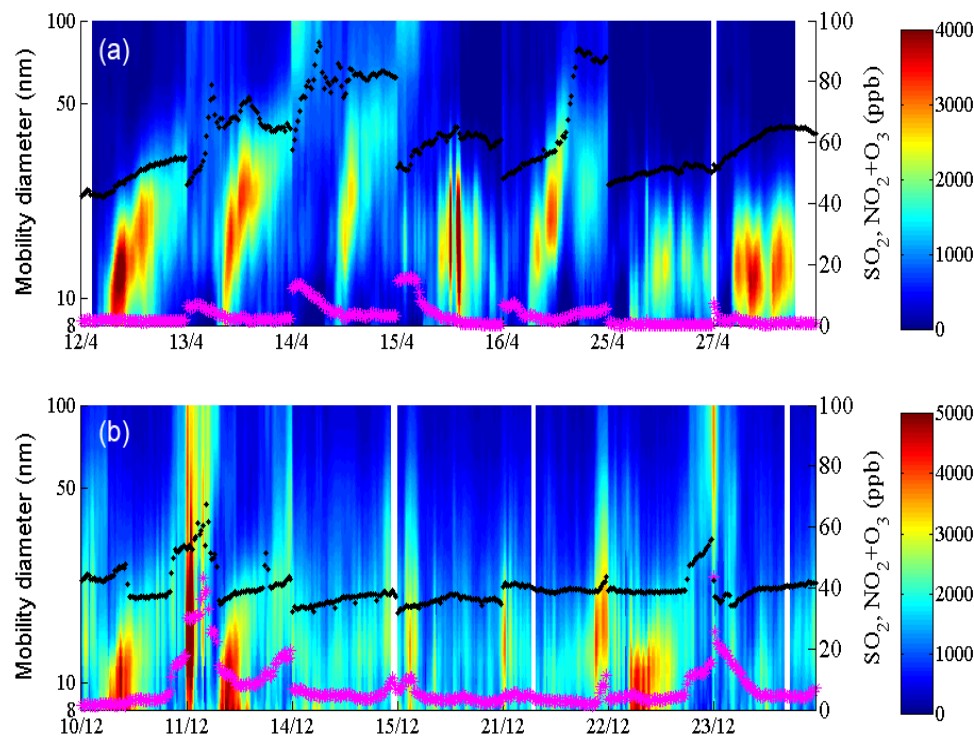

**Fig. 2** Contour plots of particle number concentration (# cm$^{-3}$) and time series of NO$_2$+O$_3$ and SO$_2$ in mixing ratio in two seasons (a: seven NPF events during 12-27 April 2012, b: seven NPF events during 10-23 Decemebr 2011; there was only data from 8:00 to 18:00 in each day to be shown, full black diamonds and peak magenta stars represent the mixing ratios of NO$_2$+O$_3$ and SO$_2$, repectively, while the values correspond to the right Y Axis).

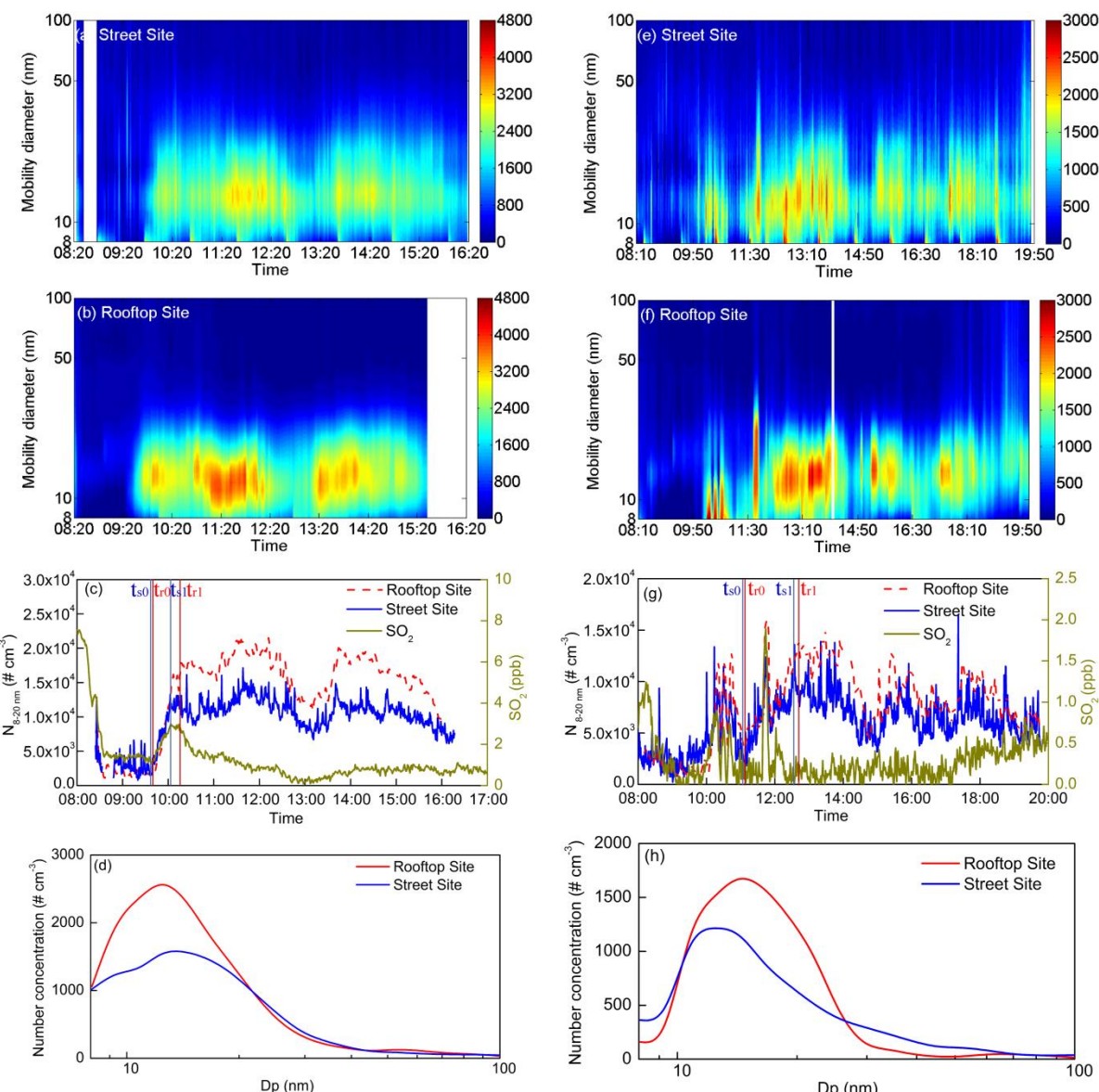

**Fig. 3** Contour plots, time series of number concentrations and size distributions of atmospheric particles at two sampling sites on 27 April 2012 (left column) and 25 April 2012 (right column) (a, b, e, f: Contour plots of particle number concentration (# cm$^{-3}$); c, g: time series of nucleation mode PNC ($N_{8-20nm}$) and $SO_2$ mixing ratios at two sampling sites; d, h: size distributions of $N_{(ts1)}-N_{(ts0)}$ at the street site and $N_{(tr1)}-N_{(tr0)}$ on the rooftop, $N_{(ts1)}-N_{(ts0) \text{ and }} N_{(tr1)}-N_{(tr0)}$ represent the number concentration at each size bin at $t_1$ minus that at $t_0$ at the street site and at the rooftop site, respectively ).

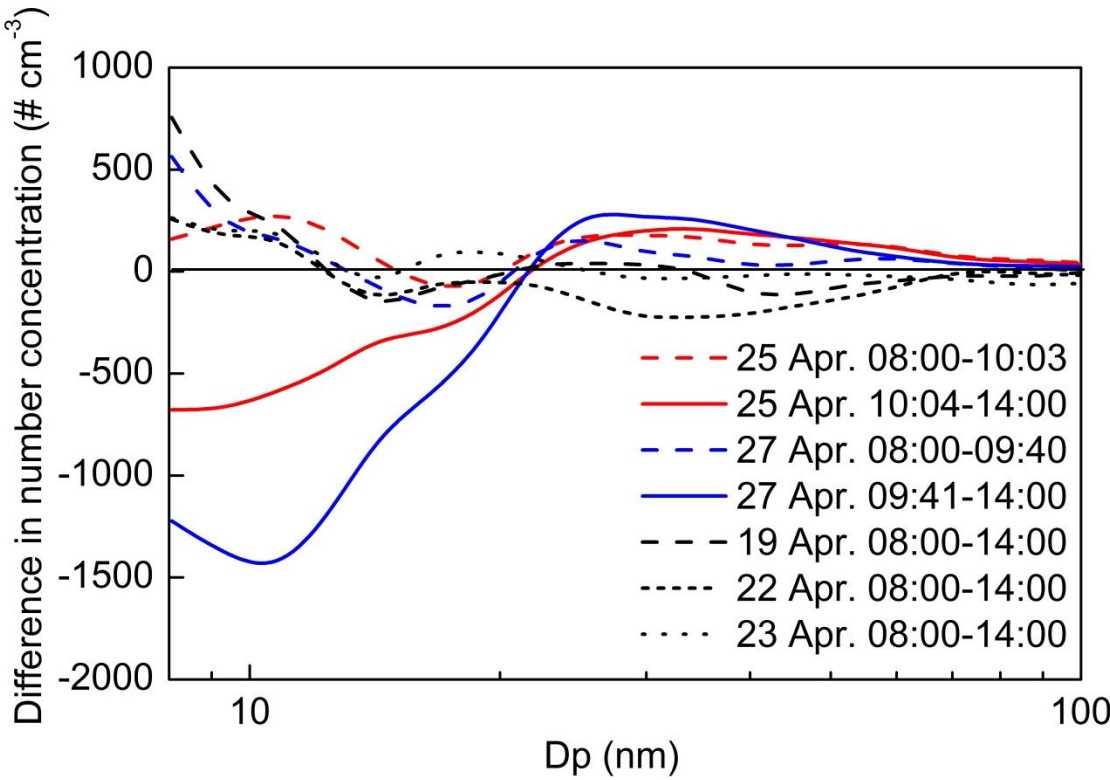

**Fig. 4** The size-dependent difference in number concentrations (calculated by the PNC of different sized particles at the street site subtracting the corresponding one at the rooftop site) during various periods in April 2012 (solid and dash lines represent the results during NPF and non-NPF periods, respectively).

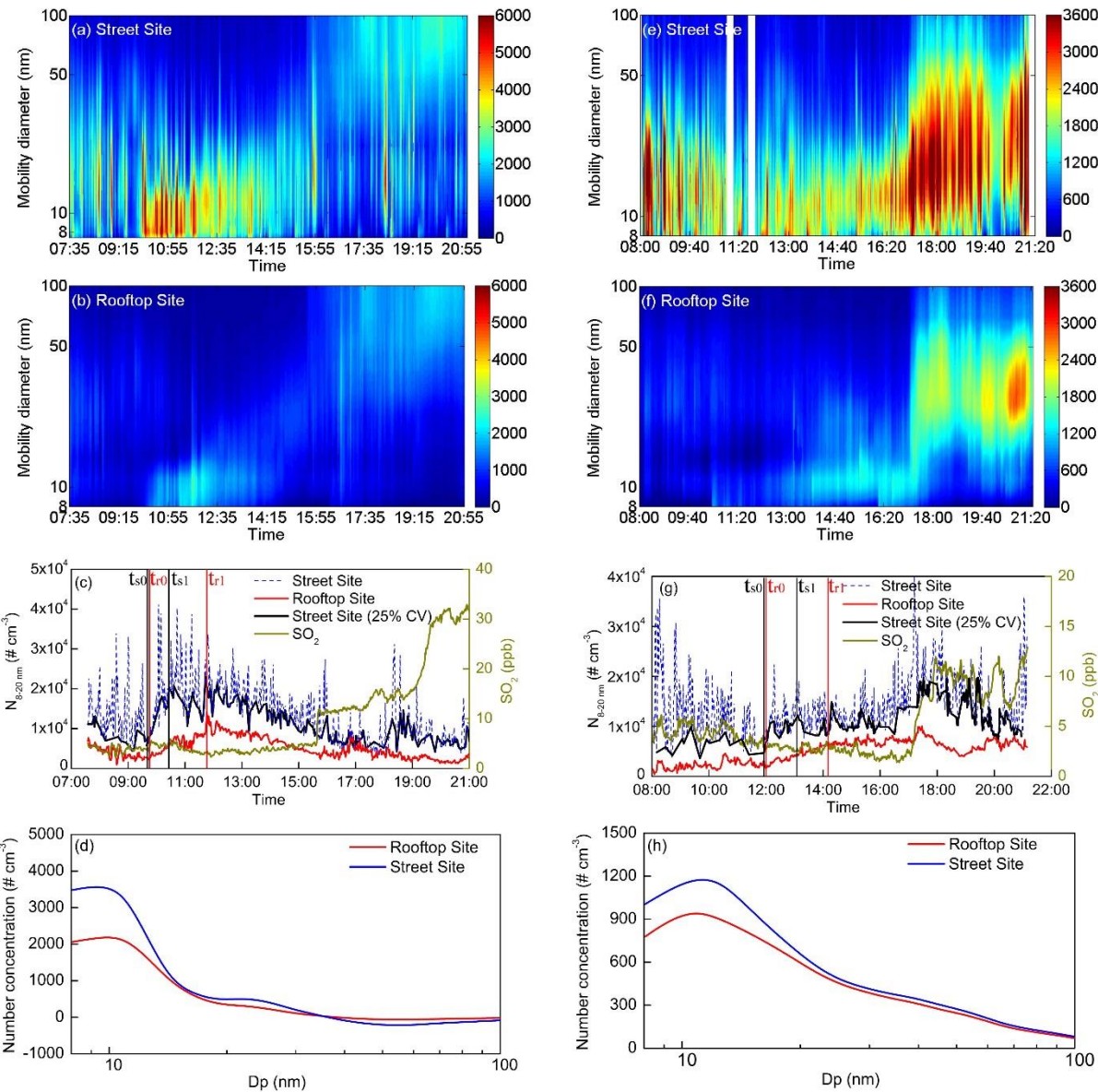

**Fig. 5** Contour plots, time series of number concentrations and size distributions of atmospheric particles at two sampling sites on 22 Decemebr 2011 (left column) and 21 Decemebr 2011 (right column) (a, b, e, f: Contour plots of particle number concentration (# cm$^{-3}$); c, g: time series of nucleation mode PNC (N$_{8-20nm}$) and SO$_2$ mixing ratios at two sampling sites; d, h: size distributions of N$_{(ts1)}$-N$_{(ts0)}$ at the street site and N$_{(tr1)}$-N$_{(tr0)}$ on the rooftop).

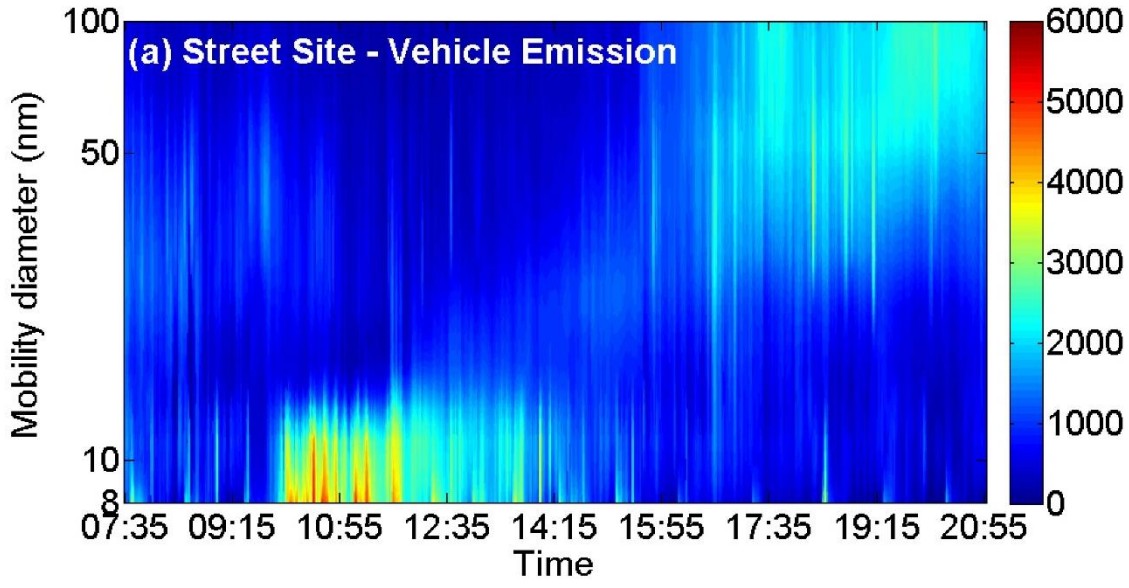

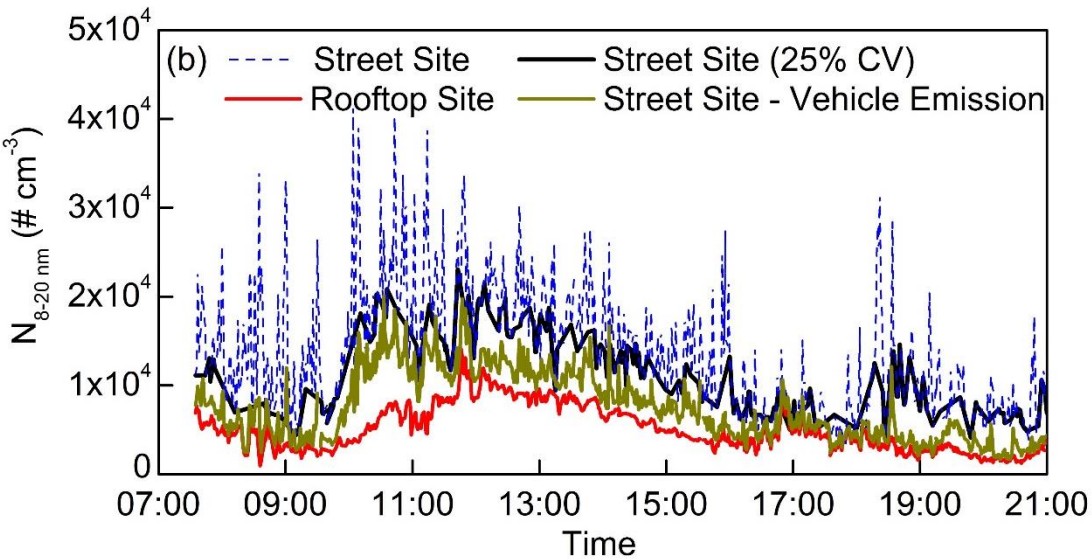

**Fig. 6** Contour plot and time series of particle number concentration with two approached used to deduct freshly emitted traffic particles

on 22 December 2011 (a: Contour plot of particle number concentration calculated from the second approach described in the supplementary; b: time series of number concentration at the street and rooftop site and those calculated from the two approachs defined in the context).

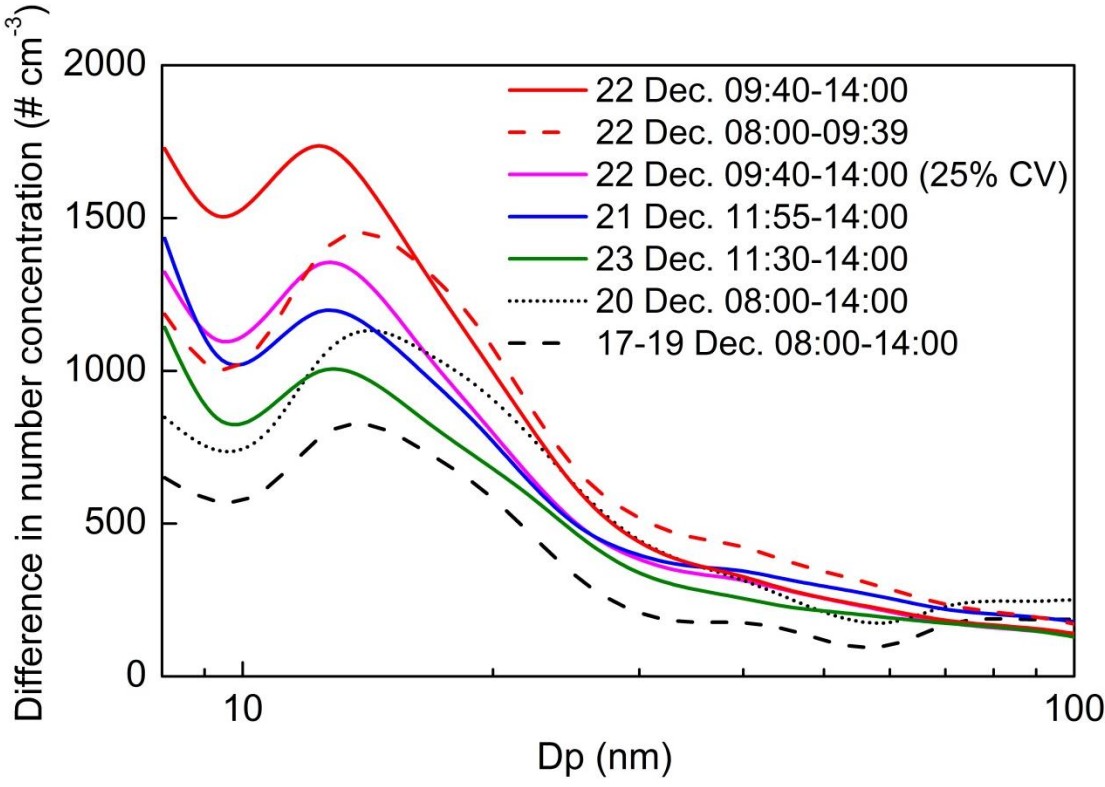

**Fig. 7** The size-dependent difference of particle number concentrations between the two sites in December, 2011 (solid and dash lines represent the results during NPF and non-NPF periods, respectively).

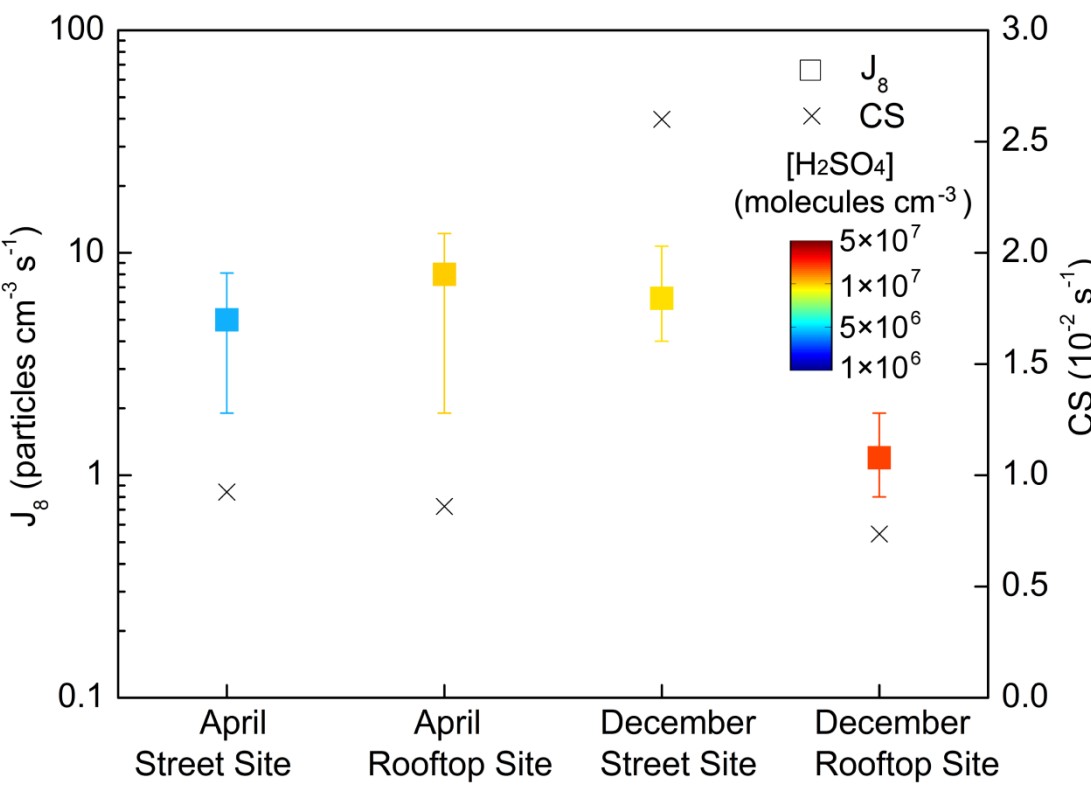

**Fig. 8** Mean and range of FR (box symbol), condensation sinks (cross symbol) and estimted sulfuric acid concentrations (mean and standard deviation) in color bar on NPF days at two sites in April 2012 and December 2011 (The sulfuric acid at the street site was calculated by assuming the mixing ratio of $SO_2$ therein same as that on the rooftop and the uncertainty on the estimation was analzyed in Supplementary).

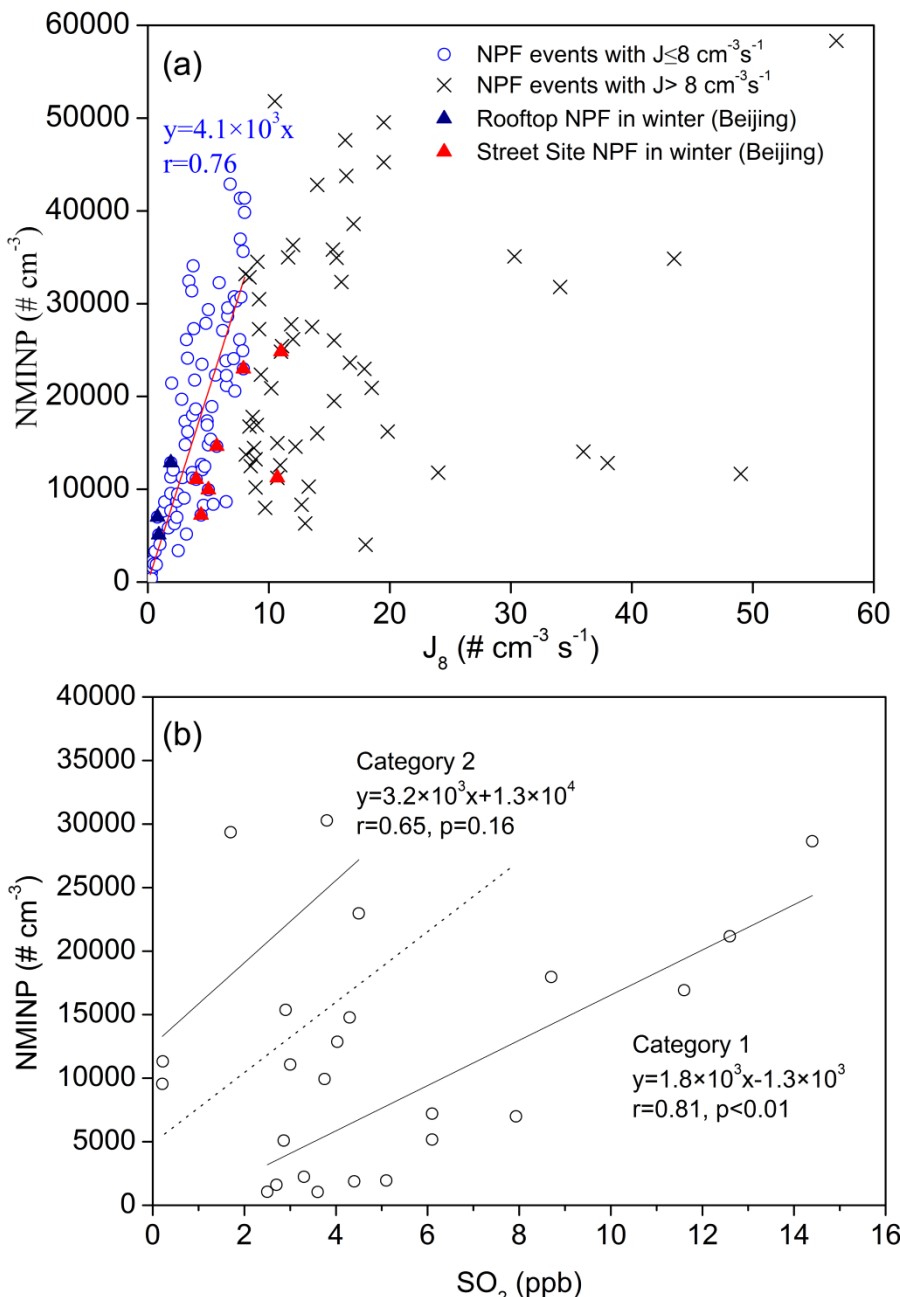

**Fig. 9** Relationship between NMINP and FR ($J_8$) in 139 cases of NPF events and betweem NMINP and $SO_2$ in some cases (a: NMINP vs. FR ($J_8$); b: NMINP vs. $SO_2$; the dash line in fig. b is arbitrarily drawn; the averaged mixing ratio of $SO_2$ during each NPF event was used and missing $SO_2$ data are either due to malfunction of instruments or further QA/QC to be required) .