# Peer review of "Simultaneous measurements of new particle formation in 1-second time resolution at a street site and a rooftop site"

_Atmospheric Chemistry and Physics, 2016_

## Referee Comment (RC1) · Anonymous Referee #1 · 11 Jan 2017

This manuscript investigates new particle formation (NPF) observed simultaneously at two sites in a polluted urban environment. The analysis is based high-time resolution measurements, which increases the originality of the results. The background for this study (section 1) as well as the used methods (section 2) are very well written. Contrary to this, there are serious problems in how many of the results, have been interpreted. As a result, a large part of section 3 needs substantial revisions, and most of the sections 3.4-3.6 need to be entirely re-written. My detailed comments in this regard are given below.

Major comments:

The authors provide two very general statements based on their results: 1) reduced

[Figure]

NPF at street site compared to rooftop during spring, and 2) enhanced NPF at street site compared with rooftop in winter. These finding are supported by only 1-2 cases (days) of observations, which is way too little to make this kind of a general conclusion.

The used proxy for gaseous sulfuric acid (SA) concentration has two problems: 1) it has been developed and evaluated for moderately-polluted sites only, so its applicability in highly-polluted sites like this one may be questionable, 2) SO2 is measured at rooftop site only, so it is unclear how well this represents SO2 in the street site. Also, the ratio in the SO2 concentration between the street site and rooftop is likely to be different between spring and winter, and there is no means to estimate this difference. As a result, the authors need to be very careful when making any interpretations that rely on estimated SA concentrations.

Class II NPF events have very low particle growth rates above 10 nm. All theoretical arguments indicate that >10 nm particles grow faster than smaller particles, and practically all observations on size-resolved particle growth rates support this view. This lead to a serious question: what is the origin of these particles? More specifically, if there are little condensable vapours to growth >10 nm, there should be even less vapors to grow smaller particles. One possible explanation for this is that particle of Class II originate from very local NPF, in which high local vapor concentrations initial nucleation and make the formed particles to grow very rapidly to a few nm, even to 10-20 nm. This rapid growth is then stopped due to atmospheric dilution of emitted vapors. This kind of process has been reported to occur in some coastal areas (Mace Head), in car exhaust to ambient air, and also close to other localised combustion sources. If Case II event are caused by very localized sources, it is questionable to compare NPF between the street site and rooftop in such cases.

The authors use condensation sink (CS) in interpreting their results. This problematic. The particles are formed below 2 nm size (J<2), but the authors calculate the formation rate of 8 nm particles (J8). The value of J8 depends on 3 quantities. J<2, CS and the growth rate of particles below 8 nm. Since neither J<2 nor the sub-8 nm growth rate

are known, it is impossible to infer how CS might affect J8 in the observed cases.

The authors assumed that only biogenic organics could influence NPF and subsequent growth. Why? There certainly large anthropogenic emissions of organic vapours in this kind of environment, and the oxidation of such vapors is very likely to produce low-volatile compounds that could affect nanoparticle formation and growth.

Considering the points highlighted above, many of the interpretations made in sections 3.4-3.6 are not justified. The most problematic of these is section 3.6 which is highly speculative.

Minor comments

I would recommend using terms other than short-term and long-term NPF events. In atmospheric time series, long-term usually means something that last for years or at least for months.

line 208: should be written: . . .only lasted for few minutes

lines 254 and 270: did't detail is a strange expression. Please modify

line 320: what is meant by ..reaction should proceed to solid state

---

## Referee Comment (RC2) · Anonymous Referee #2 · 21 Jan 2017

General Comments: In this work, Zhu et al. presented new particle formation (NPF) events observed at both a street site and a rooftop site using two TSI 3091 FMPS during both spring- and winter-time. The authors reported two major findings: 1) NPF was enhanced at the street site during wintertime due to seasonal street canyon effects; 2) Photochemically oxidized biogenic organics might contribute significantly to the growth of >10 nm particles. Overall, the manuscript is fairly well written and the subject of the research is certainly within the scope of Atmospheric Chemistry and Physics (ACP). The unique feature of this work is the high time-resolution (1Hz) observations of particle number size distribution (PNSD). It does provide an advantage in NPF research. This is an interesting study but the reviewer feels that there are several issues needed to be

further addressed before the manuscript can be considered for publication in ACP.

Specific Comments: 1. L27: I do not have too much confidence that the authors can draw a conclusion of "seasonal effects" from several days of NPF events. It is more like a case study. After all, there were no summer and autumn observations.

2. L30: "The oxidation of biogenic organics...apparent growth." There is no clear evidence of strong biogenic VOC emission at the sites. The authors may want to look into the anthropogenic VOC emissions, such as traffic-related VOC emissions.

3. NPF event consists of both nucleation and the ensuing particle growth. It would be more reasonable to replace "NPF rate" with "nucleation rate".

4. L235-240: The authors used "25% minimum coefficient of variation (CV) of particle number concentration" as an indicator to eliminate vehicle emission spikes from the NPF dataset. If possible, I suggest the authors also use NOx as an indicator of vehicle-emission plumes. For example, did the particle number concentration spikes show any correlation with NOx time series?

5. As described by the authors that the street site was "18 m away from the curb of a heavy traffic (Chengfu) Road at the northwestern area in Beijing", it would be necessary not only to remove spikes of vehicle emissions but also to take into account the small particles transported from further down the road, which may be not as distinct as the spikes caused by passing by vehicles but would certainly raise the background level of ~10 nm particles.

6. L280-290: Ambient temperature changed substantially from spring to winter as indicated in the experimental section. The authors may also want to consider the role of weather in affecting the nucleation rate.

7. L285: In northern China, SO2, especially during wintertime, may also come from domestic heating, which can substantially increase SO2 emission. The authors may want to include this possibility.

8. Section 3.5: Clear particle growth observed at a ground site is often associated with a regional NPF event. The short burst of nucleation events reported here may indicate that the air parcels were frequently disrupted by the urban micrometeorology conditions, which should be further investigated.
* * *

---

## Referee Comment (RC3) · Anonymous Referee #3 · 24 Jan 2017

The manuscript presents results from measurements of new particle formation (NPF) in Beijing during Spring and Winter using high time resolution particle sizers. The characteristics of new particle formation at two adjacent sites were compared to assess the impacts of local traffic emissions on NPF. Traffic exhausts might emit primary particles or gas phase precursors that contribute significantly to secondary particle formation and hence traffic emissions play important roles in severe haze formation in megacities such as Beijing in China. While the topic is important in atmospheric chemistry and is of interest to the general readers of this journal, the manuscript in general is yet to be improved and several major issues need to be resolved before the manuscript can be publishable in the journal.

Major comments:

1. The rational of selecting plume events rather than regional events as examples must be clearly persuasive. Apparently, good "Banana shape" regional events were measured during the campaigns. Comparison of the differences of new particle formation between the two sites is of great interest. It will be clearer to see the impacts of traffic emissions on the new particle formation processes if those well-defined events were used as examples. For example, how particle number size distribution and particle composition might be affected by local emissions. The plume events are rather not well defined in term of the formation rates and the growth rates which will need to be resolved in the next comment.

2. Particle formation rates and growth rates. First, the formation rate should be stick to J8 instead of new particle formation rate since particles in the range of 8-20 nm are rather too big to be called new particles. Also why a size range of 8-20nm is selected? Why is not 8-30 or 40 nm or others? The determination of the formation time (not the nucleation time) used in this paper seems to be objective and is of the authors' preferences, profoundly affecting the formation rate calculations. The width of the size range also affects the determination of formation time which will need to be clarified. The formation rate of nanoparticles in a plume event is difficult to calculate and needs to use a more sophisticate method than the simple one used in this paper.

3. The classification of the nucleation events. It is very awkward to denote a nucleation event longer than 1 hour "a long term event". It might sound better if "a long lasting event" or another name is adopted. Similarly, please change the notation of "a short term event" to another proper name. In addition, the two types of events are the well-known "regional events" and "plume events" in atmospheric aerosol sciences. It is not necessary to create new names for them.

4. Heterogeneity of NPF in Section 3.2. Quite a few "heterogeneity" were mentioned for NPF in both horizontal and vertical directions. It is really not that meaningful to

emphasize the spatial inhomogeneity of particle formation because in the urban atmosphere, gas phase precursors are inhomogeneous and particle formation is also greatly constrained by local emissions and meteorological conditions.

5. Reasons for the reduced NPF and the enhanced NPF at the street site respectively in the springtime and in the wintertime. It is very interesting to figure out the reasons behind those observed phenomena. First, the authors need to confirm that particle formation is always reduced in the springtime and always enhanced in the springtime at the street site. That will exclude the possibility of dominant effects from the meteorological conditions e.g. differences in wind directions or mixing heights, temperature, humidities etc. Second, the authors need to present more other companion measurements of gas phase precursors and chemical composition of nanoparticles in order to elucidate the mechanisms of reduced or enhanced NPF at the street site. Without the information, the proposed explanation for the observed opposite effects on NPF during springtime and wintertime is only speculative.

Minor comments:

1. There are a lot of typos, ill-sentences all over the manuscript. It is recommended that the manuscript should be carefully edited prior to submission. Below are a few examples: L31, specie; L208, a several minutes; L254, didn't detail; L270, didn't detail description; L292, "were available currently", . . .

2. Rewrite all the figure captions clearly as those captions are hard to read and understand.

3. L225, a few more sentences might be needed to explain why NPF inside the street canyon was reduced.

4. L189, "estimating that the NPF possibly occurred in cleaner atmospheres over the region scale of ∼120 km", "suggesting that the NPF likely occurred in cleaner atmospheres over the region scale of ∼140 km in different NPF rates.", "The NPF was

roughly estimated to occur in a semi-regional scale over ∼50 km". How do you know the scales of those events?

---

## Referee Comment (RC4) · Anonymous Referee #4 · 24 Jan 2017

Review of "Simultaneous measurements of new particle formation in 1-second time resolution at a street site and a rooftop site" by Zhu et al. (acp-2016-1143)

This manuscript presents a field measurement of new particle formation (NPF) events in urban Beijing, China. The deployment of Fast Mobility Particle Sizers (FMPs) is unique and could deliver new insights into NPF, if interpreted properly. Overall, this manuscript describes interesting phenomena that NPF was enhanced in winter at a street site comparing to a close rooftop site, whereas NPF was less pronounced at the street site in spring. The explanation for these observation, unfortunately, is not well justified, and requires a major work over again. Here are my detailed comments,

Main comments, 1. Micro-meteorology could be a major player that explains the differ-

ence between the street site and the rooftop site, which is not discussed at all in the current manuscript. Potentially, the loss of nanoparticles due to the surfaces along the street canyon is a factor too.

2. The inter-comparison between two FMPSs showed some differences, and the authors decided to use one FMPS as a reference and correct the number concentration of the other one. How did they decide which one is "the one" to trust? Nevertheless, number concentrations are used to calculate formation rates, growth rates, and condensation sink. This could lead to a major uncertainty in the discussion for nucleation mechanism.

3. The FMPSs were placed downstream of dryers, which indicates that the measured size distributions could be of from the atmospheric ones. This at least eliminates the role of relative humidity to a certain extent. Even for particles in the size ranges of 10-20 nm, the uptake of $H_2O$ is one of the major pathways for particle to grow.

4. The mixing ratio of $SO_2$ was only measured at the rooftop site. How about CO? It might be possible to deduce a street $SO_2$ simply by the mixing ratios of CO. The current assumption that concentrations of $SO_2$ are identical at the two sites are not acceptable, and could lead to mis-interpretation.

5. The authors focused on the oxidation of biogenic organics when discussing the growth of >10 nm particles. In an urban environment such as Beijing, wouldn't anthropogenic VOCs be more concentrated? Are there any measurements that point the authors to biogenic VOCs instead of anthropogenic ones? How will the interpretation

6. Throughout the manuscript, the authors are presenting J8, which is fine. However, particles bigger than 8 nm are larger enough that they don't really reflect the nucleation mechanism, instead, a combination of nucleation and subsequent growth, especially growth mechanisms, might actually determines how many particles were measured.

Minor comments, 7. (Page 8), clearly define long-term NPN, short-term NPF, Class I

NPF, and class II NPF.

8. (Page 10, Line 225), how about NO in the winter? Wouldn't NO be always higher in the street canyon?

9. (Page 13, Lines 280-287), the argument on H2SO4 is just speculation. Many factors determines H2SO4. Also, why is SO2 from on-road vehicles negligible comparing to the background?

10. (Page 15, the last paragraph), I am certainly not convinced by the discussion here. Different nucleation mechanisms probably explains NPF events in Beijing, Qingdao, and marginal seas of China. Again, J8 is not a good indicator for nucleation mechanisms. By definition, NMIoNP stand for "the net maximum increase of nucleation mode PNC". I don't see a clear connection between NMIoNP and J8. A cutoff of 8 cm-3 s-1 could be arbitrary. The correlation will not be bad if a cutoff of , say, 7 cm-3 s-1, was chosen.

11. In supplementary, coefficient of variation (CV) is defined, but try to define "25% minimum", especially what "minimum" stands for. Also, why 1 16.6 nm cutoff chosen in the following session?

12. Proofread the manuscript.

---

## Referee Comment (RC5) · Anonymous Referee #5 · 30 Jan 2017

Fast-response measurements of particle number size distributions of aerosol $\geq$8 nm diameter have been made at a street canyon and nearby rooftop site. The authors selectively report specific days of data from a small dataset, and draw many tentative conclusions concerning mechanisms of new particle formation (NPF) which are difficult to justify given the small dataset and the extent to which it is over-interpreted. The introduction quite reasonably states that "it is critical to evaluate the effects of nucleating species other than sulfuric acid and the dependence of NPF on pre-existing particles in the atmosphere". This is an excellent objective but unfortunately the paper does nothing to answer the question about other nucleating species, and does not event provide clear answers concerning the role of sulfuric acid.

One of the key elements towards interpretation of this dataset in relation to nucleation and growth is the role of sulfuric acid, which ideally would have been measured. However, as measurements were not available, an old parameterisation is used to estimate H2SO4 vapour concentrations in which the H2SO4 formation rate is described by the product of SO2 concentration and global solar radiation. This may be adequate for situations in the background troposphere where ozone photolysis is the predominant source of hydroxyl radical, but many studies have now shown that in polluted atmospheres such as Beijing, other processes such as photolysis of HONO and HCHO, and ozone-alkene reactions are far more important sources of hydroxyl, and equation 4 is unlikely to be a reliable means of calculation of [H2SO4].

The differences in behaviour between the sites are interesting, and if correctly interpreted could give useful insights into NPF in polluted atmospheres. However no measurements were made of potentially condensing species, or their precursors other than SO2, and the latter was measured at only one site with the unproven assumption that concentrations of SO2 were the same at both sites. Much is made of the rates of change of particle number concentrations, but the effects of wind direction changes upon concentrations in the street canyon (which can be large) do not appear to have been considered. The methods used for subtraction of fresh traffic emissions are highly questionable, and no use is made of gaseous pollutant data (e.g. NOx) which would be a strong covariate of PNC from road vehicles.

The points above justify a major reappraisal of the data, and the development of far less ambitious conclusions. Other points which need to be addressed include:

(a) The introduction lists a number of organic acids as examples of vehicle-emitted organic compounds. Most of these have far more major secondary sources, or are present in cooking emissions, with little if any arising from road traffic.

(b) Some ill-informed statements are made about the (currently uncertain) effects of exposure to ultrafine particles. These particles do not lead to "destruction of the respiratory system" and the statement that "newly formed particles inside a street canyon may become toxic when vehicle-release organics is involved in the nucleation process" is not supported by references.

(c) There is no information on quality assurance beyond an intercomparison between the two FMPS, and no consideration of how size-dependent particle losses in the inlet system affect measured size distributions.

(d) Equation (3) differs from that in the nucleation protocol paper of Kulmala et al. (2012) by a factor of two, which needs to be explained.

(e) A clear definition is needed for the "maximum increase of nucleation mode PNC (NMIoNP)" which is much used in the data analyses.

(f) The authors should establish that their Class II particles arise from an NPF event, rather than an emission source.

––––––––––––––––––––

---

## Author Comment (AC1) · 7 Apr 2017

This manuscript investigates new particle formation (NPF) observed simultaneously at two sites in a polluted urban environment. The analysis is based high-time resolution measurements, which increases the originality of the results. The background for this study (section 1) as well as the used methods (section 2) are very well written. Contrary to this, there are serious problems in how many of the results, have been interpreted. As a result, a large part of section 3 needs substantial revisions, and most of the sections 3.4-3.6 need to be entirely re-written. My detailed comments in this regard are given below.

Response: The authors thank the reviewer's comments and try our best to respond

[Figure]

and revise our manuscript accordingly.

Major comments:

The authors provide two very general statements based on their results: 1) reduced NPF at street site compared to rooftop during spring, and 2) enhanced NPF at street site compared with rooftop in winter. These finding are supported by only 1-2 cases (days) of observations, which is way too little to make this kind of a general conclusion.

Response: In revision, the authors will add "At the street site, the reduced NPF events always occurred in the springtime while the enhanced NPF events always occurred in the wintertime."

The authors would like to believe the sufficiency and uniqueness of evidences are crucial to evaluate the quality of scientific studies. This is because the number of cases for gravitational wave observation in 2016 and a recent NPF study reported by Bianchi et al. (2016) was even smaller than those presented in this study. The authors thereby abide by a principle, i.e., it is theoretically reasonable, multiple-evidences supported and no exception against it, to justify our results on reduced NPF at the street site, i.e., 1) Considered the widely recognized the importance of condensation sink in new particle formation (NPF), reduced NPF at the street site is theoretically expected and repeatedly occurred in the springtime. The authors provided three types of evidences from different angles to confirm the reduced NPF rather than simple comparison between rooftop site and street site measurements, i.e., Evidence 1: The lower particle number concentration (PNC) of nucleation mode particles at the street site mainly because of a shorter initial burst time. Evidence 2: The authors used the PNC at the street site subtracting the corresponding PNC at the rooftop site to calculate the difference. The authors then obtained the second evidence: the negative difference of nucleation mode particles against the positive difference of Aitken mode particles on NPF days. Evidence 3: Using the same approach, the authors obtained the third evidence: the negative difference of nucleation mode particles on NPF days against the positive dif-

ference of that on non-NPF days (Figs. 3 and 4 in the origin version). In addition, the authors also provided three types of evidences from different angles, rather than simple comparison between rooftop site and street site to confirm the enhanced NPF in the wintertime, i.e., Evidence 1: The significantly larger PNC of nucleation mode particles at the street site and a larger apparent formation rate of new particles mainly because of a shorter initial burst time. Evidence 2: The positive difference of nucleation mode particles in the wintertime against the negative difference of nucleation mode particles in the springtime on NPF days. Evidence 3: The larger positive difference of nucleation mode particles on NPF days against that on non-NPF days in the wintertime (Figs. 5 and 7 in the origin version).

According to the comments, the authors will revise the manuscript to make the unique evidences to be more obvious.

The used proxy for gaseous sulfuric acid (SA) concentration has two problems: 1) it has been developed and evaluated for moderately-polluted sites only, so its applicability in highly-polluted sites like this one may be questionable.

Response: In this study, the NPF events occurred under the north or northwest wind direction with wind speed >4m/s. The north or northwest wind carried less polluted or even clear ambient air to the sampling site during the NPF periods, e.g., the mixing ratio of SO2 was < 3 ppb in the springtime and < 5 ppb in the wintertime during the periods of NPF events. Less polluted or even clear ambient air exactly meets the reviewer claimed, i.e., the proxy for calculating gaseous sulfuric acid (SA) concentration is applicable only under clean to moderately-polluted atmospheres. The authors thereby believe that our approach is consistent with the well-established knowledge and is thereby scientifically valid.

2) SO2 is measured at rooftop site only, so it is unclear how well this represents SO2 in the street site. Also, the ratio in the SO2 concentration between the street site and rooftop is likely to be different between spring and winter, and there is no means to

estimate this difference. As a result, the authors need to be very careful when making any interpretations that rely on estimated SA concentrations.

Response: The authors thank the comments. In revision, we will add "The sulfur content in the gasoline and diesel was limited <50 ppm at those years. The measured BC spikes were lower than 5 $\mu$g m-3 during the NPF periods. The maximum contribution of traffic-related SO2 at the street site was roughly estimated to be 1.3 ppb according to the results in our previous studies (Meng et al., 2015 a,b)". In the wintertime, the ratio of traffic-derived SO2 to the observed values was less than 1/4 and the observed values were overwhelmingly contributed by domestic heating. The uncertainty by assuming SO2 at the street site same as the rooftop site should be minor in the wintertime and it should not affect our conclusion because the apparent formation rates of new particles at the street site were increased by 3-5 times against the rooftop site in the wintertime. In the springtime, the contribution of traffic-related SO2 might significantly increase the mixing ratio of SO2 at the street site. However, the reduced NPF was observed at the street site. The possible underestimation of SO2 at the street site further solidified our analysis results, i.e., a strong scavenge effect at the street site likely existed and caused the reduced NPF.

Class II NPF events have very low particle growth rates above 10 nm. All theoretical arguments indicate that >10 nm particles grow faster than smaller particles, and practically all observations on size-resolved particle growth rates support this view. This lead to a serious question: what is the origin of these particles? More specifically, if there are little condensable vapours to growth >10 nm, there should be even less vapors to grow smaller particles. One possible explanation for this is that particle of Class II originate from very local NPF, in which high local vapor concentrations initial nucleation and make the formed particles to grow very rapidly to a few nm, even to 10-20 nm. This rapid growth is then stopped due to atmospheric dilution of emitted vapors. This kind of process has been reported to occur in some coastal areas (Mace Head), in car exhaust to ambient air, and also close to other localised combustion sources. If

Case II event are caused by very localized sources, it is questionable to compare NPF between the street site and rooftop in such cases.

Response: The reviewer's first statement is probably contradictory to the truth. For example, the data results in Table 1 recently reported by Yu et al. (2016) fight against the reviewer's first statement. In the first publication on NPF events in Beijing (Wehner et al., 2004), the observed growth rate of new particles was as low as $\sim$1 nm h-1. In the study, however, the formation of new particles started around 07:00 after sunrise and the initial size of newly formed particles was $\sim$5 nm. The results also indirectly fight against the reviewer's statement. Theoretically, when the volume concentration of particles is considered, the amount of chemical species required for growing >10 nm particles was much larger than that for <10 nm particles. For example, the amount of chemicals required for growing particles from 10 nm to 12 nm was about six times larger than particles grew from 3 nm to 5 nm. Furthermore, the coagulation growth is important for <10 nm ambient particles while it is negligible for >10 nm ambient particles. As reviewed by Vu et al. (2015), the particle number size distribution (PNSD) of vehicle or combustion plumes character the typical peak number mode such as at 30 nm, 50 nm, 70-80 nm, etc. In our study, when the NPF events in Class II occurred, the nucleation mode particles overwhelmed and other particle modes were negligible. The duration period of Class II lasted for 4-8 hours with the wind speed >4m/s, suggesting they probably happened in regional scale. The authors have no idea to link NPF in the urban atmosphere of Beijing (an inland megacity where ocean-derived reactive iodides were unexpected) with those in rural coastal atmospheres, e.g., Mace Head where ocean-derived reactive iodides could be important precursors for NPF events. The authors may have no comments on the reviewer's speculation.

The authors use condensation sink (CS) in interpreting their results. This problematic. The particles are formed below 2 nm size (J<2), but the authors calculate the formation rate of 8 nm particles (J8). The value of J8 depends on 3 quantities. J<2, CS and the growth rate of particles below 8 nm. Since neither J<2 nor the sub-8 nm growth rate

are known, it is impossible to infer how CS might affect J8 in the observed cases.

Response: The authors fully respect the reviewer's knowledge on the issue. However, the condensation sink has been widely used to argue the occurrence of NPF in literature when neither J<2 nor the sub-8 nm growth rate were not available, e.g., Kulmala et al. (2004, 2016).

The authors assumed that only biogenic organics could influence NPF and subsequent growth. Why? There certainly large anthropogenic emissions of organic vapours in this kind of environment, and the oxidation of such vapors is very likely to produce low-volatile compounds that could affect nanoparticle formation and growth.

Response: The authors never assumed "only biogenic organics could influence NPF and subsequent growth" in the manuscript. The role of oxidation products of biogenic VOC in NPF events have been widely studied in field experiments, chamber and modeling studies, and quantum chemical calculations (Schobesberger et al., 2013; Riccobono et al., 2014; Ortega et al., 2015; Tröstl et al., 2016). According to the established knowledge, the authors argued the potential importance of oxidized biogenic VOC in growing newly formed particles in this study. The north and northwest directions of the sampling site subject to mountain areas have a high percentage of land-covered forests. Extensive biogenic VOC is theoretically expected in spring and may act as important precursors in NPF. During the NPF periods, the north or northwest wind dominated and carried less polluted or even clear ambient air from mountain areas to the sampling site, e.g., the mixing ratio of SO2 was < 3 ppb in the springtime and < 5 ppb in the wintertime during the periods of NPF events.

The role of oxidized anthropogenic VOCs in growing >10 nm newly formed particles is still poorly understood (Zhang et al., 2009; Hoyle et al., 2011). Considering the knowledge gap and lack of related data in this study, the authors are reluctantly to discuss the possibility in this study. Following the reviewer's comments, in the context, the authors will add "Theoretically, oxidized anthropogenic VOCs could also participate

in the growth of newly formed particles while the study is limited (Zhang et al., 2009; Hoyle et al., 2011). The role of oxidized anthropogenic VOCs needs further study."

Considering the points highlighted above, many of the interpretations made in sections 3.4-3.6 are not justified. The most problematic of these is section 3.6 which is highly speculative.

Response: The comments are general and don't contain helpful information for revision. The authors thank the reviewer's comments and try our best to respond and revise our manuscript accordingly.

Minor comments

I would recommend using terms other than short-term and long-term NPF events. In atmospheric time series, long-term usually means something that last for years or at least for months.

Response: Agree. It will be revised as "short-lived NPF events, regional NPF events" (Stanier et al., 2010; Jeong et al., 2010).

line 208: should be written: . . .only lasted for few minutes

Response: It will be corrected in the revision.

lines 254 and 270: did't detail is a strange expression. Please modify

Response: The sentences are indeed ambiguous and unnecessary. Therefore, it will be deleted in revision.

line 320: what is meant by ..reaction should proceed to solid state

Response: It will be revised as "reaction should proceed to solid state, i.e., the gases start to partition on the particle phase."

References:

Bianchi, F., Tröstl, J., Junninen, H., Frege, C., Henne, S., Hoyle, C. R., Molteni, U.,

Herrmann, E., Adamov, A., Bukowiecki, N., Chen, X., Duplissy, J., Gysel, M, Hutterli, M., Kangasluoma, J., Kontkanen, J., Kürten, A., Manninen, H. E., Münch, S., Peräkylä, O., Petäjä, T., Rondo, L., Williamson, C., Weingartner, E., Curtius, J., Worsnop, D. R., Kulmala, M., Dommen, J., and Baltensperger, U.: New particle formation in the free troposphere: A question of chemistry and timing. Science, 352(6289), 1109-1112, 2016.

Hoyle, C. R., Boy, M., Donahue, N. M., Fry, J. L., Glasius, M., Guenther, A. Hallar, A. G., Huff Hartz, K., Petters, M. D., Petäjä, T., Rosenoern, T., and Sullivan, A. P.: A review of the anthropogenic influence on biogenic secondary organic aerosol. Atmos. Chem. Phys., 11(1), 321-343, 2011.

Jeong, C. H., Evans, G. J., McGuire, M. L., Chang, R. W., Abbatt, J. P. D., Zeromskiene, K., Mozurkewich, M., Li, S.-M. and Leaitch, W. R.: Particle formation and growth at five rural and urban sites. Atmos. Chem. Phys., 10(16), 7979-7995, 2010.

Kulmala, M., Vehkamäki, H., Petäjä, T., Dal Maso, M., Lauri, A., Kerminen, V. M., Birmilic, W., and McMurry, P. H.: Formation and growth rates of ultrafine atmospheric particles: a review of observations. J. Aerosol Sci., 35(2), 143-176, 2004.

Kulmala, M., Petäjä, T., Kerminen, V. M., Kujansuu, J., Ruuskanen, T., Ding, A., Nie, W., Hu, M., Wang, Z., Wu, Z., Wang, L., and Worsno, D.: On secondary new particle formation in China. Front. Environ. Sci. Eng., 10(5), 1-10, 2016.

Meng, H., Zhu Y. J., Evans G., and Yao X. H.: An approach to investigate new particle formation in the vertical direction on the basis of high time-resolution measurements at ground level and sea level. Atmos. Environ., 102, 366-375, 2015a.

Meng, H., Zhu, Y. J., Evans, G., Jeong, C. H., and Yao, X. H.: Roles of SO2 oxidation in new particle formation events. J. Environ. Sci., 30, 90-101, 2015b.

Ortega, I. K., Donahue, N. M., KurteÌA̧n, T., Kulmala, M., Focsa, C., and VehkamaÌĹki, H.: Can Highly Oxidized Organics Contribute to Atmospheric New Particle Formation?

J. Phys. Chem. A, 120(9), 1452-1458, 2015.

Riccobono, F., Schobesberger, S., Scott, C. E., Dommen, J., Ortega, I. K., Rondo, L., Almeida, J., Amorim, A., Bianchi, F., Breitenlechner, M., David, A., Downard, A., Dunne, E. M., Duplissy, J., Ehrhart, S., Flagan, R. C., Franchin, A., Hansel, A., Junninen, H., Kajos, M., Keskinen, H., Kupc, A., Kürten, A., Kvashin, A. N., Laaksonen, A., Lehtipalo, K., Makhmutov, V., Mathot, S., Nieminen, T., Onnela, A., Petäjä, T., Praplan, A. P., Santos, F. D., Schallhart, S., Seinfeld, J. H., Sipilä, M., Spracklen, D. V., Stozhkov, Y., Stratmann, F., Tomé, A., Tsagkogeorgas, G., Vaattovaara, P., Viisanen, Y., Vrtala, A., Wagner, P, E., Weingartner, E., Wex, H., Wimmer, D., Carslaw, K, S., Curtius, J., Donahue, N. M., Kirkby, J., Kulmala, M., Worsnop, D. R., and Baltensperger, U.: Oxidation products of biogenic emissions contribute to nucleation of atmospheric particles, Science, 344, 717-721, 2014.

Schobesberger, S., Junninen, H., Bianchi, F., Lönn, G., Ehn, M., Lehtipalo, K., Dommen, J., Ehrhart, S., Ortega, I. K., Franchin, A., Nieminen, T., Riccobono, F., Hutterli, M., Duplissy, J., Almeida, J., Amorim, A., Breitenlechner, M., Downard, A. J., Dunne, E. M., Flagan, R. C., Kajos, M., Keskinen, H., Kirkby, J., Kupc, A., Kürten, A., Kurtén, T., Laaksonen, A., Mathot, S., Onnela, A., Praplan, A. P., Rondo, L., Santos, F. D., Schallhart, S., Schnitzhofer, R., Sipilä, M., Tomé, A., Tsagkogeorgas, G., Vehkamäki, H., Wimmer, D., Baltensperger, U., Carslaw, K. S., Curtius, J., Hansel, A., Petäjä, T., Kulmala, M., Donahue, N, M., and Worsnop, D. R.: Molecular understanding of atmospheric particle formation from sulfuric acid and large oxidized organic molecules. Proc. Natl. Acad. Sci., 110(43), 17223-17228, 2013.

Stanier, C. O., Khlystov, A. Y., and Pandis, S. N.: Nucleation events during the Pittsburgh Air Quality Study: description and relation to key meteorological, gas phase, and aerosol parameters special issue of aerosol science and technology on findings from the fine particulate matter supersites program. Aerosol Sci. Technol., 38(S1), 253-264, 2004.

Tröstl, J., Chuang, W. K., Gordon, H., Heinritzi, M., Yan, C., Molteni, U., Ahlm, L., Frege, C., Bianchi, F., Wagner, R., Simon, M., Lehtipalo, K., Williamson, C., Craven, J. S., Duplissy, J., Adamov, A., Almeida, J., Bernhammer, J. A. K., Breitenlechner, M., Brilke, S., Ehrhart, S., Dias, A., Flagan, R. C., Franchin, A., Fuchs, C., Guida, R., Gysel, M., Hansel, A., Hoyle, C. R., Jokinen, T., Junninen, H., Kangasluoma, J., Keskinen, H., Kim, J., Krapf, M., Kürten, A., Laaksonen, A., Lawler, M., Leiminger, M., Mathot, S., Möhler, O., Nieminen, T., Onnela, A., Petäjä, T., Piel, F. M., Miettinen, P., Rissanen, M. P., Rondo, L., Sarnela, N., Schobesberger, S., Sengupta, K., Sipilä, M., Smith, J. N., Steiner, G., Tomè, A., Virtanen, A., Wagner, A. C., Weingartner, E., Wimmer, D., Winkler, P. M., Ye, P. L., Carslaw, K. S., Curtius, J., Dommen, J., Kirkby, J., Kulmala, M., Riipinen, I., Worsnop, D. R., Donahue, N. M., and Baltensperger, U.: The role of low-volatility organic compounds in initial particle growth in the atmosphere, Nature, 533, 527-531, 2016.

Vu, T. V., Delgado-Saborit, J. M., and Harrison, R. M.: Review: Particle number size distributions from seven major sources and implications for source apportionment studies. Atmos. Environ., 122, 114-132, 2015.

Wehner, B., Wiedensohler, A., Tuch, T. M., Wu, Z. J., Hu, M., Slanina, J., and Kiang, C. S.: Variability of the aerosol number size distribution in Beijing, China: New particle formation, dust storms, and high continental background. Geophys. Res. Let., 31, L22108, 2004.

Yu, H., Zhou, L., Dai, L., Shen, W., Dai, W., Zheng, J., Ma, Y., and Chen, M.: Nucleation and growth of sub-3 nm particles in the polluted urban atmosphere of a megacity in China. Atmos. Chem. Phys., 16(4), 2641-2657, 2016.

Zhang, R., Wang, L., Khalizov, A. F., Zhao, J., Zheng, J., McGraw, R. L., and Molina, L. T.: Formation of nanoparticles of blue haze enhanced by anthropogenic pollution. Proc. Natl. Acad. Sci., 106(42), 17650-17654, 2009.

---

## Author Comment (AC3) · 7 Apr 2017

The manuscript presents results from measurements of new particle formation (NPF) in Beijing during Spring and Winter using high time resolution particle sizers. The characteristics of new particle formation at two adjacent sites were compared to assess the impacts of local traffic emissions on NPF. Traffic exhausts might emit primary particles or gas phase precursors that contribute significantly to secondary particle formation and hence traffic emissions play important roles in severe haze formation in megacities such as Beijing in China. While the topic is important in atmospheric chemistry and is of interest to the general readers of this journal, the manuscript in general is yet to be improved and several major issues need to be resolved before the manuscript

can be publishable in the journal.

Response: The authors thank the reviewer's comments and try our best to address the issues point-by point.

Major comments:

1. The rational of selecting plume events rather than regional events as examples must be clearly persuasive. Apparently, good "Banana shape" regional events were measured during the campaigns. Comparison of the differences of new particle formation between the two sites is of great interest. It will be clearer to see the impacts of traffic emissions on the new particle formation processes if those well-defined events were used as examples. For example, how particle number size distribution and particle composition might be affected by local emissions. The plume events are rather not well defined in term of the formation rates and the growth rates which will need to be resolved in the next comment.

Response: All simultaneous observations at two sites have been included and presented in the original version. During the two periods of this study, the authors didn't observe "banana shape" regional events simultaneously occurring at two sites. Therefore, the authors had no way to use them for discussion. This will be clarified in revision.

Similar to Class II NPF events with the particle growth to be undetectable presented in this study, extremely low growth rate of newly formed particles ( $\sim 1 \text{ nm h-1}$ ) in Beijing was also previously reported by Wehner et al. (2004). In our unpublished data, the authors simultaneously observed Class II NPF event and NPF event with extremely low growth rate at  $\sim$ 240 km distance (Fig. 1, the case will also be presented in Supplementary). In the last three years (data unpublished), we had simultaneous observations of NPF events at 100-500 km distance. The authors obtained six cases based on simultaneous observations at two locations, i.e., one case featured by Class II NPF vs Class II NPF, four cases featured by Class II NPF vs NPF with an extremely low growth rate, one case featured by Class II NPF vs NPF with "banana shape" particle growth.
The authors strongly believed that Class II NPF events lasted for 4-8 hours should be considered as regional NPF events. "Banana shape" particle growth, extremely low particle growth and no detectable growth are probably related to spatial heterogeneity of gaseous precursors supporting the growth of > 10 nm particles, but NPF events indeed occurred regionally.

FMPS measured different sized particle number concentration in one second. The high time-resolution data can allow calculating the apparent formation rate of new particles such as J8 almost in any complicated situations. The reviewer's comments may be related to the low time-resolution data such as in 5-10 minutes time resolution, but it should not be the problem for FMPS measurements. This is also probably why FMPS measurements can show a few unique findings like in this study, but it is impossible for the 5-10 minutes time-resolution measurement previously conducted.

2. Particle formation rates and growth rates. First, the formation rate should be stick to J8 instead of new particle formation rate since particles in the range of 8-20 nm are rather too big to be called new particles. Also why a size range of 8-20nm is selected? Why is not 8-30 or 40 nm or others? The determination of the formation time (not the nucleation time) used in this paper seems to be objective and is of the authors' preferences, profoundly affecting the formation rate calculations. The width of the size range also affects the determination of formation time which will need to be clarified. The formation rate of nanoparticles in a plume event is difficult to calculate and needs to use a more sophisticate method than the simple one used in this paper.

Response: The authors agree that 8-20 nm particles should be called as grown new particles rather than new particles. In revision, we will add "Noted that J8 reflects a combination of nucleation and subsequent initial growth of new particles." in the explanation of new particle formation rate.

The authors followed the conventional approach to calculate the apparent formation rate of new particles. In literature, the upper limits of the particle size was reportedly

**ACPD**
set at 20 nm, 25 nm or 30 nm (Kulmala et al., 2004; Kulmala and Kerminen, 2008; Yao et al., 2010; Kulmala et al., 2012). 20 nm was set as the upper limit in this study, following the suggestions by Kulmala et al. (2012), i.e., 1) newly formed particles rarely grow over 20 nm during the initial nucleation time, 2) nucleation may occur at some distances from the measurement point and the particles have grown a few nanometers before they were observed. In addition, the upper limit of 20 nm can minimize possible interferences from primary traffic particles. This will be clarified in revision.

FMPS measured different sized particle number concentration in one second. The high time-resolution data allow calculating the apparent formation rate of new particles such as J8 almost in any complicated situations. The reviewer's comments may be related to the low time-resolution data such as in 5-10 minutes time resolution, but it should not be the problem for FMPS measurements. This is also probably why FMPS measurements can show a few unique findings like in this study, but it is impossible for the 5-10 minutes time-resolution measurement previously conducted.

The authors used two methods to minimize the interference from primary traffic particles and the calculated apparent formation rates were consistent. Again, the high time-resolution data indeed allows calculating the apparent formation rate of new particles such as J8 in presence of strong interferences.

3. The classification of the nucleation events. It is very awkward to denote a nucleation event longer than 1 hour "a long term event". It might sound better if "a long lasting event" or another name is adopted. Similarly, please change the notation of "a short term event" to another proper name. In addition, the two types of events are the well-known "regional events" and "plume events" in atmospheric aerosol sciences. It is not necessary to create new names for them.

Response: The authors would like to correct "short-term NPF event, long-term NPF event" to "short-lived NPF event, regional NPF event" (Stanier et al., 2010; Jeong et al., 2010).
The authors strongly believed that Class II NPF events lasted for 4-8 hours should be considered as regional NPF events. Details can be found in the response to major comment 1.

4. Heterogeneity of NPF in Section 3.2. Quite a few "heterogeneity" were mentioned for NPF in both horizontal and vertical directions. It is really not that meaningful to emphasize the spatial inhomogeneity of particle formation because in the urban atmosphere, gas phase precursors are inhomogeneous and particle formation is also greatly constrained by local emissions and meteorological conditions.

Response: The authors thank and fully agree "the spatial inhomogeneity of particle formation because in the urban atmosphere, gas phase precursors are inhomogeneous and particle formation is also greatly constrained by local emissions and meteorological conditions". A question is then automatically raised, i.e., what does it mean "regional NPF events"? It is just because it regionally occurred, but could be in different formation rates, growth rates, different initial bursting times? Most potential readers may disagree.

Most readers may also disagree with the reviewer and the authors at this point, i.e., the spatial inhomogeneity of particle formation very commonly occur in regional NPF events. However, the authors agree to cut a few details of "heterogeneity" in revision.

5. Reasons for the reduced NPF and the enhanced NPF at the street site respectively in the springtime and in the wintertime. It is very interesting to figure out the reasons behind those observed phenomena. First, the authors need to confirm that particle formation is always reduced in the springtime and always enhanced in the springtime at the street site. That will exclude the possibility of dominant effects from the meteorological conditions e.g. differences in wind directions or mixing heights, temperature, humidities etc. Second, the authors need to present more other companion measurements of gas phase precursors and chemical composition of nanoparticles in order to elucidate the mechanisms of reduced or enhanced NPF at the street site. Without the

**ACPD**
information, the proposed explanation for the observed opposite effects on NPF during springtime and wintertime is only speculative.

Response: At the street site, the reduced NPF events always occurred in the springtime while the enhanced NPF events always occurred in the wintertime. This will be highlighted in revision.

The authors provided three types of evidences from different angles to confirm the reduced NPF rather than simple comparison between rooftop site and street site measurements, i.e., Evidence 1: The lower particle number concentration (PNC) of nucleation mode particles at the street site mainly because of a shorter initial burst time. Evidence 2: The authors used the PNC at the street site subtracting the corresponding PNC at the rooftop site to calculate the difference. The authors then obtained the second evidence: the negative difference of nucleation mode particles against the positive difference of Aitken mode particles on NPF days. Evidence 3: Using the same approach, the authors obtained the third evidence: the negative difference of nucleation mode particles on NPF days against the positive difference of that on non-NPF days (Figs. 3 and 4 in the origin version). The second and third evidences are not affected by these factors suggested by the reviewer. All these will be better clarified in revision.

The authors also provided three types of evidences from different angles, rather than simple comparison between rooftop site and street site to confirm the enhanced NPF in the wintertime, i.e., Evidence 1: The significantly larger PNC of nucleation mode particles at the street site and a larger apparent formation rate of new particles mainly because of a shorter initial burst time. Evidence 2: The positive difference of nucleation mode particles in the wintertime against the negative difference of nucleation mode particles in the springtime on NPF days. Evidence 3: The larger positive difference of nucleation mode particles on NPF days against that on non-NPF days in the wintertime (Figs. 5 and 7 in the origin version). Again, the second and third evidences are not affected by these factors suggested by the reviewer. All these will be better clarified in revision.
To confirm what the reviewer suggested requires simultaneous measurements of vapor precursors such as H2SO4, HOM, etc, and chemical composition of  $\sim$ 10 nm particles at the rooftop site and street site. This is indeed beyond the capacity of the whole research community, but not only for the authors. The authors tried the best and used all data we had. Other indirect evidences such as chemical composition of >50 nm particles were reportedly used to argue NPF in literatures. Theoretically, the arguments could be true, but also could be misleading by considering largely varying chemical composition in different sized atmospheric nanoparticles. The authors hope that the reviewer can agree this.

Minor comments:

1. There are a lot of typos, ill-sentences all over the manuscript. It is recommended that the manuscript should be carefully edited prior to submission. Below are a few examples: L31, specie; L208, a several minutes; L254, didn't detail; L270, didn't detail description; L292, "were available currently", . . .

Response: The authors are sorry for this and will correct the errors through the manuscript before re-submitting.

2. Rewrite all the figure captions clearly as those captions are hard to read and understand.

Response: Thank for the comment, the authors will rewrite all the figure captions.

3. L225, a few more sentences might be needed to explain why NPF inside the street canyon was reduced.

Response: The authors will revise L225 as: "Several factors can lead to the reduced NPF events at the street site, i.e.., 1) a larger condensation sink because of more preexisting atmospheric particles from primary emissions; 2) tall buildings along both the sides of urban streets can provide additional surface areas to scavenge gases and atmospheric particles (Yao et al., 2011); 3) vehicle-emitted NO reacting with RO2 and Interactive comment

suppressing NPF (Wildt et al., 2014).

4. L189, "estimating that the NPF possibly occurred in cleaner atmospheres over the region scale of ~120 km", "suggesting that the NPF likely occurred in cleaner atmospheres over the region scale of ~140 km in different NPF rates.", "The NPF was roughly estimated to occur in a semi-regional scale over  $\hat{a}$ Lij50 km". How do you know the scales of those events?

Response: The authors agree that our estimation indeed suffered from uncertainty. In revision, it will revise as "the long lasting time for NPF implied that the event may occur in regional or semi-regional scale." Again, the authors believed that the long lasting Class II NPF events may occur regionally and the argument has presented above.

Reference:

Jeong, C. H., Evans, G. J., McGuire, M. L., Chang, R. W., Abbatt, J. P. D., Zeromskiene, K., Mozurkewich, M., Li, S.-M. and Leaitch, W. R.: Particle formation and growth at five rural and urban sites. Atmos. Chem. Phys., 10(16), 7979-7995, 2010.

Kulmala, M., Vehkamäki, H., Petäjä, T., Dal Maso, M., Lauri, A., Kerminen, V. M., Birmilic, W., and McMurry, P. H.: Formation and growth rates of ultrafine atmospheric particles: a review of observations. J. Aerosol Sci., 35(2), 143-176, 2004.

Kulmala, M. and Kerminen, V. M.: On the formation and growth of atmospheric nanoparticles, Atmos. Res., 90, 132-150, 2008.

Kulmala, M., Petäjä, T., Nieminen, T., Sipilä, M., Manninen, H. E., Lehtipalo, K., Maso, M. D., Aalto, P. P., Junninen, H., Paasonen, P., Riipinen, I., Lehtinen, K. E. J., Laaksonen, A., and Kerminen, V-M.: Measurement of the nucleation of atmospheric aerosol particles, Nature Protocols., 7, 1651-1667, 2012.

Stanier, C. O., Khlystov, A. Y., and Pandis, S. N.: Nucleation events during the Pittsburgh Air Quality Study: description and relation to key meteorological, gas phase, and aerosol parameters special issue of aerosol science and technology on findings from **ACPD**
the fine particulate matter supersites program. Aerosol Sci. Technol., 38(S1), 253-264, 2004.

Wehner, B., Wiedensohler, A., Tuch, T. M., Wu, Z. J., Hu, M., Slanina, J., and Kiang, C. S.: Variability of the aerosol number size distribution in Beijing, China: New particle formation, dust storms, and high continental background. Geophys. Res. Let., 31, L22108, 2004.

Wildt, J., Mentel, T. F., Kiendler-Scharr, A., Hoffmann, T., Andres, S., Ehn, M., Kleist, E., Müsgen, P., Rohrer, F., Rudich, Y., Springer, M., Tillmann, R., and Wahner, A.: Suppression of new particle formation from monoterpene oxidation by NOx, Atmos. Chem. Phys., 14, 2789-2804, 2014.

Yao, X.H., Choi, M.Y., Lau, N.T., Lau, A. P. S., Chan, C. K., and Fang, M.: Growth and Shrinkage of New Particles in the Atmosphere in Hong Kong. Aerosol Sci. Technol., 44(8), 639–650, 2010.

Yao, X. H., Lee, C. J., Evans, G. J., Chu, A., Godri, K. J., McGuire, M. L., Ng, A. C., and Whitelaw, C.: Evaluation of ambient SO2 measurement methods at roadside sites. Atmos. Environ., 45(16), 2781-2788, 2011.
**ACPD**
Fig. 1. Simultaneous observed Class II NPF event and NPF event with extremely low growth rate at  ${\sim}240$  km distance.

---

## Author Comment (AC4) · 7 Apr 2017

This manuscript presents a field measurement of new particle formation (NPF) events in urban Beijing, China. The deployment of Fast Mobility Particle Sizers (FMPs) is unique and could deliver new insights into NPF, if interpreted properly. Overall, this manuscript describes interesting phenomena that NPF was enhanced in winter at a street site comparing to a close rooftop site, whereas NPF was less pronounced at the street site in spring. The explanation for these observation, unfortunately, is not well justified, and requires a major work over again. Here are my detailed comments.

Response: The authors thank the reviewer's comments and try our best to respond and revise our manuscript accordingly.

Main comments,

1. Micro-meteorology could be a major player that explains the difference between the street site and the rooftop site, which is not discussed at all in the current manuscript. Potentially, the loss of nanoparticles due to the surfaces along the street canyon is a factor too.

Response: The authors believe the turbulence dispersion to be more important than advection diffusion at the street site under a strong synoptic wind. The authors didn't measure wind direction and wind speed at multiple street locations and the authors didn't think one point observation of wind direction and wind speed can full reflect the complicated micro-scale wind field at the street site. The authors thereby reluctantly speculated the influence of the complicated micro-scale wind field on the observed particle number concentrations between the rooftop site and street site.

Through a deep data analysis, the authors additionally provided two types of unique evidences which were less affected by micro-meteorology at the street site as well as the simplest evidence, i.e., the simple comparison between the rooftop site and street sites, to confirm the reduced and enhanced NPF at the street site in different seasons. The authors found that the sufficiency and uniqueness of three types of evidences were not fully recognized by the reviewers because their challenges mainly focused on the simplest evidence. This means the authors' presentation strategy has to be improved to make the three types of evidences more obviously.

In revision, the authors will clarify that the reduced NPF always occurred at the street site in the springtime. Three evidences from different angles will be itemized as Evidence 1, 2 and 3, i.e., Evidence 1: The lower particle number concentration (PNC) of nucleation mode particles at the street site mainly because of a shorter initial burst time. Evidence 2: The authors used the PNC at the street site subtracting the corresponding PNC at the rooftop site to calculate the difference. The authors then obtained the second evidence: the negative difference of nucleation mode particles against the
positive difference of Aitken mode particles on NPF days. Evidence 3: Using the same approach, the authors obtained the third evidence: the negative difference of nucleation mode particles on NPF days against the positive difference of that on non-NPF days (Figs. 3 and 4 in the origin version).

In addition, the authors will clarify that the enhanced NPF always occurred at the street site in the wintertime. Three evidences from different angles will be itemized as Evidence 1, 2 and 3, i.e., Evidence 1: The significantly larger PNC of nucleation mode particles at the street site and a larger apparent formation rate of new particles mainly because of a shorter initial burst time. Evidence 2: The positive difference of nucleation mode particles in the wintertime against the negative difference of nucleation mode particles in the springtime on NPF days. Evidence 3: The larger positive difference of nucleation mode particles on NPF days against that on non-NPF days in the wintertime (Figs. 5 and 7 in the origin version).

We agree that the loss of nanoparticles due to the surfaces along the street canyon can be a potential factor of the reduced NPF in the springtime. In revision, We will revise L225 as: "Several factors can lead to the reduced NPF events at the street site, i.e.., 1) a larger condensation sink because of more pre-existing atmospheric particles from primary emissions; 2) tall buildings along both the sides of urban streets providing additional surface areas to scavenge gases and atmospheric particles (Yao et al., 2011); 3) vehicle-emitted NO reacting with RO2 and suppressing NPF (Wildt et al., 2014).

2. The inter-comparison between two FMPSs showed some differences, and the authors decided to use one FMPS as a reference and correct the number concentration of the other one. How did they decide which one is "the one" to trust? Nevertheless, number concentrations are used to calculate formation rates, growth rates, and condensation sink. This could lead to a major uncertainty in the discussion for nucleation mechanism.

Response: As reported by Zimmerman et al. (2015), an independent measurement

**ACPD**
of CPC simultaneously with FMPS can be used to accurately correct the FMPS data. The reference has been cited in the manuscript. In this study, a CPC was operating simultaneously with a FMPS at the street site and the FMPS was thereby used to correct the other. The two FMPS were then processed the second-step correction proposed by Zimmerman et al. (2015). The total particle number concentration of the FMPS at the rooftop site was multiplied by a correction factor which was equal to the minutely averaged ratios of the FMPS and CPC data at the street site. The calculated value and the CPC data was used to complete the third correction for the FMPS data at the rooftop site and at the street site, respectively. This will be clarified in revision.

3. The FMPSs were placed downstream of dryers, which indicates that the measured size distributions could be of from the atmospheric ones. This at least eliminates the role of relative humidity to a certain extent. Even for particles in the size ranges of 10-20 nm, the uptake of H2O is one of the major pathways for particle to grow.

Response: Relative humidity (RH) varied from 13% to 49% during NPF event days in the wintertime and below 55% during NPF event days in the springtime. At such RH levels, the growth factor of 10-20 nm particles (assume to be (NH4)2SO4) were less than 1.02 (Hämeri et al., 2000). This will be added in revision.

4. The mixing ratio of SO2 was only measured at the rooftop site. How about CO? It might be possible to deduce a street SO2 simply by the mixing ratios of CO. The current assumption that concentrations of SO2 are identical at the two sites are not acceptable, and could lead to mis-interpretation.

Response: The authors thank the comments and the SO2 concentration at the two sites indeed needs more interpretation.

In this study, CO was not measured at the street site. However, black carbon (BC) was measured by a potable aethalometer at the street site and BC is also a good indicator of traffic emission (Fruin, et al., 2004; Meng et al., 2015 a,b). The authors tried to use BC to deduce the traffic-related SO2 at the street site. In revision, we will add

**ACPD**
"The sulfur content in the gasoline and diesel was limited <50 ppm at those years. The measured BC spikes were lower than 5  $\mu$ g m-3 during the NPF periods. The maximum contribution of traffic-related SO2 at the street site was roughly estimated to be 1.3 ppb according to the results in our previous studies (Meng et al., 2015 a,b)". In the wintertime, the ratio of traffic-derived SO2 to the observed values was less than 1/4 and the observed values were overwhelmingly contributed by domestic heating. The uncertainty by assuming SO2 at the street site same as the rooftop site should be minor in the wintertime and it should not affect our conclusion because the formation rates of new particles at the street site were increased by 3-5 times against the rooftop site in the wintertime. In the springtime, the contribution of traffic-related SO2 might significantly increase the mixing ratio of SO2 at the street site. However, the reduced NPF was observed at the street site. The possible underestimation of SO2 at the street site likely existed and caused the reduced NPF.

5. The authors focused on the oxidation of biogenic organics when discussing the growth of >10 nm particles. In an urban environment such as Beijing, wouldn't anthropogenic VOCs be more concentrated? Are there any measurements that point the authors to biogenic VOCs instead of anthropogenic ones? How will the interpretation

Response: The authors thank the comments. The role of oxidation products of biogenic VOC in NPF events were widely studied in field experiments, chamber and modeling studies, and quantum chemical calculations (Schobesberger et al., 2013; Riccobono et al., 2014; Ortega et al., 2015; Tröstl et al., 2016). According to the established knowledge, the authors argued the potential importance of oxidized biogenic VOC in growing newly formed particles in this study. The north and northwest directions of the sampling site subject to mountain areas have a high percentage of land-covered forests. Extensive biogenic VOC is theoretically expected in spring may act as important precursor in NPF. During the NPF periods, the north or northwest wind dominated and carried less polluted or even clear ambient air from mountain areas to the sampling site, e.g., the
mixing ratio of SO2 was < 3 ppb in the springtime and < 5 ppb in the wintertime during the periods of NPF events.

The role of oxidized anthropogenic VOCs in growing >10 nm newly formed particles is still poorly understood (Zhang et al., 2009; Hoyle et al., 2011). Considering the knowledge gap and lack of related data in this study, the authors are reluctantly to discuss the possibility in this study. Following the reviewer's comments, in the context, the authors will add "Theoretically, oxidized anthropogenic VOCs could also participate in the growth of newly formed particles while the study is limited (Zhang et al., 2009; Hoyle et al., 2011). The role of oxidized anthropogenic VOCs needs further study."

6. Throughout the manuscript, the authors are presenting J8, which is fine. However, particles bigger than 8 nm are larger enough that they don't really reflect the nucleation mechanism, instead, a combination of nucleation and subsequent growth, especially growth mechanisms, might actually determines how many particles were measured.

Response: Different reviewers clearly had different views on the apparent formation rate of new particles. As a compromise by considering all reviewers comments, J8 will be used in revision. We will add additional interpretation: "Noted that J8 reflects a combination of nucleation and subsequent initial growth of new particles." in the explanation of new particle formation rate.

Minor comments,

7. (Page 8), clearly define long-term NPN, short-term NPF, Class I NPF, and class II NPF.

Response: The authors thank the comments. The authors would like to correct "longterm NPF events, short-term NPF events" to "regional NPF events, short-lived NPF events" following Stanier et al. (2010) and Jeong et al. (2010). Regional NPF events represent NPF events occurred lasting longer than 1 hour, and short-lived NPF events represent the NPF periods was less than 1 hour. When new particles showed "banana
shape" growth, and the new particles grow to larger than 30 nm, we define as Class I NPF event. When the new particles showed no apparent growth and the particles diameter was about 11 nm, we define as Class II NPF event. In this study, the duration of NPF events in Class I and Class II were lasting longer than 1 hour, which belong to the regional NPF events.

8. (Page 10, Line 225), how about NO in the winter? Wouldn't NO be always higher in the street canyon?

Response: Theoretically, NO could be always higher in the street canyon not only in the springtime but also in the wintertime by considering on-road traffic emissions closer to the street site. The enhanced NPF at the street site implied that the enhanced effects overwhelming the reduced effects. A revision will be done to reflect this argument.

9. (Page 13, Lines 280-287), the argument on H2SO4 is just speculation. Many factors determines H2SO4. Also, why is SO2 from on-road vehicles negligible comparing to the background?

Response: In this study, the NPF events occurred under the north or northwest wind direction with wind speed >4m/s. The north or northwest wind carried less polluted or even clear air to the sampling site during the NPF event periods, e.g., the mixing ratio of SO2 was

10. (Page 15, the last paragraph), I am certainly not convinced by the discussion here. Different nucleation mechanisms probably explains NPF events in Beijing, Qingdao, and marginal seas of China. Again, J8 is not a good indicator for nucleation mechanisms. By definition, NMIoNP stand for "the net maximum increase of nucleation mode PNC". I don't see a clear connection between NMIoNP and J8. A cutoff of 8 cm-3 s-1 could be arbitrary. The correlation will not be bad if a cutoff of , say, 7 cm-3 s-1, was chosen.

Response: Theoretically, the formation rate of new particles alone has no direct relationship with the environmental and climate effects of new particles. However, NMIoNP defined as "the net maximum increase of nucleation mode PNC" is directly related to the environmental and climate effects of new particles. Thus, it is important to establish the link between NMIoNP and J8.

Considered 1) formation rate of new particles, e.g., J= kNucOrg[H2SO4]m[NucOrg]n (Zhang et al., 2012), 2) the subsequent particle growth, and 3) H2SO4 vapor to be necessary for nucleation in ambient air except at sea beach, two scenarios are considered. One is: H2SO4 vapor is relatively sufficient against NucOrg and J8 is thereby mainly determined by availableness of NucOrg. A good correlation is theoretically expected for J8 and NMIoNP. The other scenario is: NucOrg is relatively sufficient against H2SO4 vapor and J8 is thereby mainly determined by availableness of H2SO4 vapor. J8 could be high, but the total yield of new particles could be low because of a rapid consumption of H2SO4 vapor. A poor or no correlation is theoretically expected for J8 and NMIoNP.

The authors found when the formation rate less than 8 cm-3 s-1, the new particle yield increased with the increasing formation rate. Of course, for any values

the following session?

Response: The authors revise the description of 25% minimum CV as: "Coefficient of variation (CV) is the ratio of standard deviation to mean for particle number concentration in every 60s. We sort the data sequence in an order from the smallest to largest, and the minimum 25% values of CV reflect smaller changes of particle number concentration."

N16.5nm used as the reference to conduct the deduction relied on two reasons: 1) Zhu et al. (2006) also observed the vehicle particle mode around 16 nm at 30 m downwind. 2) In this study, 10 nm-30 nm was the major mode in the vehicle particle size distribution, and 16.5 nm was the peak of this particle mode. This will be added in revision.

12. Proofread the manuscript.

Response: The authors are sorry for this and the language-editing will be processed before re-submitting.

Reference:

Fruin, S. A., Winer, A. M., and Rodes, C. E.: Black carbon concentrations in California vehicles and estimation of in-vehicle diesel exhaust particulate matter exposures. Atmos. Environ., 38(25), 4123-4133, 2004.

Hämeri, K., Väkevä, M., Hansson, H. C., and Laaksonen, A.: Hygroscopic growth of ultrafine ammonium sulphate aerosol measured using an ultrafine tandem differential mobility analyzer. J. Geophys. Res., 105(D17), 22231-22242, 2000.

Hoyle, C. R., Boy, M., Donahue, N. M., Fry, J. L., Glasius, M., Guenther, A. Hallar, A. G., Huff Hartz, K., Petters, M. D., Petäjä, T., Rosenoern, T., and Sullivan, A. P.: A review of the anthropogenic influence on biogenic secondary organic aerosol. Atmos. Chem. Phys., 11(1), 321-343, 2011.

ACPD
Jeong, C. H., Evans, G. J., McGuire, M. L., Chang, R. W., Abbatt, J. P. D., Zeromskiene, K., Mozurkewich, M., Li, S.-M. and Leaitch, W. R.: Particle formation and growth at five rural and urban sites. Atmos. Chem. Phys., 10(16), 7979-7995, 2010.

Meng, H., Zhu Y. J., Evans G., and Yao X. H.: An approach to investigate new particle formation in the vertical direction on the basis of high time-resolution measurements at ground level and sea level. Atmos. Environ., 102, 366-375, 2015a.

Meng, H., Zhu, Y. J., Evans, G., Jeong, C. H., and Yao, X. H.: Roles of SO2 oxidation in new particle formation events. J. Environ. Sci., 30, 90-101, 2015b.

Ortega, I. K., Donahue, N. M., KurtelAn, T., Kulmala, M., Focsa, C., and VehkamalLki, H.: Can Highly Oxidized Organics Contribute to Atmospheric New Particle Formation? J. Phys. Chem. A, 120(9), 1452-1458, 2015.

Riccobono, F., Schobesberger, S., Scott, C. E., Dommen, J., Ortega, I. K., Rondo, L., Almeida, J., Amorim, A., Bianchi, F., Breitenlechner, M., David, A., Downard, A., Dunne, E. M., Duplissy, J., Ehrhart, S., Flagan, R. C., Franchin, A., Hansel, A., Junninen, H., Kajos, M., Keskinen, H., Kupc, A., Kürten, A., Kvashin, A. N., Laaksonen, A., Lehtipalo, K., Makhmutov, V., Mathot, S., Nieminen, T., Onnela, A., Petäjä, T., Praplan, A. P., Santos, F. D., Schallhart, S., Seinfeld, J. H., Sipilä, M., Spracklen, D. V., Stozhkov, Y., Stratmann, F., Tomé, A., Tsagkogeorgas, G., Vaattovaara, P., Viisanen, Y., Vrtala, A., Wagner, P, E., Weingartner, E., Wex, H., Wimmer, D., Carslaw, K, S., Curtius, J., Donahue, N. M., Kirkby, J., Kulmala, M., Worsnop, D. R., and Baltensperger, U.: Oxidation products of biogenic emissions contribute to nucleation of atmospheric particles, Science, 344, 717-721, 2014.

Schobesberger, S., Junninen, H., Bianchi, F., Lönn, G., Ehn, M., Lehtipalo, K., Dommen, J., Ehrhart, S., Ortega, I. K., Franchin, A., Nieminen, T., Riccobono, F., Hutterli, M., Duplissy, J., Almeida, J., Amorim, A., Breitenlechner, M., Downard, A. J., Dunne, E. M., Flagan, R. C., Kajos, M., Keskinen, H., Kirkby, J., Kupc, A., Kürten, A., Kurtén, T., Laaksonen, A., Mathot, S., Onnela, A., Praplan, A. P., Rondo, L., Santos, F. D., Interactive comment

Schallhart, S., Schnitzhofer, R., Sipilä, M., Tomé, A., Tsagkogeorgas, G., Vehkamäki, H., Wimmer, D., Baltensperger, U., Carslaw, K. S., Curtius, J., Hansel, A., Petäjä, T., Kulmala, M., Donahue, N, M., and Worsnop, D. R.: Molecular understanding of atmospheric particle formation from sulfuric acid and large oxidized organic molecules. Proc. Natl. Acad. Sci., 110(43), 17223-17228, 2013.

Tröstl, J., Chuang, W. K., Gordon, H., Heinritzi, M., Yan, C., Molteni, U., Ahlm, L., Frege, C., Bianchi, F., Wagner, R., Simon, M., Lehtipalo, K., Williamson, C., Craven, J. S., Duplissy, J., Adamov, A., Almeida, J., Bernhammer, J. A. K., Breitenlechner, M., Brilke, S., Ehrhart, S., Dias, A., Flagan, R. C., Franchin, A., Fuchs, C., Guida, R., Gysel, M., Hansel, A., Hoyle, C. R., Jokinen, T., Junninen, H., Kangasluoma, J., Keskinen, H., Kim, J., Krapf, M., Kürten, A., Laaksonen, A., Lawler, M., Leiminger, M., Mathot, S., Möhler, O., Nieminen, T., Onnela, A., Petäjä, T., Piel, F. M., Miettinen, P., Rissanen, M. P., Rondo, L., Sarnela, N., Schobesberger, S., Sengupta, K., Sipilä, M., Smith, J. N., Steiner, G., Tomè, A., Virtanen, A., Wagner, A. C., Weingartner, E., Wimmer, D., Winkler, P. M., Ye, P. L., Carslaw, K. S., Curtius, J., Dommen, J., Kirkby, J., Kulmala, M., Riipinen, I., Worsnop, D. R., Donahue, N. M., and Baltensperger, U.: The role of low-volatility organic compounds in initial particle growth in the atmosphere, Nature, 533, 527-531, 2016.

Stanier, C. O., Khlystov, A. Y., and Pandis, S. N.: Nucleation events during the Pittsburgh Air Quality Study: description and relation to key meteorological, gas phase, and aerosol parameters special issue of aerosol science and technology on findings from the fine particulate matter supersites program. Aerosol Sci. Technol., 38(S1), 253-264, 2004.

Wildt, J., Mentel, T. F., Kiendler-Scharr, A., Hoffmann, T., Andres, S., Ehn, M., Kleist, E., Müsgen, P., Rohrer, F., Rudich, Y., Springer, M., Tillmann, R., and Wahner, A.: Suppression of new particle formation from monoterpene oxidation by NOx, Atmos. Chem. Phys., 14, 2789-2804, 2014.
Yao, X. H., Lee, C. J., Evans, G. J., Chu, A., Godri, K. J., McGuire, M. L., Ng, A. C., and Whitelaw, C.: Evaluation of ambient SO2 measurement methods at roadside sites. Atmos. Environ. 45(16), 2781-2788, 2011.

Zhang, R., Wang, L., Khalizov, A. F., Zhao, J., Zheng, J., McGraw, R. L., and Molina, L. T.: Formation of nanoparticles of blue haze enhanced by anthropogenic pollution. Proc. Natl. Acad. Sci., 106(42), 17650-17654, 2009.

Zimmerman, N., Jeong, C. H., Wang, J. M., Ramos, M., Wallace, J. S., and Evans, G. J.: A source-independent empirical correction procedure for the fast mobility and engine exhaust particle sizers, Atmos. Environ., 100, 178-184, 2015.

Zhang, R., Khalizov, A., Wang, L., Hu, M., and Xu, W.: Nucleation and growth of nanoparticles in the atmosphere, Chem. Rev., 112, 1957-2011, 2012.

---

## Author Comment (AC5) · 8 Apr 2017

Fast-response measurements of particle number size distributions of aerosol $\geq 8$ nm diameter have been made at a street canyon and nearby rooftop site. The authors selectively report specific days of data from a small dataset, and draw many tentative conclusions concerning mechanisms of new particle formation (NPF) which are difficult to justify given the small dataset and the extent to which it is over-interpreted. The introduction quite reasonably states that "it is critical to evaluate the effects of nucleating species other than sulfuric acid and the dependence of NPF on pre-existing particles in the atmosphere". This is an excellent objective but unfortunately the paper does nothing to answer the question about other nucleating species, and does not event provide

clear answers concerning the role of sulfuric acid.

Response: The authors didn't selectively present the data. All simultaneous measurements on particle number concentrations during the springtime and wintertime campaigns have been included in this study. At the street site, the reduced NPF events always occurred in the springtime and the enhanced NPF events always occurred in the wintertime. This will be highlighted in revision.

The authors would like to believe the sufficiency and uniqueness of evidences are critical to evaluate the quality of scientific studies. This is because the number of cases either for gravitational wave observation in 2016 or a recent NPF study reported by Bianchi et al. (2016) was even smaller than those presented in this study.

This study is definitely not a first study for NPF events and all analyses build on previously well-established knowledge, particularly the progressing in the last few years. The authors did provide unique evidences to analyze the effects of nucleating species other than sulfuric acid at street site. The authors also agree that more analyses should be added so that they can be easily understood. For example, in revision, the authors will add "Considered 1) formation rate of new particles, e.g., $J=kNucOrg[H2SO4]m[NucOrg]n$ (Zhang et al., 2012), 2) the subsequent particle growth, and 3) H2SO4 vapor to be necessary for nucleation in ambient air except at sea beach, two scenarios are considered. One is: H2SO4 vapor is relatively sufficient against NucOrg and J8 is thereby mainly determined by availableness of NucOrg. A good correlation is theoretically expected for J8 and NMIoNP. The other scenario is: NucOrg is relatively sufficient against H2SO4 vapor and J8 is thereby mainly determined by availableness of H2SO4 vapor. J8 could be high, but the total yield of new particles could be low because of a rapid consumption of H2SO4 vapor. A poor or no correlation is theoretically expected for J8 and NMIoNP."

One of the key elements towards interpretation of this dataset in relation to nucleation and growth is the role of sulfuric acid, which ideally would have been measured. How-

ever, as measurements were not available, an old parameterisation is used to estimate H2SO4 vapour concentrations in which the H2SO4 formation rate is described by the product of SO2 concentration and global solar radiation. This may be adequate for situations in the background troposphere where ozone photolysis is the predominant source of hydroxyl radical, but many studies have now shown that in polluted atmospheres such as Beijing, other processes such as photolysis of HONO and HCHO, and ozone-alkene reactions are far more important sources of hydroxyl, and equation 4 is unlikely to be a reliable means of calculation of [H2SO4].

Response: The authors would like to believe that the reviewer may capture a piece of information about air pollution in Beijing, but it is hard to say the piece reflecting the full picture.

During the periods of NPF events in this study, the air mass at the sampling site was less polluted or even clear. The NPF events occurred under the north or northwest wind direction with wind speed >4m/s. The north and northwest directions of the sampling site subject to mountain areas have a high percentage of land-covered forests. The north or northwest wind carried less polluted or even clear ambient air to the sampling site during the NPF event periods, e.g., the mixing ratio of SO2 was < 3 ppb in the spring and < 5 ppb in the winter during the periods of NPF events. Less polluted or even clear air exactly meets equation 4 in the manuscript. The authors cannot find what problem in our approach is.

The differences in behaviour between the sites are interesting, and if correctly interpreted could give useful insights into NPF in polluted atmospheres. However no measurements were made of potentially condensing species, or their precursors other than SO2, and the latter was measured at only one site with the unproven assumption that concentrations of SO2 were the same at both sites. Much is made of the rates of change of particle number concentrations, but the effects of wind direction changes upon concentrations in the street canyon (which can be large) do not appear to have been considered. The methods used for subtraction of fresh traffic emissions are highly

questionable, and no use is made of gaseous pollutant data (e.g. NOx) which would be a strong covariate of PNC from road vehicles.

Response: The authors would like to believe that the reviewer may capture a piece of information about air pollution in Beijing, but it is hard to say the piece reflecting the full picture. The technical term "polluted atmospheres" is probably not applicable for NPF events reported in this study.

The authors had no simultaneous measurements of other condensing species, such as H2SO4, HOM, etc., and had no way to discuss them. The authors agree that the SO2 concentration at the two sites indeed needs more interpretation. In revision, we will add "The sulfur content in the gasoline and diesel was limited <50 ppm at that years. The measured BC spikes were lower than 5 $\mu$g m-3 during the NPF periods. The maximum contribution of traffic-related SO2 at the street site was roughly estimated to be 1.3 ppb according to the results in our previous studies (Meng et al., 2015 a,b)". In the wintertime, the ratio of traffic-derived SO2 to the observed values was less than 1/4 and the observed values were overwhelmingly contributed by domestic heating. The uncertainty by assuming SO2 at the street site same as the rooftop site should be minor in the wintertime and it should not affect our conclusion because the formation rates of new particles at the street site were increased by 3-5 times against the rooftop site in the wintertime. In the springtime, the contribution of traffic-related SO2 might significantly increase the mixing ratio of SO2 at the street site. However, the reduced NPF was observed at the street site. The possible underestimation of SO2 at the street site further solidified our analysis results, i.e., a strong scavenge effect at the street site likely existed and caused the reduced NPF.

The authors believe the turbulence dispersion to be more important than advection diffusion at the street site under a strong synoptic wind. The authors didn't measure wind direction and wind speed at multiple street locations and the authors didn't think one point observation of wind direction and wind speed can full reflect the complicated micro-scale wind field at the street site. The authors thereby are reluctantly speculated

the influence of the complicated micro-scale wind field on the observed particle number concentrations between the rooftop site and street site.

Through a deep data analysis, the authors additionally provided two types of unique evidences which were less affected by micro-meteorology at the street site as well as the simplest evidence, i.e., the simple comparison between the rooftop site and street sites, to confirm the reduced and enhanced NPF at the street site in different seasons. The authors found that the sufficiency and uniqueness of three types of evidences were not fully recognized by the reviewers because their challenges mainly focused on the simplest evidence. This means the authors' presentation strategy has to be improved to make the three types of evidences more obviously.

In revision, the authors will clarify that the reduced NPF always occurred at the street site in the springtime. Three evidences from different angles will be itemized as Evidence 1, 2 and 3, i.e., Evidence 1: The lower particle number concentration (PNC) of nucleation mode particles at the street site mainly because of a shorter initial burst time. Evidence 2: The authors used the PNC at the street site subtracting the corresponding PNC at the rooftop site to calculate the difference. The authors then obtained the second evidence: the negative difference of nucleation mode particles against the positive difference of Aitken mode particles on NPF days. Evidence 3: Using the same approach, the authors obtained the third evidence: the negative difference of nucleation mode particles on NPF days against the positive difference of that on non-NPF days (Figs. 3 and 4 in the origin version).

In addition, the authors will clarify that the enhanced NPF always occurred at the street site in the wintertime. Three evidences from different angles will be itemized as Evidence 1, 2 and 3, i.e., Evidence 1: The significantly larger PNC of nucleation mode particles at the street site and a larger apparent formation rate of new particles mainly because of a shorter initial burst time. Evidence 2: The positive difference of nucleation mode particles in the wintertime against the negative difference of nucleation mode particles in the springtime on NPF days. Evidence 3: The larger positive difference of nucleation mode particles on NPF days against that on non-NPF days in the wintertime (Figs. 5 and 7 in the origin version).

In this study, NOx was not measured at the street site. However, black carbon (BC) was measured by a potable aethalometer at the street site and BC is also a good indicator of traffic emission (Fruin, et al., 2004; Meng et al., 2015 a,b). The authors tried to use BC as an indicator of vehicle emission plumes to deduct primary traffic particles. It does not work because the one-minute time resolution is too low to successfully deduct primary traffic particles. To best of our knowledge, NOx analyzers are usually set for operating in one-minute time resolution and the data of NOx may suffer from the same problem. An example is presented to illustrate the problem for using BC to deduct primary traffic particles.

During the entire sampling period on 22 December 2010, BC shows no correlation with the nucleation mode PNC (shown in Fig. 1a). During a few short periods, the BC spikes appeared to be visibly consistent with the PNC spikes as shown in Fig. 1b. However, the correlation obtained was much poor, e.g., during the period of 10:30-12:30 (shown in Fig. 1c). This is not surprised because the aethalometer reported the instantaneous value of BC in one minute, but the vehicle spikes physically occurred in a few seconds (shown in Fig. 2). Under such poor correlation, the regression equation is invalid to accurately deduct primary traffic particles.

The points above justify a major reappraisal of the data, and the development of far less ambitious conclusions.

Response: The authors will try the best to revise to improve the manuscript. The authors have to say that "polluted atmospheres" in Beijing much claimed by the reviewer may be not closely related to NPF events reported in this study. Polluted atmospheres in Beijing indeed take place in presence of stagnant metrological conditions or under a dominant south or southwest wind.

Other points which need to be addressed include:

(a) The introduction lists a number of organic acids as examples of vehicle-emitted organic compounds. Most of these have far more major secondary sources, or are present in cooking emissions, with little if any arising from road traffic.

Response: Agree, the authors will revise the part as: "Urban street canyons provide semi-enclosed environments trapping vehicle exhausts that contain aromatic and aliphatic hydrocarbons, SO2, NOx, amines, black carbon, etc. (Pierson et al., 1983; Stemmler et al., 2005; Burgard et al., 2006; Liu et al., 2008; Buccolieri et al., 2009; Sun et al., 2012; Gentner, et al., 2012)"

(b) Some ill-informed statements are made about the (currently uncertain) effects of exposure to ultrafine particles. These particles do not lead to "destruction of the respiratory system" and the statement that "newly formed particles inside a street canyon may become toxic when vehicle-release organics is involved in the nucleation process" is not supported by references.

Response: The authors will revise as: "Ultrafine particles (<100 nm) have been reported to have adverse human health impacts through the deposition in the pulmonary region and penetration into the bloodstream (Oberdörster et al., 2004; Schlesinger et al., 2006; Zhang et al., 2012, 2015)." "In addition, the newly formed particles inside a street canyon may become toxic when vehicle-released organics is involved in the nucleation process (Sgro et al., 2009; Gualtieri et al., 2014)."

(c) There is no information on quality assurance beyond an intercomparison between the two FMPS, and no consideration of how size-dependent particle losses in the inlet system affect measured size distributions.

Response: As reported by Zimmerman et al. (2015), an independent measurement of CPC simultaneously with FMPS can be used to accurately correct the FMPS data including the size-dependent particle loss. The reference has been cited in the manuscript. In this study, a CPC was operating simultaneously with a FMPS at the street site and the FMPS was thereby used to correct the other. The two FMPS were

then processed the second-step correction proposed by Zimmerman et al. (2015). The total particle number concentration of the FMPS at the rooftop site was multiplied by a correction factor which was equal to the minutely averaged ratios of the FMPS and CPC data at the street site. The calculated value and the CPC data was used to complete the third correction for the FMPS data at the rooftop site and at the street site, respectively. This will be clarified in revision.

The particle loss for >8 nm particles were undetectable in 2.8 m sampling lines. This will also be added in the revision.

(d) Equation (3) differs from that in the nucleation protocol paper of Kulmala et al. (2012) by a factor of two, which needs to be explained.

Response: The equation (3) in this study was exactly same as reported by Dal Maso et al. (2005) and Kulmala et al. (2005). The references have been cited in this study. The authors don't understand the equation presented by Kulmala et al. (2012) and prefer to use the equation widely adopted in literature.

(e) A clear definition is needed for the "maximum increase of nucleation mode PNC (NMIoNP)" which is much used in the data analyses.

Response: "the net maximum increase of nucleation mode PNC (NMIoNP)" is calculated as "N8-20nm(t1)-N8-20nm(t0)". t0 is set as the time when the apparent NPF started to be observed and t1 as the time when the nucleation mode PNC reaches the maximum value. This will be added in revision.

(f) The authors should establish that their Class II particles arise from an NPF event, rather than an emission source.

Response: In revision, the authors will reorganize the evidences to confirm Class II particles to be a regional NPF event, i.e.,

1) As reviewed by Vu et al. (2015), the particle number size distribution (PNSD) of emission source (e.g., traffic emissions, industrial emissions, biomass burning, cooking) character the typical peak number mode, such as at 30 nm, 50 nm, 70-80 nm, 120-140 nm, et al. When the NPF events in Class II occurred in our study, the nucleation mode particles overwhelmed and other particle modes were negligible.

2) Similar to Class II NPF events with the particle growth to be undetectable presented in this study, extremely low growth rate of newly formed particles ($\sim$ 1 nm h-1) in Beijing was also previously reported by Wehner et al. (2004). In our unpublished data, the authors simultaneously observed Class II NPF events and NPF events with extremely low growth rate at $\sim$240 km distance (shown in Fig. 3, the case will also be presented in Supplementary). In the last three years (data unpublished), we had simultaneous observations of NPF events at 100-500 km distance. The authors obtained six cases based on simultaneous observations at two locations, i.e., one case featured by Class II NPF vs Class II NPF, four cases featured by Class II NPF vs NPF with an extremely low growth rate, one case featured by Class II NPF vs NPF with "banana shape" particle growth.

3) In this study, the duration period of Class II events lasted for 4-8 hours with the wind speed >4m/s. The authors strongly believed that they should be considered as regional NPF events.

Reference:

Bianchi, F., Tröstl, J., Junninen, H., Frege, C., Henne, S., Hoyle, C. R., Molteni, U., Herrmann, E., Adamov, A., Bukowiecki, N., Chen, X., Duplissy, J., Gysel, M, Hutterli, M., Kangasluoma, J., Kontkanen, J., Kürten, A., Manninen, H. E., Münch, S., Peräkylä, O., Petäjä, T., Rondo, L., Williamson, C., Weingartner, E., Curtius, J., Worsnop, D. R., Kulmala, M., Dommen, J., and Baltensperger, U.: New particle formation in the free troposphere: A question of chemistry and timing. Science, 352(6289), 1109-1112, 2016.

[revised manuscript text omitted]

––––––––––––––––––––––––––––––

[Figure]

**Fig. 1.** The nucleation mode PNC and BC on 22 December, 2010. (a, b: time series of nucleation mode PNC and BC; c: relationship between nucleation mode PNC and BC during 10:30-12:30)

[Figure]

**Fig. 2.** Raw FMPS data showing vehicle spikes.

[Figure]

**Fig. 3.** Simultaneous observed Class II NPF event and NPF event with extremely low growth rate at ∼240 km distance.

---

## Author Response (AR1)

*This manuscript investigates new particle formation (NPF) observed simultaneously at two sites in a polluted urban environment. The analysis is based high-time resolution measurements, which increases the originality of the results. The background for this study (section 1) as well as the used methods (section 2) are very well written. Contrary to this, there are serious problems in how many of the results, have been interpreted. As a result, a large part of section 3 needs substantial revisions, and most of the sections 3.4-3.6 need to be entirely re-written. My detailed comments in this regard are given below.*

**Response:** The authors thank the reviewer's comments and try our best to respond and revise our manuscript accordingly.

*Major comments:*
*The authors provide two very general statements based on their results: 1) reduced NPF at street site compared to rooftop during spring, and 2) enhanced NPF at street site compared with rooftop in winter. These finding are supported by only 1-2 cases (days) of observations, which is way too little to make this kind of a general conclusion.*

**Response:** In revision, the authors will add "At the street site, the reduced NPF events always occurred in the springtime while the enhanced NPF events always occurred in the wintertime."

The authors would like to believe the sufficiency and uniqueness of evidences are crucial to evaluate the quality of scientific studies. This is because the number of cases for gravitational wave observation in 2016 and a recent NPF study reported by Bianchi et al. (2016) was even smaller than those presented in this study. The authors thereby abide by a principle, i.e., it is theoretically reasonable, multiple-evidences supported and no exception against it, to justify our results on reduced NPF at the street site, i.e., 1) Considered the widely recognized the importance of condensation sink in new particle formation (NPF), reduced NPF at the street site is theoretically expected and repeatedly occurred in the springtime. The authors provided three types of evidences from different angles to confirm the reduced NPF rather than simple comparison between rooftop site and street site measurements, i.e., Evidence 1: The lower particle number concentration (PNC) of nucleation mode particles at the street site mainly because of a shorter initial burst time. Evidence 2: The authors used the PNC at the street site subtracting the corresponding PNC at the rooftop site to calculate the difference. The authors then obtained the second evidence: the negative difference of nucleation mode particles against the positive difference of Aitken mode particles on NPF days. Evidence 3: Using the same approach, the authors obtained the third evidence: the negative difference of nucleation mode particles on NPF days against the positive difference of that on non-NPF days (Figs. 3 and 4 in the origin

version).

In addition, the authors also provided three types of evidences from different angles, rather than simple comparison between rooftop site and street site to confirm the enhanced NPF in the wintertime, i.e., Evidence 1: The significantly larger PNC of nucleation mode particles at the street site and a larger apparent formation rate of new particles mainly because of a shorter initial burst time. Evidence 2: The positive difference of nucleation mode particles in the wintertime against the negative difference of nucleation mode particles in the springtime on NPF days. Evidence 3: The larger positive difference of nucleation mode particles on NPF days against that on non-NPF days in the wintertime (Figs. 5 and 7 in the origin version).

According to the comments, the authors will revise the manuscript to make the unique evidences to be more obvious.

*The used proxy for gaseous sulfuric acid (SA) concentration has two problems: 1) it has been developed and evaluated for moderately-polluted sites only, so its applicability in highly-polluted sites like this one may be questionable.*

**Response:** In this study, the NPF events occurred under the north or northwest wind direction with wind speed >4m/s. The north or northwest wind carried less polluted or even clear ambient air to the sampling site during the NPF periods, e.g., the mixing ratio of $SO_2$ was < 3 ppb in the springtime and < 5 ppb in the wintertime during the periods of NPF events. Less polluted or even clear ambient air exactly meets the reviewer claimed, i.e., the proxy for calculating gaseous sulfuric acid (SA) concentration is applicable only under clean to moderately-polluted atmospheres. The authors thereby believe that our approach is consistent with the well-established knowledge and is thereby scientifically valid.

*2) SO2 is measured at rooftop site only, so it is unclear how well this represents SO2 in the street site. Also, the ratio in the SO2 concentration between the street site and rooftop is likely to be different between spring and winter, and there is no means to estimate this difference. As a result, the authors need to be very careful when making any interpretations that rely on estimated SA concentrations.*

**Response:** The authors thank the comments. In revised manuscript and supplementary, the authors will add: "The sulfur content in the gasoline and diesel was limited <50 ppm at those years. The measured BC spikes were lower than 5 $\mu g\ m^{-3}$ during the NPF periods. The maximum contribution of traffic-related $SO_2$ at the street site was roughly estimated to be 1.3 ppb according to the results in our previous studies (Meng et al., 2015 a,b). In the wintertime, the ratio of traffic-derived $SO_2$ to the observed values was less than 1/4 and the observed values were overwhelmingly contributed by domestic heating. The uncertainty by assuming $SO_2$ at the street site same as the rooftop site should be minor in the wintertime and it should not affect our conclusion

because the apparent formation rates of new particles at the street site were increased by 3-5 times against the rooftop site in the wintertime. In the springtime, the contribution of traffic-related $SO_2$ might significantly increase the mixing ratio of $SO_2$ at the street site. However, the reduced NPF was observed at the street site. The possible underestimation of $SO_2$ at the street site further solidified our analysis results, i.e., a strong scavenge effect at the street site likely existed and caused the reduced NPF."

*Class II NPF events have very low particle growth rates above 10 nm. All theoretical arguments indicate that >10 nm particles grow faster than smaller particles, and practically all observations on size-resolved particle growth rates support this view. This lead to a serious question: what is the origin of these particles? More specifically, if there are little condensable vapours to growth >10 nm, there should be even less vapors to grow smaller particles. One possible explanation for this is that particle of Class II originate from very local NPF, in which high local vapor concentrations initial nucleation and make the formed particles to grow very rapidly to a few nm, even to 10-20 nm. This rapid growth is then stopped due to atmospheric dilution of emitted vapors. This kind of process has been reported to occur in some coastal areas (Mace Head), in car exhaust to ambient air, and also close to other localised combustion sources. If Case II event are caused by very localized sources, it is questionable to compare NPF between the street site and rooftop in such cases.*

**Response:** The reviewer's first statement is probably contradictory to the truth. For example, the data results in Table 1 recently reported by Yu et al. (2016) fight against the reviewer's first statement. In the first publication on NPF events in Beijing (Wehner et al., 2004), the observed growth rate of new particles was as low as ~1 nm $h^{-1}$. In the study, however, the formation of new particles started around 07:00 after sunrise and the initial size of newly formed particles was ~5 nm. The results also indirectly fight against the reviewer's statement.

Theoretically, when the volume concentration of particles is considered, the amount of chemical species required for growing >10 nm particles was much larger than that for <10 nm particles. For example, the amount of chemicals required for growing particles from 10 nm to 12 nm was about six times larger than particles grew from 3 nm to 5 nm. Furthermore, the coagulation growth is important for <10 nm ambient particles while it is negligible for >10 nm ambient particles.

As reviewed by Vu et al. (2015), the particle number size distribution (PNSD) of vehicle or combustion plumes character the typical peak number mode such as at 30 nm, 50 nm, 70-80 nm, etc. In our study, when the NPF events in Class II occurred, the nucleation mode particles overwhelmed and other particle modes were negligible. The duration period of Class II lasted for 4-8 hours with the wind speed >4m/s, suggesting they probably happened in regional scale.

The authors have no idea to link NPF in the urban atmosphere of Beijing (an inland megacity where ocean-derived reactive iodides were unexpected) with those in rural coastal atmospheres, e.g., Mace Head where ocean-derived reactive iodides could be important precursors for NPF events. The authors may have no comments on the reviewer's speculation.

*The authors use condensation sink (CS) in interpreting their results. This problematic. The particles are formed below 2 nm size (J<2), but the authors calculate the formation rate of 8 nm particles (J8). The value of J8 depends on 3 quantities. J<2, CS and the growth rate of particles below 8 nm. Since neither J<2 nor the sub-8 nm growth rate are known, it is impossible to infer how CS might affect J8 in the observed cases.*

**Response:** The authors fully respect the reviewer's knowledge on the issue. However, the condensation sink has been widely used to argue the occurrence of NPF in literature when neither J<2 nor the sub-8 nm growth rate were not available, e.g., Kulmala et al. (2004, 2016).

*The authors assumed that only biogenic organics could influence NPF and subsequent growth. Why? There certainly large anthropogenic emissions of organic vapours in this kind of environment, and the oxidation of such vapors is very likely to produce low-volatile compounds that could affect nanoparticle formation and growth.*

**Response:** The authors never assumed "only biogenic organics could influence NPF and subsequent growth" in the manuscript. The role of oxidation products of biogenic VOC in NPF events have been widely studied in field experiments, chamber and modeling studies, and quantum chemical calculations (Schobesberger et al., 2013; Riccobono et al., 2014; Ortega et al., 2015; Tröstl et al., 2016). According to the established knowledge, the authors argued the potential importance of oxidized biogenic VOC in growing newly formed particles in this study. The north and northwest directions of the sampling site subject to mountain areas have a high percentage of land-covered forests. Extensive biogenic VOC is theoretically expected in spring and may act as important precursors in NPF. During the NPF periods, the north or northwest wind dominated and carried less polluted or even clear ambient air from mountain areas to the sampling site, e.g., the mixing ratio of $SO_2$ was < 3 ppb in the springtime and < 5 ppb in the wintertime during the periods of NPF events.

The role of oxidized anthropogenic VOCs in growing >10 nm newly formed particles is still poorly understood (Zhang et al., 2009; Hoyle et al., 2011). Considering the knowledge gap and lack of related data in this study, the authors are reluctantly to discuss the possibility in this study. Following the reviewer's comments, in the context, the authors will add "Oxidized anthropogenic VOCs could theoretically participate in the growth of newly formed particles while the study is limited (Zhang et al., 2009; Hoyle et al., 2011), but the role of oxidized anthropogenic VOCs needs

further study."

*Considering the points highlighted above, many of the interpretations made in sections 3.4-3.6 are not justified. The most problematic of these is section 3.6 which is highly speculative.*

**Response:** The comments are general and don't contain helpful information for revision. The authors thank the reviewer's comments and try our best to respond and revise our manuscript accordingly.

**Minor comments**
*I would recommend using terms other than short-term and long-term NPF events. In atmospheric time series, long-term usually means something that last for years or at least for months.*

**Response:** Agree. It will be revised as "short-lived NPF events, regional NPF events" (Stanier et al., 2010; Jeong et al., 2010).

*line 208: should be written: . . .only lasted for few minutes*

**Response:** It will be corrected in the revision.

*lines 254 and 270: did't detail is a strange expression. Please modify*

**Response:** The sentences are indeed ambiguous and unnecessary. Therefore, it will be deleted in revision.

*line 320: what is meant by ..reaction should proceed to solid state*

**Response:** It will be revised as "reaction should proceed to solid state, i.e., the gases start to partition on the particle phase."

**References:**
Bianchi, F., Tröstl, J., Junninen, H., Frege, C., Henne, S., Hoyle, C. R., Molteni, U., Herrmann, E., Adamov, A., Bukowiecki, N., Chen, X., Duplissy, J., Gysel, M, Hutterli, M., Kangasluoma, J., Kontkanen, J., Kürten, A., Manninen, H. E., Münch, S., Peräkylä, O., Petäjä, T., Rondo, L., Williamson, C., Weingartner, E., Curtius, J., Worsnop, D. R., Kulmala, M., Dommen, J., and Baltensperger, U.: New particle formation in the free troposphere: A question of chemistry and timing. Science, 352(6289), 1109-1112, 2016.

Hoyle, C. R., Boy, M., Donahue, N. M., Fry, J. L., Glasius, M., Guenther, A. Hallar, A. G., Huff Hartz, K., Petters, M. D., Petäjä, T., Rosenoern, T., and Sullivan, A. P.: A review of the anthropogenic influence on biogenic secondary organic aerosol. Atmos. Chem. Phys., 11(1), 321-343, 2011.

Jeong, C. H., Evans, G. J., McGuire, M. L., Chang, R. W., Abbatt, J. P. D., Zeromskiene, K.,

Mozurkewich, M., Li, S.-M. and Leaitch, W. R.: Particle formation and growth at five rural and urban sites. Atmos. Chem. Phys., 10(16), 7979-7995, 2010.

Kulmala, M., Vehkamäki, H., Petäjä, T., Dal Maso, M., Lauri, A., Kerminen, V. M., Birmilic, W., and McMurry, P. H.: Formation and growth rates of ultrafine atmospheric particles: a review of observations. J. Aerosol Sci., 35(2), 143-176, 2004.

Kulmala, M., Petäjä, T., Kerminen, V. M., Kujansuu, J., Ruuskanen, T., Ding, A., Nie, W., Hu, M., Wang, Z., Wu, Z., Wang, L., and Worsno, D.: On secondary new particle formation in China. Front. Environ. Sci. Eng., 10(5), 1-10, 2016.

[revised manuscript text omitted]

*Anonymous Referee #2*

*General Comments:*
*In this work, Zhu et al. presented new particle formation (NPF) events observed at both a street site and a rooftop site using two TSI 3091 FMPS during both spring- and winter-time. The authors reported two major findings: 1) NPF was enhanced at the street site during wintertime due to seasonal street canyon effects; 2) Photochemically oxidized biogenic organics might contribute significantly to the growth of >10 nm particles. Overall, the manuscript is fairly well written and the subject of the research is certainly within the scope of Atmospheric Chemistry and Physics (ACP). The unique feature of this work is the high time-resolution (1Hz) observations of particle number size distribution (PNSD). It does provide an advantage in NPF research. This is an interesting study but the reviewer feels that there are several issues needed to be further addressed before the manuscript can be considered for publication in ACP.*

**Response:** The authors thank the reviewer's comments and try our best to address the issues point-by point.

*Specific Comments:*
*1. L27: I do not have too much confidence that the authors can draw a conclusion of "seasonal effects" from several days of NPF events. It is more like a case study. After all, there were no summer and autumn observations.*

**Response:** Agree. The findings obtained in this study were mainly related to direct and indirect effects of largely changed ambient temperature. It will thereby revise as "the street canyon likely exerts distinct effects on NPF under warm and cold ambient temperatures because of on-road vehicle emissions" in revision.

*2. L30: "The oxidation of biogenic organics. . .apparent growth." There is no clear evidence of strong biogenic VOC emission at the sites. The authors may want to look into the anthropogenic VOC emissions, such as traffic-related VOC emissions.*

**Response:** The role of oxidation products of biogenic VOC in NPF events were widely studied in field experiments, chamber and modeling studies, and quantum chemical calculations (Schobesberger et al., 2013; Riccobono et al., 2014; Ortega et al., 2015; Tröstl et al., 2016). According to the established knowledge, the authors argued the potential importance of oxidized biogenic VOC in growing newly formed particles in this study. The north and northwest directions of the sampling site subject to mountain areas have a high percentage of land-covered forests. Extensive biogenic VOC is theoretically expected in spring and may act as important precursors in NPF. During the NPF periods, the north or northwest wind dominated and carried less polluted or even clear ambient air from mountain areas to the sampling site, e.g., the mixing ratio of $SO_2$ was < 3 ppb in the springtime and < 5 ppb in the wintertime during the periods of NPF events.

The role of oxidized anthropogenic VOCs in growing >10 nm newly formed particles is still poorly understood (Zhang et al., 2009; Hoyle et al., 2011). Considering the knowledge gap and lack of related data in this study, the authors are reluctantly to discuss the possibility in this study. Following the reviewer's comments, in the context, the authors will add "Oxidized anthropogenic VOCs could theoretically participate in the growth of newly formed particles while the study is limited (Zhang et al., 2009; Hoyle et al., 2011), but the role of oxidized anthropogenic VOCs needs further study."

*3. NPF event consists of both nucleation and the ensuing particle growth. It would be more reasonable to replace "NPF rate" with "nucleation rate".*

**Response:** In our study, the authors calculated the apparent formation rate of new particles >8nm. "Nucleation rate" currently refers to the formation of <1-3 nm new particles and clusters with more novel technologies available for measuring <3 nm particles. The authors prefer to use apparent formation rate (FR) in revision.

*4. L235-240: The authors used "25% minimum coefficient of variation (CV) of particle number concentration" as an indicator to eliminate vehicle emission spikes from the NPF dataset. If possible, I suggest the authors also use NOx as an indicator of vehicle emission plumes. For example, did the particle number concentration spikes show any correlation with NOx time series?*

**Response:** NOx was not measured at the street site. However, black carbon (BC) was measured by a potable aethalometer at the street site and BC is also a good indicator of traffic emission (Fruin, et al., 2004; Meng et al., 2015 a,b). The authors tried to use BC as an indicator of vehicle emission plumes to deduct primary traffic particles. It does not work because the one-minute time resolution is too low to successfully deduct primary traffic particles. To best of our knowledge, NOx analyzers are usually set for operating in one-minute time resolution and the data of NOx may suffer from the same problem. An example is presented to illustrate the problem for using BC to deduct primary traffic particles.

During the entire sampling period on 22 December 2010, BC shows no correlation with the nucleation mode PNC (Fig. R1a). During a few short periods, the BC spikes appeared to be visibly consistent with the PNC spikes as shown in Fig. R1b. However, the correlation obtained was much poor, e.g., during the period of 10:30-12:30 (Fig. R1c). This is not surprised because the aethalometer reported the instantaneous value of BC in one minute, but the vehicle spikes physically occurred in a few seconds (Fig. R2). Under such poor correlation, the regression equation is invalid to accurately deduct primary traffic particles.

[Figure]

Fig. R1 The nucleation mode PNC and BC on 22 December, 2010. (a, b: time series of nucleation mode PNC and BC; c: relationship between nucleation mode PNC and BC during 10:30-12:30)

[Figure]

Fig. R2 Raw FMPS data showing vehicle spikes.

According to the comments, the authors will revise the manuscript as: "Black carbon (BC) or NOx were also proposed to deduct the contribution of vehicle spikes (Fruin et al., 2004; Wang et al., 2012). The measured concentration of BC was thereby tried for deduction and much poor correlation was obtained, as presented in supplementary document. Under such poor correlation, the regression equation between PNC and BC was invalid to accurately deduct the contribution of vehicle spikes. Theoretically, the correlation could be improved substantially with increasing distance of the sampling site from traffic roads because of less dynamic changes in PNC and PNSD (Zhu et al., 2002 a,b, 2006). Although the two approaches used in this study may still suffer from uncertainty to some extent, they should be much better than those reported in literature. However, the two approaches can be done only when high time resolution particle sizers were used for measurements."

*5. As described by the authors that the street site was "18 m away from the curb of a heavy traffic (Chengfu) Road at the northwestern area in Beijing", it would be necessary not only to remove spikes of vehicle emissions but also to take into account the small particles transported from further down the road, which may be not as distinct as the spikes caused by passing by vehicles but would certainly raise the background level of ~10 nm particles.*

**Response:** When the authors calculated the new particle formation rate, we use the difference of nucleation mode PNC after and before the PNC increasing period. The background level of ~10 nm particles can be eliminated and does not affect our calculation results.

*6. L280-290: Ambient temperature changed substantially from spring to winter as indicated in the experimental section. The authors may also want to consider the role of weather in affecting the nucleation rate.*

**Response:** On the same day, ambient temperature at the street site and on the rooftop can be reasonably assumed to be same by considering only ~500 m distance between. Considered ambient temperature alone, it should not be the direct cause for the reduced or enhanced NPF at the street site relative to the rooftop site.

*7. L285: In northern China, SO2, especially during wintertime, may also come from domestic heating, which can substantially increase SO2 emission. The authors may want to include this possibility.*

**Response:** The authors thank the comments. L285 will revise to: "2) In the December, the mixing ratios of $SO_2$ at 1-2 hours immediately before NPF were 3-5 ppb at the rooftop site. The $SO_2$ was mainly from domestic heating and the traffic-derived $SO_2$ at the street site was roughly estimated to be <1.3 ppb according to the results in our previous studies (Meng et al., 2015 a,b). The concentrations of sulfuric acid at the street site may be very likely close to or even lower than those at the rooftop site since the stronger scavenging effect probably canceled out the traffic-derived contribution to sulfuric acid." More information will be added in the revised supplementary: "The sulfur content in the gasoline and diesel was limited <50 ppm at those years. The measured BC spikes were lower than 5 $\mu g\ m^{-3}$ during the NPF periods. The maximum contribution of traffic-related $SO_2$ at the street site was roughly estimated to be 1.3 ppb according to the results in our previous studies (Meng et al., 2015 a,b). In the wintertime, the ratio of traffic-derived $SO_2$ to the observed values was less than 1/4 and the observed values were overwhelmingly contributed by domestic heating. The uncertainty by assuming $SO_2$ at the street site same as the rooftop site should be minor. It should not affect our conclusion because the formation rates of new particles at the street site were increased by 3-5 times against the rooftop site in the wintertime. In the springtime, the contribution of traffic-related $SO_2$ might significantly increase the mixing ratio of $SO_2$ at the street site. However, the reduced NPF was observed at the

street site. The possible underestimation of SO₂ at the street site further solidified our analysis results, i.e., a strong scavenge effect at the street site likely existed and caused the reduced NPF."

*8. Section 3.5: Clear particle growth observed at a ground site is often associated with a regional NPF event. The short burst of nucleation events reported here may indicate that the air parcels were frequently disrupted by the urban micrometeorology conditions, which should be further investigated.*

**Response:** In section 3.5, the duration period of Class II NPF events lasted for 4-8 hours when the wind speed larger than 4m/s. This strongly implied the Class II NPF events were likely to occur in a regional scale. Similar to Class II NPF events with the particle growth to be undetectable presented in this study, extremely low growth rate of newly formed particles (~ 1nm h⁻¹) in Beijing was also previously reported by Wehner et al. (2004). In our unpublished data, the authors simultaneously observed Class II NPF event and NPF event with extremely low growth rate at ~240 km distance (Fig. R3, the case will also be presented in Supplementary). In the last three years (data unpublished), we had simultaneous observations of NPF events at 100-500 km distance. The authors obtained six cases based on simultaneous observations at two locations, i.e., one case featured by Class II NPF vs Class II NPF, four cases featured by Class II NPF vs NPF with an extremely low growth rate, one case featured by Class II NPF vs NPF with "banana shape" particle growth. The authors strongly believed that Class II NPF events lasted for 4-8 hours should be considered as regional NPF events. Evidences of Class II NPF events regarding as regional NPF events will be add in supplementary.

[Figure]

Fig. R3 Simultaneous observed Class II NPF event and NPF event with extremely low growth rate at ~240 km distance.

**References:**

[revised manuscript text omitted]

*Anonymous Referee #3*

*The manuscript presents results from measurements of new particle formation (NPF) in Beijing during Spring and Winter using high time resolution particle sizers. The characteristics of new particle formation at two adjacent sites were compared to assess the impacts of local traffic emissions on NPF. Traffic exhausts might emit primary particles or gas phase precursors that contribute significantly to secondary particle formation and hence traffic emissions play important roles in severe haze formation in megacities such as Beijing in China. While the topic is important in atmospheric chemistry and is of interest to the general readers of this journal, the manuscript in general is yet to be improved and several major issues need to be resolved before the manuscript can be publishable in the journal.*

Response: The authors thank the reviewer's comments and try our best to address the issues point-by point.

*Major comments:*
*1. The rational of selecting plume events rather than regional events as examples must be clearly persuasive. Apparently, good "Banana shape" regional events were measured during the campaigns. Comparison of the differences of new particle formation between the two sites is of great interest. It will be clearer to see the impacts of traffic emissions on the new particle formation processes if those well-defined events were used as examples. For example, how particle number size distribution and particle composition might be affected by local emissions. The plume events are rather not well defined in term of the formation rates and the growth rates which will need to be resolved in the next comment.*

Response: All simultaneous observations at two sites have been included and presented in the original version. During the two periods of this study, the authors didn't observe "banana shape" regional events simultaneously occurring at two sites. Therefore, the authors had no way to use them for discussion. This will be clarified in revision.

Similar to Class II NPF events with the particle growth to be undetectable presented in this study, extremely low growth rate of newly formed particles ($\sim 1$ nm h$^{-1}$) in Beijing was also previously reported by Wehner et al. (2004). In our unpublished data, the authors simultaneously observed Class II NPF event and NPF event with extremely low growth rate at ~240 km distance (Fig. R1, the case will also be presented in Supplementary). In the last three years (data unpublished), we had simultaneous observations of NPF events at 100-500 km distance. The authors obtained six cases based on simultaneous observations at two locations, i.e., one case featured by Class II NPF vs Class II NPF, four cases featured by Class II NPF vs NPF with an extremely low growth rate, one case featured by Class II NPF vs NPF with "banana shape" particle growth. The authors strongly believed that Class II NPF

events lasted for 4-8 hours should be considered as regional NPF events. "Banana shape" particle growth, extremely low particle growth and no detectable growth are probably related to spatial heterogeneity of gaseous precursors supporting the growth of > 10 nm particles, but NPF events indeed occurred regionally.

[Figure]

Fig. R1 Simultaneous observed Class II NPF event and NPF event with extremely low growth rate at ~240 km distance.

FMPS measured different sized particle number concentration in one second. The high time-resolution data can allow calculating the apparent formation rate of new particles such as $J_8$ almost in any complicated situations. The reviewer's comments may be related to the low time-resolution data such as in 5-10 minutes time resolution, but it should not be the problem for FMPS measurements. This is also probably why FMPS measurements can show a few unique findings like in this study, but it is impossible for the 5-10 minutes time-resolution measurement previously conducted.

*2. Particle formation rates and growth rates. First, the formation rate should be stick to J8 instead of new particle formation rate since particles in the range of 8-20 nm are rather too big to be called new particles. Also why a size range of 8-20nm is selected? Why is not 8-30 or 40 nm or others? The determination of the formation time (not the nucleation time) used in this paper seems to be objective and is of the authors' preferences, profoundly affecting the formation rate calculations. The width of the size range also affects the determination of formation time which will need to be clarified. The formation rate of nanoparticles in a plume event is difficult to calculate and needs to use a more sophisticate method than the simple one used in this paper.*

**Response:** The authors agree that 8-20 nm particles should be called as grown new particles rather than new particles. In revision, we will add "$J_8$ reflects a combination result of nucleation and subsequent initial growth of new particles." in the explanation of new particle formation rate.

The authors followed the conventional approach to calculate the apparent formation rate of new particles. In literature, the upper limits of the particle size was reportedly set at 20 nm, 25 nm or 30 nm (Kulmala et al., 2004; Kulmala and Kerminen, 2008; Yao et al., 2010; Kulmala et al., 2012). 20 nm was set as the upper limit in this study, following the suggestions by Kulmala et al. (2012), i.e., 1) newly formed particles rarely grow over 20 nm during the initial nucleation time, 2) nucleation may occur at

some distances from the measurement point and the particles have grown a few nanometers before they were observed. In addition, the upper limit of 20 nm can minimize possible interferences from primary traffic particles. This will be clarified in revision.

FMPS measured different sized particle number concentration in one second. The high time-resolution data allow calculating the apparent formation rate of new particles such as $J_8$ almost in any complicated situations. The reviewer's comments may be related to the low time-resolution data such as in 5-10 minutes time resolution, but it should not be the problem for FMPS measurements. This is also probably why FMPS measurements can show a few unique findings like in this study, but it is impossible for the 5-10 minutes time-resolution measurement previously conducted.

The authors used two methods to minimize the interference from primary traffic particles and the calculated apparent formation rates were consistent. Again, the high time-resolution data indeed allows calculating the apparent formation rate of new particles such as $J_8$ in presence of strong interferences.

*3. The classification of the nucleation events. It is very awkward to denote a nucleation event longer than 1 hour "a long term event". It might sound better if "a long lasting event" or another name is adopted. Similarly, please change the notation of "a short term event" to another proper name. In addition, the two types of events are the wellknown "regional events" and "plume events" in atmospheric aerosol sciences. It is not necessary to create new names for them.*

**Response:** The authors would like to correct "short-term NPF event, long-term NPF event" to "short-lived NPF event, regional NPF event" (Stanier et al., 2010; Jeong et al., 2010).

The authors strongly believed that Class II NPF events lasted for 4-8 hours should be considered as regional NPF events. Details can be found in the response to major comment 1.

*4. Heterogeneity of NPF in Section 3.2. Quite a few "heterogeneity" were mentioned for NPF in both horizontal and vertical directions. It is really not that meaningful to emphasize the spatial inhomogeneity of particle formation because in the urban atmosphere, gas phase precursors are inhomogeneous and particle formation is also greatly constrained by local emissions and meteorological conditions.*

**Response:** The authors thank and fully agree "the spatial inhomogeneity of particle formation because in the urban atmosphere, gas phase precursors are inhomogeneous and particle formation is also greatly constrained by local emissions and meteorological conditions". A question is then automatically raised, i.e., what does it mean "regional NPF events"? It is just because it regionally occurred, but could be in

different formation rates, growth rates, different initial bursting times? Most potential readers may disagree.

Most readers may also disagree with the reviewer and the authors at this point, i.e., the spatial inhomogeneity of particle formation very commonly occur in regional NPF events. However, the authors agree to cut a few details of "heterogeneity" in revision.

*5. Reasons for the reduced NPF and the enhanced NPF at the street site respectively in the springtime and in the wintertime. It is very interesting to figure out the reasons behind those observed phenomena. First, the authors need to confirm that particle formation is always reduced in the springtime and always enhanced in the springtime at the street site. That will exclude the possibility of dominant effects from the meteorological conditions e.g. differences in wind directions or mixing heights, temperature, humidities etc. Second, the authors need to present more other companion measurements of gas phase precursors and chemical composition of nanoparticles in order to elucidate the mechanisms of reduced or enhanced NPF at the street site. Without the information, the proposed explanation for the observed opposite effects on NPF during springtime and wintertime is only speculative.*

**Response**: At the street site, the reduced NPF events always occurred in the springtime while the enhanced NPF events always occurred in the wintertime. This will be highlighted in revision.

The authors provided three types of evidences from different angles to confirm the reduced NPF rather than simple comparison between rooftop site and street site measurements, i.e., Evidence 1: The lower particle number concentration (PNC) of nucleation mode particles at the street site mainly because of a shorter initial burst time. Evidence 2: The authors used the PNC at the street site subtracting the corresponding PNC at the rooftop site to calculate the difference. The authors then obtained the second evidence: the negative difference of nucleation mode particles against the positive difference of Aitken mode particles on NPF days. Evidence 3: Using the same approach, the authors obtained the third evidence: the negative difference of nucleation mode particles on NPF days against the positive difference of that on non-NPF days (Figs. 3 and 4 in the origin version). The second and third evidences are not affected by these factors suggested by the reviewer. All these will be better clarified in revision.

The authors also provided three types of evidences from different angles, rather than simple comparison between rooftop site and street site to confirm the enhanced NPF in the wintertime, i.e., Evidence 1: The significantly larger PNC of nucleation mode particles at the street site and a larger apparent formation rate of new particles mainly because of a shorter initial burst time. Evidence 2: The positive difference of nucleation mode particles in the wintertime against the negative difference of nucleation mode particles in the springtime on NPF days. Evidence 3: The larger

positive difference of nucleation mode particles on NPF days against that on non-NPF days in the wintertime (Figs. 5 and 7 in the origin version). Again, the second and third evidences are not affected by these factors suggested by the reviewer. All these will be better clarified in revision.

To confirm what the reviewer suggested requires simultaneous measurements of vapor precursors such as $H_2SO_4$, HOM, etc, and chemical composition of ~10 nm particles at the rooftop site and street site. This is indeed beyond the capacity of the whole research community, but not only for the authors. The authors tried the best and used all data we had. Other indirect evidences such as chemical composition of >50 nm particles were reportedly used to argue NPF in literatures. Theoretically, the arguments could be true, but also could be misleading by considering largely varying chemical composition in different sized atmospheric nanoparticles. The authors hope that the reviewer can agree this.

*Minor comments:*
*1. There are a lot of typos, ill-sentences all over the manuscript. It is recommended that the manuscript should be carefully edited prior to submission. Below are a few examples: L31, specie; L208, a several minutes; L254, didn't detail; L270, didn't detail description; L292, "were available currently", . . .*

**Response:** The authors are sorry for this and will correct the errors through the manuscript before re-submitting.

*2. Rewrite all the figure captions clearly as those captions are hard to read and understand.*

**Response:** Thank for the comment, the authors will rewrite all the figure captions.

*3. L225, a few more sentences might be needed to explain why NPF inside the street canyon was reduced.*

**Response:** The authors will revise L225 as: "Although the number of the cases was not large, the reduced NPF events at the street site were theoretically expected on basis of well recognized factors in literature, e.g., 1) a larger condensation sink associated with more pre-existing atmospheric particles from primary emissions; 2) tall buildings along both the sides of urban streets can provide additional surface areas to scavenge gases and atmospheric particles (Yao et al., 2011); 3) vehicle-emitted NO reacting with $RO_2$ and suppressing NPF (Wildt et al., 2014)."

*4. L189, "estimating that the NPF possibly occurred in cleaner atmospheres over the region scale of ~120 km", "suggesting that the NPF likely occurred in cleaner atmospheres over the region scale of ~140 km in different NPF rates.", "The NPF was roughly estimated to occur in a semi-regional scale over ~50 km". How do you*

*know the scales of those events?*

**Response:** The authors agree that our estimation indeed suffered from uncertainty. In revision, it will revise as "the long lasting time for NPF implied that the events occur in regional or semi-regional scale." Again, the authors believed that the long lasting Class II NPF events may occur regionally and the argument has presented above.

**Response:** The authors believe the turbulence dispersion to be more important than advection diffusion at the street site under a strong synoptic wind. The authors didn't measure wind direction and wind speed at multiple street locations and the authors didn't think one point observation of wind direction and wind speed can full reflect the complicated micro-scale wind field at the street site. The authors thereby reluctantly speculated the influence of the complicated micro-scale wind field on the observed particle number concentrations between the rooftop site and street site.

Through a deep data analysis, the authors additionally provided two types of unique evidences which were less affected by micro-meteorology at the street site as well as the simplest evidence, i.e., the simple comparison between the rooftop site and street sites, to confirm the reduced and enhanced NPF at the street site in different seasons. The authors found that the sufficiency and uniqueness of three types of evidences were not fully recognized by the reviewers because their challenges mainly focused on the simplest evidence. This means the authors' presentation strategy has to be improved to make the three types of evidences more obviously.

In revision, the authors will clarify that the reduced NPF always occurred at the street site in the springtime. Three evidences from different angles will be itemized as Evidence 1, 2 and 3, i.e., Evidence 1: The lower particle number concentration (PNC) of nucleation mode particles at the street site mainly because of a shorter initial burst time. Evidence 2: The authors used the PNC at the street site subtracting the corresponding PNC at the rooftop site to calculate the difference. The authors then obtained the second evidence: the negative difference of nucleation mode particles against the positive difference of Aitken mode particles on NPF days. Evidence 3:

Using the same approach, the authors obtained the third evidence: the negative difference of nucleation mode particles on NPF days against the positive difference of that on non-NPF days (Figs. 3 and 4 in the origin version).

In addition, the authors will clarify that the enhanced NPF always occurred at the street site in the wintertime. Three evidences from different angles will be itemized as Evidence 1, 2 and 3, i.e., Evidence 1: The significantly larger PNC of nucleation mode particles at the street site and a larger apparent formation rate of new particles mainly because of a shorter initial burst time. Evidence 2: The positive difference of nucleation mode particles in the wintertime against the negative difference of nucleation mode particles in the springtime on NPF days. Evidence 3: The larger positive difference of nucleation mode particles on NPF days against that on non-NPF days in the wintertime (Figs. 5 and 7 in the origin version).

We agree that the loss of nanoparticles due to the surfaces along the street canyon can be a potential factor of the reduced NPF in the springtime. In revision, We will revise L225 as: "the reduced NPF events at the street site were theoretically expected on basis of well recognized factors in literature, e.g., 1) a larger condensation sink associated with more pre-existing atmospheric particles from primary emissions; 2) tall buildings along both the sides of urban streets can provide additional surface areas to scavenge gases and atmospheric particles (Yao et al., 2011); 3) vehicle-emitted NO reacting with $RO_2$ and suppressing NPF (Wildt et al., 2014)."

*2. The inter-comparison between two FMPSs showed some differences, and the authors decided to use one FMPS as a reference and correct the number concentration of the other one. How did they decide which one is "the one" to trust? Nevertheless, number concentrations are used to calculate formation rates, growth rates, and condensation sink. This could lead to a major uncertainty in the discussion for nucleation mechanism.*

**Response:** As reported by Zimmerman et al. (2015), an independent measurement of CPC simultaneously with FMPS can be used to accurately correct the FMPS data. The reference has been cited in the manuscript. In this study, the FMPS data perform a three-step calibration as clarified in revised supplementary:

"First step: When the two FMPS operated side-by-side during 12-17 April 2012 for inter-comparison, we get the correction factors for one FMPS (Table S1). The FMPS which operating simultaneously with a CPC at the street site afterwards was used as the reference to correct the other.

Second step: The FMPS operating simultaneously with a CPC at the street site was then processed the second-step correction proposed by Zimmerman et al. (2015). The ratio of $FMPS_{total}/CPC_{total}$ was calculated to be 1.28.

Third step: The calculated ratio of 1.28 and correction method in Zimmerman et al. (2015) was used to correct the FMPS data at the rooftop site."

*3. The FMPSs were placed downstream of dryers, which indicates that the measured size distributions could be of from the atmospheric ones. This at least eliminates the role of relative humidity to a certain extent. Even for particles in the size ranges of 10-20 nm, the uptake of H2O is one of the major pathways for particle to grow.*

**Response:** Relative humidity (RH) varied from 13% to 49% during NPF event days in the wintertime and below 55% during NPF event days in the springtime. At such RH levels, the growth factor of 10-20 nm particles (assume to be $(NH_4)_2SO_4$) were less than 1.02 (Hämeri et al., 2000). This will be added in revision.

*4. The mixing ratio of SO2 was only measured at the rooftop site. How about CO? It might be possible to deduce a street SO2 simply by the mixing ratios of CO. The current assumption that concentrations of SO2 are identical at the two sites are not acceptable, and could lead to mis-interpretation.*

**Response:** The authors thank the comments and the $SO_2$ concentration at the two sites indeed needs more interpretation.

In this study, CO was not measured at the street site. However, black carbon (BC) was measured by a potable aethalometer at the street site and BC is also a good indicator of traffic emission (Fruin, et al., 2004; Meng et al., 2015 a,b). The authors tried to use BC to deduce the traffic-related $SO_2$ at the street site. In revised manuscript and supplementary, the authors will add: "The sulfur content in the gasoline and diesel was limited <50 ppm at those years. The measured BC spikes were lower than 5 µg $m^{-3}$ during the NPF periods. The maximum contribution of traffic-related $SO_2$ at the street site was roughly estimated to be 1.3 ppb according to the results in our previous studies (Meng et al., 2015 a,b). In the wintertime, the ratio of traffic-derived $SO_2$ to the observed values was less than 1/4 and the observed values were overwhelmingly contributed by domestic heating. The uncertainty by assuming $SO_2$ at the street site same as the rooftop site should be minor in the wintertime and it should not affect our conclusion because the formation rates of new particles at the street site were increased by 3-5 times against the rooftop site in the wintertime. In the springtime, the contribution of traffic-related $SO_2$ might significantly increase the mixing ratio of $SO_2$ at the street site. However, the reduced NPF was observed at the street site. The possible underestimation of $SO_2$ at the street site further solidified our analysis results, i.e., a strong scavenge effect at the street site likely existed and caused the reduced NPF."

*5. The authors focused on the oxidation of biogenic organics when discussing the growth of >10 nm particles. In an urban environment such as Beijing, wouldn't anthropogenic VOCs be more concentrated? Are there any measurements that point*

*the authors to biogenic VOCs instead of anthropogenic ones? How will the interpretation*

**Response:** The authors thank the comments. The role of oxidation products of biogenic VOC in NPF events were widely studied in field experiments, chamber and modeling studies, and quantum chemical calculations (Schobesberger et al., 2013; Riccobono et al., 2014; Ortega et al., 2015; Tröstl et al., 2016). According to the established knowledge, the authors argued the potential importance of oxidized biogenic VOC in growing newly formed particles in this study. The north and northwest directions of the sampling site subject to mountain areas have a high percentage of land-covered forests. Extensive biogenic VOC is theoretically expected in spring may act as important precursor in NPF. During the NPF periods, the north or northwest wind dominated and carried less polluted or even clear ambient air from mountain areas to the sampling site, e.g., the mixing ratio of SO2 was < 3 ppb in the springtime and < 5 ppb in the wintertime during the periods of NPF events.

The role of oxidized anthropogenic VOCs in growing >10 nm newly formed particles is still poorly understood (Zhang et al., 2009; Hoyle et al., 2011). Considering the knowledge gap and lack of related data in this study, the authors are reluctantly to discuss the possibility in this study. Following the reviewer's comments, in the context, the authors will add "Oxidized anthropogenic VOCs could theoretically participate in the growth of newly formed particles while the study is limited (Zhang et al., 2009; Hoyle et al., 2011), but the role of oxidized anthropogenic VOCs needs further study."

*6. Throughout the manuscript, the authors are presenting J8, which is fine. However, particles bigger than 8 nm are larger enough that they don't really reflect the nucleation mechanism, instead, a combination of nucleation and subsequent growth, especially growth mechanisms, might actually determines how many particles were measured.*

**Response:** Different reviewers clearly had different views on the apparent formation rate of new particles. As a compromise by considering all reviewers comments, $J_8$ will be used in revision. We will add additional interpretation: "$J_8$ reflects a combination result of nucleation and subsequent initial growth of new particles." in the explanation of new particle formation rate.

***Minor comments,***
*7. (Page 8), clearly define long-term NPN, short-term NPF, Class I NPF, and class II NPF.*

**Response:** The authors thank the comments. The authors would like to correct "long-term NPF events, short-term NPF events" to "regional NPF events, short-lived NPF events" following Stanier et al. (2010) and Jeong et al. (2010). Regional NPF

events represent NPF events lasting over 1 hour, and short-lived NPF events represent the NPF lasting only for 10-20 minutes. When new particles showed "banana shape" growth, and the new particles grow to larger than 30 nm, we define as Class I NPF event. When the new particles showed no apparent growth and the particles diameter was about 11 nm, we define as Class II NPF event. In this study, the duration of NPF events in Class I and Class II were lasting longer than 1 hour, which belong to the regional NPF events.

*8. (Page 10, Line 225), how about NO in the winter? Wouldn't NO be always higher in the street canyon?*

**Response:** Theoretically, NO could be always higher in the street canyon not only in the springtime but also in the wintertime by considering on-road traffic emissions closer to the street site. The enhanced NPF at the street site implied that the enhanced effects overwhelming the reduced effects. A revision will be done to reflect this argument.

*9. (Page 13, Lines 280-287), the argument on H2SO4 is just speculation. Many factors determines H2SO4. Also, why is SO2 from on-road vehicles negligible comparing to the background?*

**Response:** In this study, the NPF events occurred under the north or northwest wind direction with wind speed >4m/s. The north or northwest wind carried less polluted or even clear air to the sampling site during the NPF event periods, e.g., the mixing ratio of $SO_2$ was < 3 ppb in the springtime and < 5 ppb in the wintertime during the periods of NPF events. Less polluted or even clear air exactly meets the criteria of equation (4), i.e., the proxy for calculating gaseous sulfuric acid (SA) concentration is applicable under clean to moderately-polluted atmospheres.

In revision, the authors use BC to deduce that the traffic-related $SO_2$ was less than 1.3 ppb (details can be found in the response to comment 4). In the wintertime, the maximum contribution of traffic-related $SO_2$ to the observed values was roughly estimated as 1/4 and the observed values were overwhelmingly contributed by domestic heating. In the springtime, the contribution of traffic-related $SO_2$ might significantly increase the mixing ratio of $SO_2$ at the street site.

*10. (Page 15, the last paragraph), I am certainly not convinced by the discussion here. Different nucleation mechanisms probably explains NPF events in Beijing, Qingdao, and marginal seas of China. Again, J8 is not a good indicator for nucleation mechanisms. By definition, NMIoNP stand for "the net maximum increase of nucleation mode PNC". I don't see a clear connection between NMIoNP and J8. A cutoff of 8 cm-3 s-1 could be arbitrary. The correlation will not be bad if a cutoff of , say, 7 cm-3 s-1, was chosen.*

**Response:** Theoretically, the formation rate of new particles alone has no direct relationship with the environmental and climate effects of new particles. However, NMIoNP defined as "the net maximum increase of nucleation mode PNC" (revise to NMINP, the net maximum increase in nucleation mode PNC in revision) is directly related to the environmental and climate effects of new particles. Thus, it is important to establish the link between NMINP and $J_8$.

In revision, the authors will add: "Considered 1) formation rate of new particles, e.g., $J=k_{NucOrg}[H_2SO_4]^m[NucOrg]^n$ (Zhang et al., 2012) where $k_{NucOrg}$ is a constant, NucOrg represents organics involved in nucleation, m and n are two integers, 2) the subsequent particle growth, and 3) $H_2SO_4$ vapor to be necessary for nucleation in ambient air except at sea beach, two scenarios are analyzed. Scenario 1: $H_2SO_4$ vapor is relatively sufficient against NucOrg and $J_8$ is thereby mainly determined by availableness of NucOrg vapor. A good correlation is theoretically expected for $J_8$ and NMINP. Scenario 2: NucOrg vapor is relatively sufficient against $H_2SO_4$ vapor and $J_8$ is thereby mainly determined by availableness of $H_2SO_4$ vapor. $J_8$ could be high, but the total yield of new particles could be low because of a rapid consumption of $H_2SO_4$ vapor. A poor or no correlation is theoretically expected for $J_8$ and NMINP. "

The authors found when the formation rate less than $8$ $cm^{-3}$ $s^{-1}$, the new particle yield increased with the increasing formation rate. Of course, for any values $<8$ $cm^{-3}$ $s^{-1}$, the statement is valid. The authors didn't find any logic problem.

*11. In supplementary, coefficient of variation (CV) is defined, but try to define "25% minimum", especially what "minimum" stands for. Also, why 1 16.6 nm cutoff chosen in the following session?*

**Response:** The authors revise the description of 25% minimum CV as: "Coefficient of variation (CV) is the ratio of standard deviation to mean for particle number concentration in every 60s. We sort the data sequence in an order from the smallest to the largest, and the minimum 25% values of CV reflect smaller changes of particle number concentration."

$N_{16.5nm}$ used as the reference to conduct the deduction relied on two reasons: 1) Zhu et al. (2006) also observed the vehicle particle mode around 16 nm at 30 m downwind. 2) In this study, 10 nm-30 nm was the major mode in the vehicle particle size distribution, and 16.5 nm was the peak of this particle mode. This will be added in revision.

*12. Proofread the manuscript.*

**Response:** The authors are sorry for this and the language-editing will be processed before re-submitting.

**Reference:**

[revised manuscript text omitted]

*Anonymous Referee #5*

*Fast-response measurements of particle number size distributions of aerosol ≥8 nm diameter have been made at a street canyon and nearby rooftop site. The authors selectively report specific days of data from a small dataset, and draw many tentative conclusions concerning mechanisms of new particle formation (NPF) which are difficult to justify given the small dataset and the extent to which it is over-interpreted. The introduction quite reasonably states that "it is critical to evaluate the effects of nucleating species other than sulfuric acid and the dependence of NPF on pre-existing particles in the atmosphere". This is an excellent objective but unfortunately the paper does nothing to answer the question about other nucleating species, and does not event provide clear answers concerning the role of sulfuric acid.*

**Response:** The authors didn't selectively present the data. All simultaneous measurements on particle number concentrations during the springtime and wintertime campaigns have been included in this study. At the street site, the reduced NPF events always occurred in the springtime and the enhanced NPF events always occurred in the wintertime. This will be highlighted in revision.

The authors would like to believe the sufficiency and uniqueness of evidences are critical to evaluate the quality of scientific studies. This is because the number of cases either for gravitational wave observation in 2016 or a recent NPF study reported by Bianchi et al. (2016) was even smaller than those presented in this study.

This study is definitely not a first study for NPF events and all analyses build on previously well-established knowledge, particularly the progressing in the last few years. The authors did provide unique evidences to analyze the effects of nucleating species other than sulfuric acid at street site. The authors also agree that more analyses should be added so that they can be easily understood. For example, in revision, the authors will add "Considered 1) formation rate of new particles, e.g., $J=k_{NucOrg}[H_2SO_4]^m[NucOrg]^n$ (Zhang et al., 2012) where $k_{NucOrg}$ is a constant, NucOrg represents organics involved in nucleation, m and n are two integers, 2) the subsequent particle growth, and 3) $H_2SO_4$ vapor to be necessary for nucleation in ambient air except at sea beach, two scenarios are analyzed. Scenario 1: $H_2SO_4$ vapor is relatively sufficient against NucOrg and $J_8$ is thereby mainly determined by availableness of NucOrg vapor. A good correlation is theoretically expected for $J_8$ and NMINP. Scenario 2: NucOrg vapor is relatively sufficient against $H_2SO_4$ vapor and $J_8$ is thereby mainly determined by availableness of $H_2SO_4$ vapor. $J_8$ could be high, but the total yield of new particles could be low because of a rapid consumption of $H_2SO_4$ vapor. A poor or no correlation is theoretically expected for $J_8$ and NMINP. "

*One of the key elements towards interpretation of this dataset in relation to nucleation and growth is the role of sulfuric acid, which ideally would have been measured. However, as measurements were not available, an old parameterisation is used to*

*estimate H2SO4 vapour concentrations in which the H2SO4 formation rate is described by the product of SO2 concentration and global solar radiation. This may be adequate for situations in the background troposphere where ozone photolysis is the predominant source of hydroxyl radical, but many studies have now shown that in polluted atmospheres such as Beijing, other processes such as photolysis of HONO and HCHO, and ozone-alkene reactions are far more important sources of hydroxyl, and equation 4 is unlikely to be a reliable means of calculation of [H2SO4].*

**Response:** The authors would like to believe that the reviewer may capture a piece of information about air pollution in Beijing, but it is hard to say the piece reflecting the full picture.

During the periods of NPF events in this study, the air mass at the sampling site was less polluted or even clear. The NPF events occurred under the north or northwest wind direction with wind speed >4m/s. The north and northwest directions of the sampling site subject to mountain areas have a high percentage of land-covered forests. The north or northwest wind carried less polluted or even clear ambient air to the sampling site during the NPF event periods, e.g., the mixing ratio of $SO_2$ was < 3 ppb in the spring and < 5 ppb in the winter during the periods of NPF events. Less polluted or even clear air exactly meets equation 4 in the manuscript. The authors cannot find what problem in our approach is.

*The differences in behaviour between the sites are interesting, and if correctly interpreted could give useful insights into NPF in polluted atmospheres. However no measurements were made of potentially condensing species, or their precursors other than SO2, and the latter was measured at only one site with the unproven assumption that concentrations of SO2 were the same at both sites. Much is made of the rates of change of particle number concentrations, but the effects of wind direction changes upon concentrations in the street canyon (which can be large) do not appear to have been considered. The methods used for subtraction of fresh traffic emissions are highly questionable, and no use is made of gaseous pollutant data (e.g. NOx) which would be a strong covariate of PNC from road vehicles.*

**Response:** The authors would like to believe that the reviewer may capture a piece of information about air pollution in Beijing, but it is hard to say the piece reflecting the full picture. The technical term "polluted atmospheres" is probably not applicable for NPF events reported in this study.

The authors had no simultaneous measurements of other condensing species, such as $H_2SO_4$, HOM, etc., and had no way to discuss them. The authors agree that the $SO_2$ concentration at the two sites indeed needs more interpretation. In revised manuscript and supplementary, the authors will add "The sulfur content in the gasoline and diesel was limited <50 ppm at that years. The measured BC spikes were lower than 5 μg m$^{-3}$ during the NPF periods. The maximum contribution of traffic-related $SO_2$ at the street

site was roughly estimated to be 1.3 ppb according to the results in our previous studies (Meng et al., 2015 a,b). In the wintertime, the ratio of traffic-derived $SO_2$ to the observed values was less than 1/4 and the observed values were overwhelmingly contributed by domestic heating. The uncertainty by assuming $SO_2$ at the street site same as the rooftop site should be minor in the wintertime and it should not affect our conclusion because the formation rates of new particles at the street site were increased by 3-5 times against the rooftop site in the wintertime. In the springtime, the contribution of traffic-related $SO_2$ might significantly increase the mixing ratio of $SO_2$ at the street site. However, the reduced NPF was observed at the street site. The possible underestimation of $SO_2$ at the street site further solidified our analysis results, i.e., a strong scavenge effect at the street site likely existed and caused the reduced NPF."

The authors believe the turbulence dispersion to be more important than advection diffusion at the street site under a strong synoptic wind. The authors didn't measure wind direction and wind speed at multiple street locations and the authors didn't think one point observation of wind direction and wind speed can full reflect the complicated micro-scale wind field at the street site. The authors thereby are reluctantly speculated the influence of the complicated micro-scale wind field on the observed particle number concentrations between the rooftop site and street site.

Through a deep data analysis, the authors additionally provided two types of unique evidences which were less affected by micro-meteorology at the street site as well as the simplest evidence, i.e., the simple comparison between the rooftop site and street sites, to confirm the reduced and enhanced NPF at the street site in different seasons. The authors found that the sufficiency and uniqueness of three types of evidences were not fully recognized by the reviewers because their challenges mainly focused on the simplest evidence. This means the authors' presentation strategy has to be improved to make the three types of evidences more obviously.

In revision, the authors will clarify that the reduced NPF always occurred at the street site in the springtime. Three evidences from different angles will be itemized as Evidence 1, 2 and 3, i.e., Evidence 1: The lower particle number concentration (PNC) of nucleation mode particles at the street site mainly because of a shorter initial burst time. Evidence 2: The authors used the PNC at the street site subtracting the corresponding PNC at the rooftop site to calculate the difference. The authors then obtained the second evidence: the negative difference of nucleation mode particles against the positive difference of Aitken mode particles on NPF days. Evidence 3: Using the same approach, the authors obtained the third evidence: the negative difference of nucleation mode particles on NPF days against the positive difference of that on non-NPF days (Figs. 3 and 4 in the origin version).

In addition, the authors will clarify that the enhanced NPF always occurred at the street site in the wintertime. Three evidences from different angles will be itemized as

Evidence 1, 2 and 3, i.e., Evidence 1: The significantly larger PNC of nucleation mode particles at the street site and a larger apparent formation rate of new particles mainly because of a shorter initial burst time. Evidence 2: The positive difference of nucleation mode particles in the wintertime against the negative difference of nucleation mode particles in the springtime on NPF days. Evidence 3: The larger positive difference of nucleation mode particles on NPF days against that on non-NPF days in the wintertime (Figs. 5 and 7 in the origin version).

In this study, NOx was not measured at the street site. However, black carbon (BC) was measured by a potable aethalometer at the street site and BC is also a good indicator of traffic emission (Fruin, et al., 2004; Meng et al., 2015 a,b). The authors tried to use BC as an indicator of vehicle emission plumes to deduct primary traffic particles. It does not work because the one-minute time resolution is too low to successfully deduct primary traffic particles. To best of our knowledge, NOx analyzers are usually set for operating in one-minute time resolution and the data of NOx may suffer from the same problem. An example is presented to illustrate the problem for using BC to deduct primary traffic particles.

During the entire sampling period on 22 December 2010, BC shows no correlation with the nucleation mode PNC (shown in Fig. R1a). During a few short periods, the BC spikes appeared to be visibly consistent with the PNC spikes as shown in Fig. R1b. However, the correlation obtained was much poor, e.g., during the period of 10:30-12:30 (shown in Fig. R1c). This is not surprised because the aethalometer reported the instantaneous value of BC in one minute, but the vehicle spikes physically occurred in a few seconds (shown in Fig. R2). Under such poor correlation, the regression equation is invalid to accurately deduct primary traffic particles.

[Figure]

Fig. R1 The nucleation mode PNC and BC on 22 December, 2010. (a, b: time series

of nucleation mode PNC and BC; c: relationship between nucleation mode PNC and BC during 10:30-12:30)

[Figure]

Fig. R2 Raw FMPS data showing vehicle spikes.

*The points above justify a major reappraisal of the data, and the development of far less ambitious conclusions.*

**Response:** The authors will try the best to revise to improve the manuscript. The authors have to say that "polluted atmospheres" in Beijing much claimed by the reviewer may be not closely related to NPF events reported in this study. Polluted atmospheres in Beijing indeed take place in presence of stagnant metrological conditions or under a dominant south or southwest wind.

*Other points which need to be addressed include:*

*(a) The introduction lists a number of organic acids as examples of vehicle-emitted organic compounds. Most of these have far more major secondary sources, or are present in cooking emissions, with little if any arising from road traffic.*

**Response:** Agree, the authors will revise the part as: "Urban street canyons provide semi-enclosed environments trapping vehicle exhausts that contain aromatic and aliphatic hydrocarbons, $SO_2$, NOx, amines, black carbon, etc. (Pierson et al., 1983; Stemmler et al., 2005; Burgard et al., 2006; Liu et al., 2008; Buccolieri et al., 2009; Sun et al., 2012; Gentner, et al., 2012)"

*(b) Some ill-informed statements are made about the (currently uncertain) effects of exposure to ultrafine particles. These particles do not lead to "destruction of the respiratory system" and the statement that "newly formed particles inside a street canyon may become toxic when vehicle-release organics is involved in the nucleation process" is not supported by references.*

**Response:** The authors will revise as: "Ultrafine particles (<100 nm) have been reported to have adverse human health impacts through the deposition in the pulmonary region and penetration into the bloodstream (Oberdörster et al., 2004; Schlesinger et al., 2006; Zhang et al., 2012, 2015)." "In addition, the newly formed

particles inside a street canyon might become toxic when vehicles released organics were involved in the nucleation process (Sgro et al., 2009; Gualtieri et al., 2014)."

*(c) There is no information on quality assurance beyond an intercomparison between the two FMPS, and no consideration of how size-dependent particle losses in the inlet system affect measured size distributions.*

**Response:** As reported by Zimmerman et al. (2015), an independent measurement of CPC simultaneously with FMPS can be used to accurately correct the FMPS data including the size-dependent particle loss. The reference has been cited in the manuscript. In this study, the FMPS data perform a three-step calibration as clarified in revised supplementary:

"First step: When the two FMPS operated side-by-side during 12-17 April 2012 for inter-comparison, we get the correction factors for one FMPS (Table S1). The FMPS which operating simultaneously with a CPC at the street site afterwards was used as the reference to correct the other.

Second step: The FMPS operating simultaneously with a CPC at the street site was then processed the second-step correction proposed by Zimmerman et al. (2015). The ratio of $FMPS_{total}/CPC_{total}$ was calculated to be 1.28.

Third step: The calculated ratio of 1.28 and correction method in Zimmerman et al. (2015) was used to correct the FMPS data at the rooftop site."

The particle loss for >8 nm particles were undetectable in 2.8 m sampling lines. This will also be added in the revision.

*(d) Equation (3) differs from that in the nucleation protocol paper of Kulmala et al. (2012) by a factor of two, which needs to be explained.*

**Response:** The equation (3) in this study was exactly same as reported by Dal Maso et al. (2005) and Kulmala et al. (2005). The references have been cited in this study. The authors don't understand the equation presented by Kulmala et al. (2012) and prefer to use the equation widely adopted in literature.

*(e) A clear definition is needed for the "maximum increase of nucleation mode PNC (NMIoNP)" which is much used in the data analyses.*

**Response:** "the net maximum increase of nucleation mode PNC (NMIoNP) (revise to NMINP, the net maximum increase in nucleation mode PNC in revision)" is calculated as "$N_{8-20nm}(t_1)-N_{8-20nm}(t_0)$". $t_0$ is set as the time when the apparent NPF started to be observed and $t_1$ as the time when the nucleation mode PNC reaches the maximum value. This will be added in revision.

*(f) The authors should establish that their Class II particles arise from an NPF event, rather than an emission source.*

**Response:** In revision, the authors will reorganize the evidences to confirm Class II particles to be a regional NPF event, i.e.,

1) As reviewed by Vu et al. (2015), the particle number size distribution (PNSD) of emission source (e.g., traffic emissions, industrial emissions, biomass burning, cooking) character the typical peak number mode, such as at 30 nm, 50 nm, 70-80 nm, 120-140 nm, et al. When the NPF events in Class II occurred in our study, the nucleation mode particles overwhelmed and other particle modes were negligible.

2) Similar to Class II NPF events with the particle growth to be undetectable presented in this study, extremely low growth rate of newly formed particles (~ 1 nm h$^{-1}$) in Beijing was also previously reported by Wehner et al. (2004). In our unpublished data, the authors simultaneously observed Class II NPF events and NPF events with extremely low growth rate at ~240 km distance (shown in Fig. R3, the case will also be presented in Supplementary). In the last three years (data unpublished), we had simultaneous observations of NPF events at 100-500 km distance. The authors obtained six cases based on simultaneous observations at two locations, i.e., one case featured by Class II NPF vs Class II NPF, four cases featured by Class II NPF vs NPF with an extremely low growth rate, one case featured by Class II NPF vs NPF with "banana shape" particle growth.

3) In this study, the duration period of Class II events lasted for 4-8 hours with the wind speed >4m/s. The authors strongly believed that they should be considered as regional NPF events.

[Figure]

Fig. R3 Simultaneous observed Class II NPF event and NPF event with extremely low growth rate at ~240 km distance.

[revised manuscript text omitted]

---

## Author Response (AR2)

Dear Editor,

We have made some revisions to clarify the issues according to reviewer's comments, and resubmitting the revised manuscript for publishing in the Journal.
Thank you and best regards.

Your Sincerely,

Xiaohong Yao

Key Lab of Marine Environmental Science and Ecology
Ocean University of China
Qingdao 266100
China

*Report #1*

*Anonymous Referee #3*

***Suggestions for revision or reasons for rejection (will be published if the paper is accepted for final publication)***

*The manuscript was improved to some extent in the revision. However, there are some arguments/conclusions that needed to be further clarified.*

**Response:** The authors thank the reviewer's comments and try our best to respond and revise our manuscript accordingly.

*1. The authors compared the number of their observations with the case of the gravitational wave in 2016 in response to reviewer 1. I have to say that the comparison is nevertheless meaningless and they cannot be compared in any way. The reviewer 1 raised a very important point that the authors aught to fully address. The authors present three evidences that support their conclusion. However, evidence one is ambiguous, i.e., the authors didn't give any strong evidences that micro-meteorology played a minor role in their case and they just presented a general experienced judgement which was not supported by any data. Evidence two and three are not evidences at all. They are just a representation of the data or to be straight, they are just what the data look like. I have the same concern as reviewer one if a general conclusion can be drawn based on just one or two observations.*

**Response:** Considering the comments, we further improve our manuscript accordingly. We agree more data are needed to draw a general conclusion. In abstract, we added "However, the number of datasets used in this study is relatively small and

larger datasets are essential to draw a general conclusion".

To address possible influence of micro-meteorology, in page 10, line 228, we added "Micro-meteorology at street sites may cause accumulation or dilution of atmospheric particles." To better defend our evidences presented to support reduced and enhanced NPF occurring in spring and winter times, respectively. We made a few revisions as listed below:

Page 11, lines 231-244, it was revised as "Evidences 2-3 were obtained by subtracting the PNC of different sized particles at the rooftop site from the corresponding one at the street site and the size-segregated difference of PNC between the two sites in April was thereby calculated (Fig. 4). The difference was largely negative for particles <14 nm during the NPF periods on 25 and 27 April (solid lines) against the positive difference of Aitken mode particles. In contrast, such a difference was slightly positive for particles <14 nm during the non-NPF days and during the morning rush hours on 25 and 27 April prior to the occurrence of NPF events (Fig. 4, dash lines), because of increasing contributions from on-road vehicles at the street site as well as the accumulation effect associated with micro-meteorology at the street site. The same accumulation effect should theoretically exist during the NPF periods on 25 and 27 April, but the observed result showed the reverse. We thus obtained Evidence 2, i.e., the negative difference of nucleation mode particles on NPF days against the positive difference of those on non-NPF days. Considered the positive difference of Aitken mode particles during the NPF periods on 25 and 27 April (solid lines), it can be inferred that micro-meteorology favored an increase of nucleation mode particle number concentration at the street site. However, the observed result was contradictory to the interference. We thus obtained Evidence 3, i.e., the negative difference of nucleation mode particles on NPF days against the positive difference of those Aitken mode particles."

Page 12, lines 269-270, we added "The different increasing patterns of nucleation mode particles at two sites strongly indicated that they were subject to different NPF mechanisms."

Page 13, lines 277-278, we added "The measured concentration of BC was thereby tried for deduction and much poor correlation was obtained due to one-minute time resolution can't allow successfully capturing vehicle spikes varying in a few seconds (see supplementary)."

Page 13, lines 292-293, we added "Theoretically, larger condensation sinks should cause stronger scavenging effects at the street site during the NPF periods in the wintertime than in the springtime. However, the difference of nucleation mode particles in the wintertime was positive."

Page 13, line 297, we added "The reverse was true for 20-80 nm particles."

Our Evidences 2-3 are unique and are self-independent on each other. We strongly believe that the revision has been demonstrated this issue clearly.

*2. The authors emphasized many times on the importance of the instrument they employed. As far as I know, the fast mobility particle sizer has limited ability and limited applications so far. The data might have high uncertainties when the measurements were taken too fast compared to the traditional SMPS in term of accuracy of the sizes and concentrations.*

**Response:** Over sixty publications related to FMPS measurements can be searched from popular databases, e.g., Web of Science, Google Scholar, etc, and dozens of these were published on international-recognized journals including Atmospheric Chemistry and Physics, Environmental Science and Technology, Journal of Geophysical Research (Atmosphere), Aerosol Science and Technology, Atmospheric Environment, Science of Total Environment and Environmental Pollution, etc. Building on these publications, the measurements made by two FMPS in this study were processed the correction for accuracy in size and concentration using recently developed approaches by Zimmerman et al. (2015).

Any instruments had advantages and disadvantages. This study fully took advantage of the high time resolution instrument and minimized the uncertainty on the measured number concentration and size. However, we delete a few statements on the importance of the instrument in revision.

*3. The reginal nucleation event by its definition is an event representing regional phenomenon. I am quite doubtful that a lot of such events are observed in Beijing at the sites as used in this study. Otherwise the measurements only represent nucleation events in a very local scale, say the environment around the site.*

**Response:** NPF events were widely studied in Beijing since 2004. Previous studies showed a high frequency (~40%) of NPF events during spring and winter in the urban air of Beijing (e.g., Wu et al., 2007; Wang et al., 2013; Wang et al., 2017). In this study, the NPF occurrence frequency was 44% in the springtime and 50% in the wintertime, which was consistent with previous studies. Class II NPF events were previously assumed to occur in a very local scale because of short databases, particularly lack of simultaneous observations in polluted marine and coastal atmospheres. The data presented in the supplementary clearly showed that Class II NPF events can occur regionally.

Class II NPF events were confirm to occur regionally because 1) the duration period of NPF events lasted for 4-8 hours with the wind speed >4m/s, 2) the initial diameter of observed new particles was about 10 nm, needing at least dozens of minutes or 1-2 hours growing from the sub-3 nm clusters (Kulmala et al., 2007, 2013). The long

lasting time indicated the NPF events occur in a region range over tens to hundreds of kilometers. In revision, Class II NPF events to occur regionally has been well defended.

*4. The authors used the proxy to calculate sulfuric acid concentrations and discussed a little bit the role of sulfuric acid. However I didn't see any presentation of the concentration or the detailed application of sulfuric acid concentrations. As discussed on p17, it is not clear when sulfuric acid should play a role quantitatively in Fig.9. The arguments seem to be speculative but I believe the authors should present a clearer picture to illustrate such an important point.*

**Response:** In revision, the authors add some detail descriptions on the sulfuric acid, e.g., Page 14, lines 314-315, "the calculated concentrations of sulfuric acid were between $2 \times 10^6$ and $2 \times 10^7$ cm$^{-3}$ during NPF periods, which was comparable with previous observations in Beijing (Yue et al., 2010; Zheng et al., 2011; Wang et al., 2017)." Lines 320-322, "However, the number of datasets in this study was too small to gain a reasonable equation to link the calculated concentrations of sulfuric acid with FRs."

In addition, we added a new Fig 9b. Fig 9b showed that NMINP was determined by SO$_2$ to some extent, which further supported our argument on the importance of sulfuric acid on NMINP.

*5. There are still lots of typos and ill-sentences that need to be proofread. A few examples:*
*1) Some of the picture captions are still not well written. For example, "the difference in number concentration" in Fig. 4 should be clearly defined in the caption. The readers don't have to refer to the text when read the figure; wrong link provides in Fig. 1.*

**Response:** The authors have rewritten the caption of Fig.4 and check the other captions. The 3D-view figure in Fig. 1 was downloaded in the year of 2012. Now the map has been updated in revision.

*2) P5 and others, a lot of times "on day-day" was used, while "during day-day" was also used. I think it might be better to be consistent and "during day-day" might be better here.*

**Response:** Corrected.

*3) P6: we write "cut a length of 2.8 m" not "cut as 2.8m in length; "in the supplementary" is enough to use, "document" in the "supplementary document" seems to be redundant. I believe there are still a lot in the text that the authors need to carefully edit.*

**Response:** The authors corrected the errors and carefully edit the text accordingly.

**Reference:**

Kulmala, M., Riipinen, I., Sipilä, M., Manninen, H. E., Petäjä, T., Junninen, H., Dal Maso, M., Mordas, G., Mirme, A., Vana, M., Hirsikko, A., Laakso, L., Harrison, R. M., Hanson, I., Leung, C., Lehtinen, K. E. J., Kerminen, V-M., Toward direct measurement of atmospheric nucleation. Science, 318(5847), 89-92, doi: 10.1126/science.1144124, 2007.

Kulmala, M., Kontkanen, J., Junninen, H., Lehtipalo, K., Manninen, H. E., Nieminen, T., Petäjä, T.; Sipilä, M., Schobesberger, S., Rantala, P., Franchin. A., Jokinen, T., Järvinen, E., Äijälä, M., Kangasluoma, J., Hakala, J., Aalto, P. P., Paasonen, P., Mikkilä, J., Vanhanen, J., Aalto, J., Hakola, H., Makkonen, U., Ruuskanen, T., Mauldin III, R. L., Duplissy, J., Vehkamäki, H., Bäck, J., Kortelainen, A., Riipinen, I., Kurtén, T., Johnston, M. V., Smith, J. N., Ehn, M., Mentel, T. F., Lehtinen, K. E., Laaksonen, A., Kerminen, V. M., Worsnop, D. R.: Direct Observations of Atmospheric Aerosol Nucleation, Science, 339, 943-946, doi: 10.1126/science.1227385, 2013.

Riccobono, F., Schobesberger, S., Scott, C. E., Dommen, J., Ortega, I. K., Rondo, L., Almeida, J., Amorim, A., Bianchi, F., Breitenlechner, M., David, A., Downard, A., Dunne, E. M., Duplissy, J., Ehrhart, S., Flagan, R. C., Franchin, A., Hansel, A., Junninen, H., Kajos, M., Keskinen, H., Kupc, A., Kürten, A., Kvashin, A. N., Laaksonen, A., Lehtipalo, K., Makhmutov, V., Mathot, S., Nieminen, T., Onnela, A., Petäjä, T., Praplan, A. P., Santos, F. D., Schallhart, S., Seinfeld, J. H., Sipilä, M., Spracklen, D. V., Stozhkov, Y., Stratmann, F., Tomé, A., Tsagkogeorgas, G., Vaattovaara, P., Viisanen, Y., Vrtala, A., Wagner, P, E., Weingartner, E., Wex, H., Wimmer, D., Carslaw, K, S., Curtius, J., Donahue, N. M., Kirkby, J., Kulmala, M., Worsnop, D. R., and Baltensperger, U.: Oxidation products of biogenic emissions contribute to nucleation of atmospheric particles, Science, 344, 717-721, doi: 10.1126/science.1243527, 2014.

Yue, D. L., Hu, M., Zhang, R. Y., Wang, Z. B., Zheng, J., Wu, Z. J., Wiedensohler, A., He, L. Y., Huang, X. F., and Zhu, T.: The roles of sulfuric acid in new particle formation and growth in the mega-city of Beijing, Atmos. Chem. Phys., 10, 4953-4960, doi:10.5194/acp-10-4953-2010, 2010.

Wang, Z. B., Hu, M., Sun, J. Y., Wu, Z. J., Yue, D. L., Shen, X. J., Zhang, Y. M., Pei, X. Y., Cheng, Y. F., Wiedensohler, A. Characteristics of regional new particle formation in urban and regional background environments in the North China Plain. Atmos. Chem. Phys., 13(24), 12495-12506, doi:10.5194/acp-13-12495-2013, 2013.

Wang, Z., Wu, Z., Yue, D., Shang, D., Guo, S., Sun, J., Ding, A., Wang, L., Jiang, J., Guo, H., Gao, J., Cheung, H.C., Morawska, L., Keywood, M., and Hu, M.: New particle formation in China: Current knowledge and further directions, Sci. Total Environ., 577, 258-266, doi: 10.1016/j.scitotenv.2016.10.177, 2017.

Wu, Z., Hu, M., Liu, S., Wehner, B., Bauer, S., Maßling, A., Wiedensohler, A., Petäjä, T., Dal Maso, M., and Kulmala, M.: New particle formation in Beijing, China: Statistical analysis of a 1-year data set, J. Geophys. Res., 112, doi: 10.1029/2006JD007406, 2007.

Zheng, J., Hu, M., Zhang, R., Yue, D., Wang, Z., Guo, S., Li, X., Bohn, B., Shao, M., He, L., Huang, X., Wiedensohler, A., and Zhu, T. Measurements of gaseous $H_2SO_4$ by AP-ID-CIMS during CAREBeijing 2008 campaign. Atmos. Chem. Phys., 11(15), 7755-7765, doi:10.5194/acp-11-7755-2011, 2011.

Zimmerman, N., Jeong, C. H., Wang, J. M., Ramos, M., Wallace, J. S., and Evans, G. J.: A source-independent empirical correction procedure for the fast mobility and engine exhaust particle sizers, Atmos. Environ., 100, 178-184, doi: 10.1016/j.atmosenv.2014.10.054, 2015.

*Report #2*

*Anonymous Referee #2*

*Suggestions for revision or reasons for rejection (will be published if the paper is accepted for final publication)*

*In the revised manuscript, the authors have done a good job to fully address the reviewer's concerns. Especially, they have provided additional analyses to validate the NPF events observed at both the street and rooftop levels. The reviewer indeed agree with the authors that amines, HOMs or other organics may play important roles in the new particle formation events, during both nucleation and the following growth processes. However, the reviewer must point out that no direct observations of amines and HOMs were provided in this study. Therefore, the reviewer suggests the authors insert a sentence in the conclusion (end of L416) to state that "although further validations including direct measurements of amines and HOMs in the newly formed particles are still needed." After addressing the above issue, the reviewer will recommend the manuscript for publication in ACP.*

**Response:** The authors thank the reviewer's comment and have added the statement in the conclusion section.

---

## Author Response (AR3)

**Co-Editor Decision: Reconsider after minor revisions (Editor review) (09 Jun 2017) by Zhanqing Li**

**Comments to the Author:**

*I am satisfied with your responses to the comments. However, the paper needs professional editing to assure the use of more standard English to eliminate such terms as "for > 10 nm new particles" (may say "new particles of radius greater than 10 nm"), as well as a few grammar errors. Some shaded texts should also be cleaned. A very careful check of references is also necessary.*

**Response:**

Dear Editor,

According to your comments, all references have been double-checked through the manuscript and the language has been edited using a profession service. We are resubmitting the revised manuscript in a clean version. We hope that the revision can meet the high standard for publishing in your Journal. Thank you again.

Your Sincerely,

Xiaohong Yao

Key Lab of Marine Environmental Science and Ecology
Ocean University of China
Qingdao 266100
China